



# Exploring the role of snow metamorphism on the isotopic composition of the surface snow at EastGRIP

Romilly Harris Stuart[1,4], Anne-Katrine Faber[2], Sonja Wahl[2], Maria Hörhold[3], Sepp Kipfstuhl[3], Kristian Vasskog[4], Melanie Behrens[3], Alexandra Zuhr[5,6], and Hans Christian Steen-Larsen[2]

[1]Laboratoire des Sciences du Climat et de l'Environnement, UMR8212, CNRS – Gif sur Yvette, France
[4]Department of Geography, University of Bergen, and Bjerknes Centre for Climate Research, Bergen, Norway
[2]Geophysical Institute, University of Bergen and Bjerknes Centre for Climate Research, Bergen, Norway
[3]Alfred-Wegener-Institut Helmholtz-Zentrum für Polar- und Meeresforschung, Bremerhaven, Germany
[5]Alfred-Wegener-Institut Helmholtz Zentrum für Polar- und Meeresforschung, Research Unit Potsdam, Telegrafenberg A45, 14473 Potsdam, Germany
[6]University of Potsdam, Institute of Geosciences, Karl-Liebknecht-Str. 24-25, 14476 Potsdam-Golm, Germany

**Correspondence:** Romilly Harris Stuart (Romilly.Harris-Stuart@lsce.ipsl.fr)

**Abstract.**

Stable water isotopes from polar ice cores are invaluable high-resolution climate proxy records. Recent studies have aimed to improve knowledge of how the climate signal is stored in the water isotope record by addressing the influence of post-depositional processes on the surface snow isotopic composition. In this study, the relationship between changes in surface snow microstructure after precipitation/deposition events and water isotopes is explored using measurements of snow specific surface area (SSA). Continuous daily SSA measurements from the East Greenland Ice Core Project site (EastGRIP) situated in the accumulation zone of the Greenland Ice Sheet during the summer seasons of 2017, 2018 and 2019 are used to develop an empirical decay model to describe events of rapid decrease in SSA, driven predominantly by vapour diffusion in the pore space and atmospheric vapour exchange. The SSA decay model is described by the exponential equation $SSA(t) = (SSA_0 - 26.8)\, e^{-0.54t} + 26.8$. The model performance is optimal for daily mean values of surface temperature in the range $0\,°C$ to $-25\,°C$ and wind speed $< 6\,\mathrm{m\,s^{-1}}$. The findings from the SSA analysis are used to explore the influence of surface snow metamorphism on altering the isotopic composition of surface snow. It is found that rapid SSA decay events correspond to decreases in d-excess over a 2-day period in $72\,\%$ of the samples. Detailed studies using Empirical Orthogonal Function (EOF) analysis revealed a coherence between the dominant mode of variance of SSA and d-excess during periods of low spatial variability of surface snow over the sampling transect, suggesting that processes driving change in SSA also influence d-excess. Our findings highlight the need for future studies to decouple the processes driving surface snow metamorphism in order to quantify the fractionation effect of individual processes on the snow isotopic composition.

## 1 Introduction

The traditional interpretation of stable water isotopes in ice cores is based on the linear relationship between local temperature and first order parameters $\delta^{18}O$ and $\delta D$ of surface snow on ice sheets (Dansgaard, 1964). The second order parameter d-





excess (d-excess = $\delta D - 8 \cdot \delta^{18}O$) is a result of kinetic fractionation caused by different molecular diffusivities of oxygen and hydrogen and has traditionally been interpreted in ice core records as reflecting moisture source conditions (Merlivat and Jouzel, 1979). Many factors must be accounted for when reconstructing temperature in ice cores, including precipitation intermittency (Casado et al., 2020; Laepple et al., 2018), past variations in ice-sheet elevation (Vinther et al., 2009), sea ice extent (Faber

et al., 2017; Sime et al., 2013), and firn diffusion (Johnsen et al., 2000; Landais et al., 2006; Holme et al., 2018). In addition, recent studies have documented isotopic composition change during precipitation-free periods (Steen-Larsen et al., 2014; Ritter et al., 2016; Casado et al., 2018; Hughes et al., 2021), linked to synoptic variations in atmospheric water vapour composition and subsequent snow-vapour exchange (Steen-Larsen et al., 2014). Current research aims to quantify the influence of post-depositional processes on isotopic change of the surface snow (Steen-Larsen et al., 2014; Ritter et al., 2016; Madsen et al.,

2019; Wahl et al., 2021).

Surface snow undergoes structural changes, as grains form bonds, grow. This process is called snow metamorphism, which is active at the surface and at greater depths, depending on temperature (gradient) conditions (Colbeck, 1983; Pinzer and Schneebeli, 2009). A major change the snow is undergoing, is the reduction of the ice-air interface to reduce energy (Legagneux and Domine, 2005). The snow-air interface can be described by the widely used parameter SSA. It is assumed to be linked to

the optical grain size equivalent (Linow et al., 2012) and can be utilized as a measure for snow metamorphism (Cabanes et al., 2002, 2003; Legagneux et al., 2002). In this study we use SSA to describe the (rapid) change of surface snow as one measure for snow metamorphism.

This manuscript focuses on surface snow property changes after precipitation. We here explicitly refer to snow which is lying at the surface for an unknown amount of time and thus does not directly represent freshly precipitated snow. Fresh snow

crystals have a high value of SSA. After deposition of the crystals on the surface, the SSA rapidly decreases from its initial value due to crystal growth (Cabanes et al., 2002; Legagneux et al., 2004; Domine et al., 2007). The reasons for the SSA decrease are wind-driven fragmentation (Comola et al., 2017; Neumann et al., 2009), interstitial vapour diffusion in the pore space between snow crystals (Pinzer et al., 2012; Flin and Brzoska, 2008) and sublimation (Sokratov and Golubev, 2009).

Models can provide a quantitative description of the rapid SSA decrease after precipitation. Previous studies have pro-

posed SSA decay models using a combination of field measurements and controlled laboratory experiments (Cabanes et al., 2002, 2003; Legagneux et al., 2003, 2004; Flanner and Zender, 2006; Taillandier et al., 2007). While current versions of the so-called decay models exist, these are mostly based on lab-experiments and non-polar snow observations. Conditions for surface snow on polar ice sheets such as Greenland are however not necessarily comparable to other alpine regions. The dry-accumulation zone of the Greenland ice sheet has only small amounts of intermittent precipitation. Furthermore, the high-

latitude radiation budget is different than in other alpine regions.

Only few continuous datasets of daily SSA measurements exist from the remote regions of the polar ice sheets (Libois et al., 2014; Picard et al., 2014). While SSA observations from Greenland exist (Carmagnola et al., 2013; Linow et al., 2012), diurnal datasets covering multiple months and years provide a better foundation for understanding the relevance of snow metamorphism for ice core studies. In particular, studies of SSA and snow metamorphism from Greenland are relevant for isotope ice

core studies. This is because snow metamorphism is expected to influence the snow isotopic composition as documented in





laboratory studies (Ebner et al., 2017) and field experiments (Hughes et al., 2021). Nonetheless, few studies have focused on the direct relationship between physical snow properties, such as SSA and post-depositional changes in isotopic composition.

An SSA decay model optimized for Greenland conditions would provide a better quantitative foundation for a process-based understanding of surface snow metamorphism on Greenland. Furthermore, a quantitative description of Greenland SSA decay

would provide a basis to explore how snow metamorphism at the surface plays a role for the alteration of isotopic composition of Greenland snow after deposition.

In this manuscript the aim is to explore the behaviour of surface snow metamorphism on polar ice sheets using daily SSA measurements, and compare change in physical properties to the isotopic composition measurements. The primary focus is to document events where changes in SSA occur rapidly over a duration of a few days. We first identify events of rapid

SSA decreases (decays) and explore how the isotopic composition of the snow changes during these events. Using daily field observations of snow properties from Northeast Greenland during summer, events of rapid SSA decrease are used to 1) quantify and model surface snow metamorphism in polar snow and, 2) assess isotopic change during surface snow metamorphism. The data presented here has the potential to contribute to the understanding of the influence of post-depositional processes on physical and isotopic changes in the polar ice sheet surface snow. This allows for better understanding of snow properties at

remote regions of polar ice sheets, and contributes the interpretation of water isotopes in polar ice cores.

## 2   Study site and methods

### 2.1   EastGRIP site overview and meteorological data

All data used in this paper were collected as part of the Surface Program corresponding to the international deep ice core drilling project at the East Greenland Ice Core Project site (EastGRIP 75.65°N, 35.99°W; 2,700 m.a.s.l) during summer field

seasons (May-August) of 2017, 2018 and 2019. Meteorological data used for this study are from the Program for Monitoring of the Greenland Ice Sheet (PROMICE) Automatic Weather Station set up by the Geological Survey of Denmark and Greenland (GEUS) at EastGRIP in 2016 (Fausto et al., 2021). Mean weather conditions vary between sampling years, as outlined in Table 1. Instrument specifics can be found in Fausto et al. (2021). Mean summer surface temperatures for 2019 were -10.6±5°C, 5°C higher than 2017 and 2018. Westerly winds prevail, with mean wind speed of 4.5 m s$^{-1}$ (Madsen et al., 2019).

An Eddy-Covariance tower (EC) was set up at EastGRIP in 2016. The relevant variable measured from this system is latent heat flux (LHF) which is directly determined by the measurement of humidity fluxes between the surface and atmosphere. Positive LHF indicates upwards energy flux in the form of sublimation in Table 1. All field seasons had net sublimation, with the highest magnitude observed in 2019 (See Data Availability Section A).

### 2.2   Snow sampling procedure

Each summer season of 2017, 2018 and 2019 snow samples were taken once a day from May to August at 10 sampling sites, each marked by a stick, along a 90 m transect with 10 m spacing upwind of the EastGRIP camp to ensure clean snow (Fig.



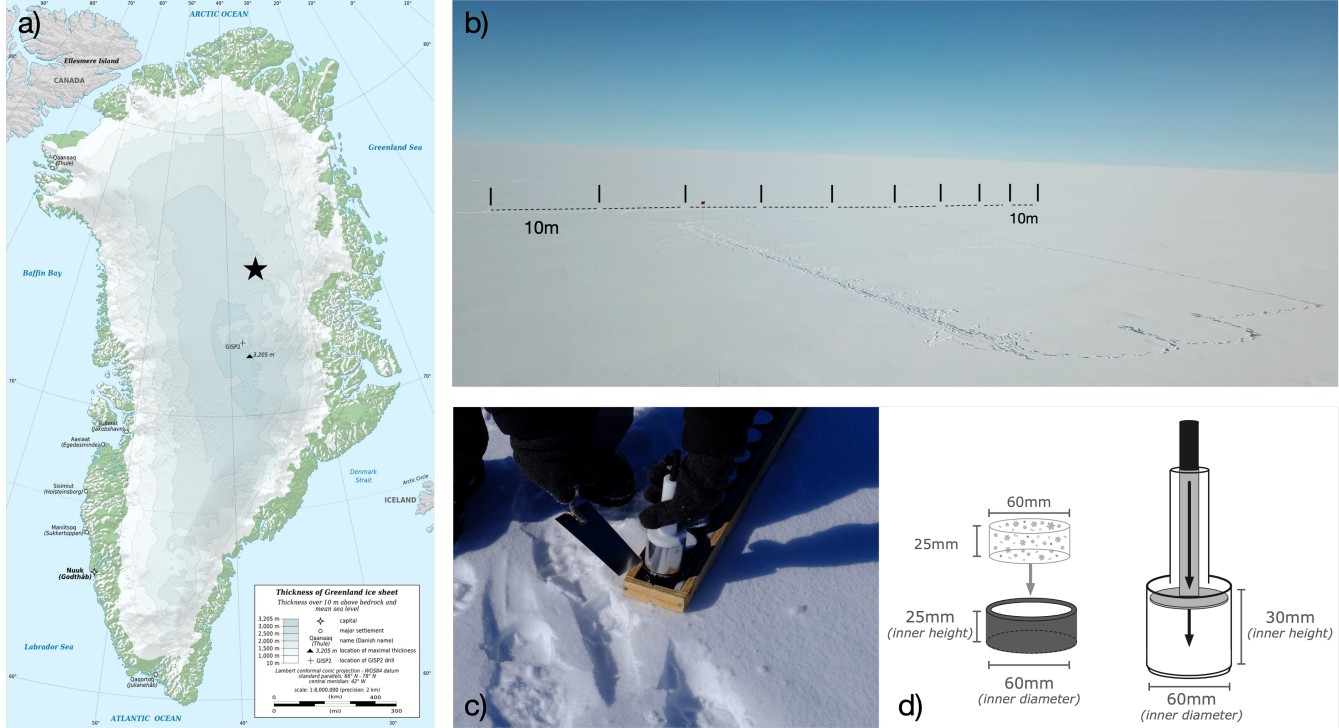

**Figure 1.** *SSA Sampling Procedure*

a) A map of Greenland with a black star indicating the EastGRIP site (Source: Eric Gaba – Wikimedia Commons user: Sting). b) A photograph of the clean snow area at the field site (Credit: Bruce Vaughn), with black lines indicating the SSA sampling transect with 10 m spacing shown as dashed lines. c) A photograph of SSA sampling cups (Credit: Sonja Wahl), and d) an illustration of the sampling device from Klein (2014).

1b). The specific dates for each season are given in Table 1. The precise location of each sample was marked by a small stick to ensure the adjacent snow is sampled the next day and to avoid sampling snow from different depths. A 6 cm diameter sampling device collected the top 2.5 cm of surface snow (Fig. 1c). Snow density is determined using the weight of each snow

sample with a known volume. At the start of each season, sticks were placed at each site and snow height was determined by the distance between the snow surface and top of the stick. Accumulation was calculated using the daily difference between measurements of snow height from each site. The resultant datasets consist of 10 daily measurements of three parameters, SSA, density and accumulation, over a 92, 100 and 66 day period for 2017, 2018 and 2019 respectively.

Although samples were measured each day, the exact sampling time varies. Snow sampled during the afternoon would have

had extended time exposed to solar radiation maximum, compared to snow sampled during the morning. Furthermore, the sampling time has implications for capturing precipitation events.





**Table 1.** *Weather statistics - 2017, 2018 and 2019*

The table present the mean and standard deviation for the weather variables, surface temperature, relative humidity with respect to ice, wind speed and latent heat flux. Surface temperature and wind speed are from the PROMICE weather station based on hourly measurements during the field seasons of 2017, 2018 and 2019. Relative humidity with respect to ice is calculated from vapour pressure of the air and saturation vapour pressure over ice. Latent heat flux is taken from the EC tower dataset.

|  | 2017 | 2018 | 2019 |
| --- | --- | --- | --- |
|  | *06/05 - 05/08* | *04/05 - 07/08* | *24/05 - 01/08* |
|  | *Mean* | *Mean* | *Mean* |
| **Surface Temperature** (°C) | -14.5 ± 6.2 | -15.76 ± 7.6 | -10.6 ± 5.4 |
| **Relative Humidity** (with respect to ice) (%) | 96 ± 15 | 96 ± 16 | 94 ± 14 |
| **Wind Speed** ($\mathrm{m\,s^{-1}}$) | 4.9 ±. 2 | 4.2 ± 1.9 | 4.5 ± 1.6 |
| **Latent Heat Flux** ($W\,m^{-2}$) | 1.3 ± 4 | 1.1 ± 3.9 | 2.6 ± 5.9 |

## 2.3 Ice Cube calibration

Each snow sample is placed into the Ice Cube sampling container below an Infra-Red (IR) laser diode (1310 nm), where the SSA is calculated based on IR hemispherical reflectance, explained in Gallet et al. (2009), while information on the Ice Cube device can be found in Zuanon (2013). Light penetration depth in snow of $200\,\mathrm{kg\,m^{-3}}$ is approximately 1 cm (Gallet et al., 2011), resulting in a measurement of the top <1 cm of each sample (Mean snow density at EastGRIP 2017, 2018 and 2019 = $294\,\mathrm{kg\,m^{-3}}$). The light reflected from the snow samples is converted into inter-hemispheric IR reflectance using a calibration curve based on methane absorption methods (Gallet et al., 2009). A radiative-transfer model is used to retrieve SSA from inter-hemispherical IR reflectance. To avoid influence from solar radiation, SSA was measured inside a ventilated white tent kept at temperatures between -5 °C and -10 °C. SSA measurements have an uncertainty of 10 % for values between 5-130 $\mathrm{m^2\,kg^{-1}}$ (Gallet et al., 2009).

## 2.4 Surface snow isotopes

Samples collected following the sampling procedure outlined in Section 2.2 were also used for isotopic composition measurements, resulting in 10 daily isotope measurements taking the average composition over the top 25 mm of snow. Each sample was sealed in polyethylene bags to avoid any air to equilibrate with the snow and affect the isotopic composition. All samples were kept frozen during transportation and storage.

After melting, each bag was shaken to ensure the isotopic composition of the sample is representative. 1.25 $\mu l$ of each sample was then pipetted into a vial ready for isotopic analysis. The snow samples were then analysed at Alfed Wegener Institute in Bremerhaven using a cavity ring-down spectroscopy instrument model Picarro L-2120-i and L-2140-i following the protocol of Van Geldern and Barth (2012). This technique is used to obtain measurements of $\delta^{18}$O and $\delta$D with an uncertainty of 0.15‰ and 0.8‰ respectively. d-excess is calculated by the equation $d-excess = \delta D - 8 \cdot \delta^{18}$O with a resultant uncertainty of





1‰. Observing relationships between our SSA and isotope data requires consideration for the depth offset between the SSA measurements and the isotopic composition measurement which measures the entire 2.5 cm snow layer.

## 2.5 Data analyses

### 120 2.5.1 Defining SSA decay events

This study focuses on the events where the SSA measurements decay rapidly over a duration of a few days. SSA decays are here defined as the events where the 2-day change of daily mean values are higher than a given threshold. This threshold is the same value for all years and is calculated based on the 10th percentile of the decays and set at -13 $m^2kg^{-1}$ 2-day$^{-1}$. If the daily mean changes over a 2-day period is higher than the threshold, then this period is selected as a rapid SSA decay event. The
duration of the event is set to start at the rapid decay and end on the day when the mean SSA measurements increase (rather than decrease) again.

### 2.5.2 Modelling surface snow metamorphism

An empirical decay model is constructed building upon previous studies (Cabanes et al., 2002, 2003; Flanner and Zender, 2006; Legagneux et al., 2002, 2003; Taillandier et al., 2007). This model uses continuous daily SSA measurements from EastGRIP to
describe the behaviour of surface snow SSA in polar summer conditions. The post-precipitation decreases in SSA are hereafter referred to as decays.

$$SSA(t) = SSA_0 \, e^{-\alpha t} \tag{1}$$

Eq. (1) is proposed by Cabanes et al. (2003) as the most accurate description of SSA decay, where $SSA_0$ is the initial SSA value, $\alpha$ the decay rate. To best describe grain coarsening and the processes of sublimation and deposition driving mass
redistribution of a new snow layer, days with mean wind speeds above $6 \, m \, s^{-1}$ are removed to reduce the influence of wind redistribution. Individual sample analysis is preferentially used to avoid daily mean values possibly attenuating any signals due to spatial variability in surface snow age. Aged snow patches are expected to respond differently to surface processes than new snow patches due to different original crystal structures at the start of events.

## 3 Results

### 140 3.1 SSA decay events

SSA data collected at EastGRIP indicate continuous changes in the physical structure of the snow crystals during all sampling seasons, with both temporal and spatial variability. The temporal SSA variability shows changes in physical snow structure with peak values closely associated with precipitation and decreases due to post-depositional re-working of the snow. Summer seasonal SSA evolution is presented in Fig. 2 for 2017, 2018 and 2019 with each faded line representing individual samples



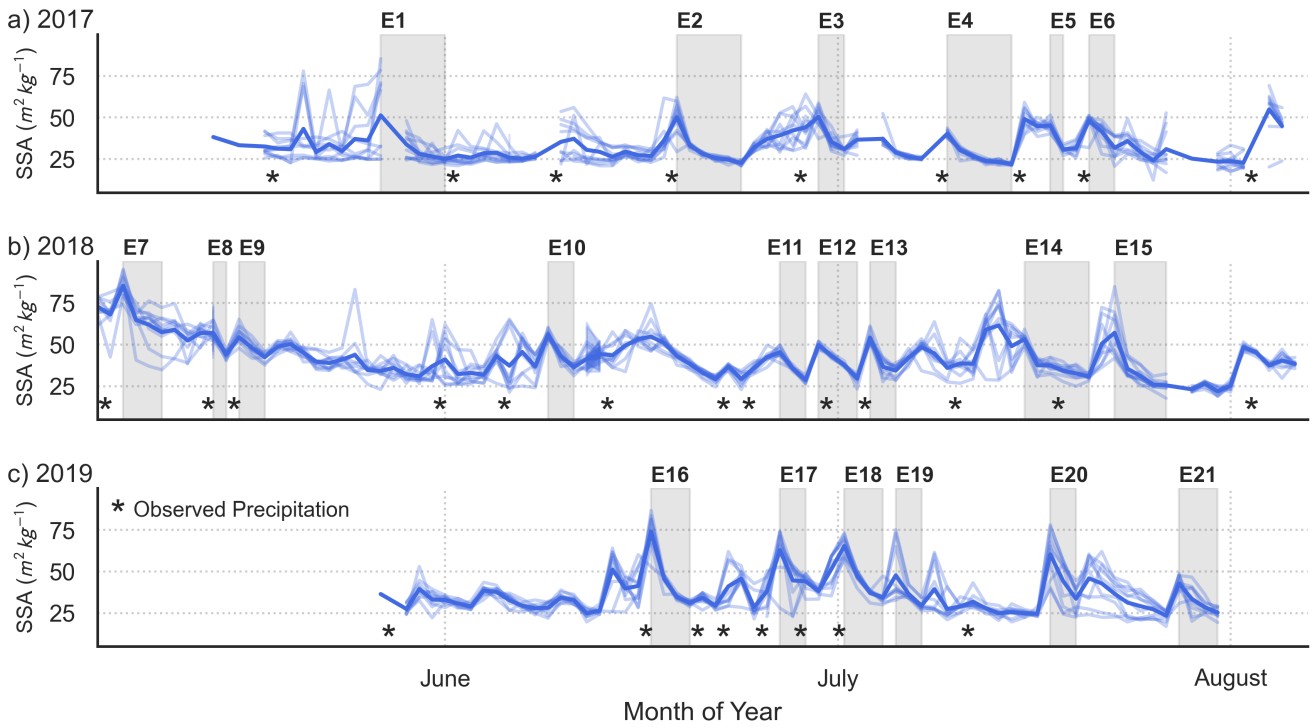

**Figure 2.** *SSA Timeseries 2017, 2018 and 2019*

SSA time-series between May and August for a) 2017, b) 2018 and c) 2019. Faded lines represent the 10 individual samples from the 90 m sampling transect, while the bold line shows the daily mean values. Gaps in the timeseries represent missing data. Grey bars highlight the periods of decrease in SSA defined by the threshold algorithm for each year. Six decrease events are observed in 2017 and 2019, while nine are observed in 2018. Decrease events are interpreted as rapid grain growth due to snow metamorphism, and stars indicating days with precipitation.

(10 per day), and the bold line showing the daily mean. Spatial variability between sites is most prevalent when there are high SSA values, indicating fresh snow.

A total of 21 rapid SSA decay events are identified, with 6, 9 and 6 events for 2017, 2018 and 2019 respectively. Grey bars in Fig. 2 highlight events defined by the decrease threshold. Maximum SSA values for 2018 and 2019 are $92\,m^2\,kg^{-1}$ and $82\,m^2\,kg^{-1}$ respectively, while during 2017 there are only two instances of daily mean SSA being above $60\,m^2\,kg^{-1}$.

A visual inspection of the decay events in Fig. 2 indicates a relationship between initial SSA and subsequent magnitude of decrease. To test whether the mechanisms of decay are consistent throughout events, observed SSA decays are analysed to construct an empirical model.





## 3.2 EastGRIP SSA decay model

Continuous SSA measurements allow for the construction of an empirical model to describe SSA decay at EastGRIP through
time while exposed to surface processes. All samples of defined SSA decrease events defined in Section 3.1 are used to
quantify surface snow metamorphism. For all events with mean temperature above -25 °C, the mean SSA of the final day is
around  $30 \, \mathrm{m^2 \, kg^{-1}}$ (referred to as the background decay state). A relationship is observed between the rate of SSA decay is
closely linked to the SSA value at the start of each event (initial SSA vs. magnitude of decrease during the decay period $r^2$ =
0.4) (Fig. 3), suggesting the rate of change is proportional to the absolute value, as described by exponential decay law.
SSA decay rate is quantified by plotting the rate of change in SSA per day against the absolute SSA value for all 10 sampling
sites for all events (Fig. 3a). We observe a linear relationship between the rate of change in SSA per day ($\Delta$SSA) and SSA.
Outliers are measurements from days with mean air temperature below -25 °C as highlighted in Fig. 3a. This observation is in
agreement with theoretical understanding of snow crystal formation transitioning from dendrites to columns at approximately
-22 °C (Domine et al., 2008). We therefore define the SSA decay model for a temperature range between -25 °C and 0 °C and
daily mean wind speeds below $6 \, \mathrm{m \, s^{-1}}$ based on hourly averaged values.

  Constructing the SSA decay model for EastGRIP is based on the differential equation for the linear relationship between
$\Delta$SSA and absolute SSA which is defined as Eq.(2). Solving the differential with respect to time (t), produces the SSA decay
model defined as Eq. (3), which follows the equation structure from Eq. (1).

$$\frac{dSSA}{dt} = -0.54 \, SSA + 14.69 \qquad (2)$$


$$SSA(t) = (SSA_0 - 26.8) \, e^{-0.54t} + 26.8 \qquad (3)$$

  Where SSA(t) is the SSA measurement at a given time, $SSA_0$ is the initial SSA value, and -0.54 $\mathrm{m^2 \, kg^{-1} \, day^{-1}}$ is the decay
rate ($\alpha$), as defined by the slope of Eq. (2). To account for a non-zero decay constant, the value 26.8 $\mathrm{m^2 \, kg^{-1}}$ is defined by the
value of $x$ when the linear regression crosses the y-axis (Fig. 3a). The SSA decay model describes rapid decrease in SSA based
on empirical data from EastGRIP, Greenland.

### 3.2.1 Model evaluation

Model performance is tested by comparing daily predicted decrease to the 10 daily observations. Model-data residuals for
daily data are normally distributed, suggesting no systematic errors in model predictions. Figure 3 shows the construction of
the model (Fig.3,a-b) and prediction of SSA decay (Fig.3,f-h). Note that events are here named E1,E2... consistent with Fig. 2
and also listed in Table A1.
  There is a minor tendency for the model to underestimate the SSA decrease and thus overestimate the predicted values of
SSA as seen in Fig. 3b. Model limitations are most evident during the first day, as seen in Fig. 3, where the modelled decay





**Figure 3.** *Decay Model Construction and Predictions*

All samples for all events are included in plot a) showing the relationship between the rate of change in SSA per day ($\Delta$SSA day$^{-1}$) against the daily absolute values. Points are coloured by the daily mean surface temperature. The linear regression is based on values for surface temperatures between -25 °C and 0 °C, and daily mean wind speeds below 6 m s$^{-1}$. b) shows a comparison between the model predicted SSA values using Eq. (3), against the SSA observations. The marker colour represents the day of the events (DOE). Marker style represents the sampling year to assess inter-annual variability for 2017 (o), 2018 (x) and 2019 ($\square$). c), d) and e) show all included events in full-line and f), g) and h) show the model predictions as the dashed line. E1-E21 refers to events as listed in Table A1. Missing data day-1 E1.





consistently underestimates the magnitude of decrease. The model has limited ability to predict observations below in the lower range of SSA observations as seen in Fig. 3f, g and h, where the modelled and observed values are compared for each event.

Following our definition in Section 3.1 the events have an extent of 2-5 days. To assess model performance in predicting magnitude of SSA decrease for events of different time periods, we compare the predicted versus measured SSA. For rapid events lasting 2-days the model tends to underestimate the rate of decrease. This is most apparent on Day 1 (24h after peak) for 2017 and 2018, while for 2019, Day 1 SSA is accurately predicted, with residuals increasing on Day 2. In comparison, events lasting 5-days show an underestimation for 2017 with negligible daily change in residuals, while the model overestimates the

decay rate of E14 in 2018. However, field documentation suggests intermittent snow fall during Day 2 of E14, causing increase in SSA. Consideration for environmental context is explored in Section 3.2.2. E16 is characterised by the highest initial SSA values, and the largest residuals, suggesting the model is limited at very high initial SSA values.

The model requires only initial SSA as a parameter and predicts SSA decrease at EastGRIP within the defined conditions with an averaged root mean squared error (RMSE) of $5.6\,\mathrm{m^2\,kg^{-1}}$ when considering all sample sites individually. The

model predicts SSA decay over 2-5 day periods ($r^2 = 0.89$), with the highest RMSE of $6.17\,\mathrm{m^2\,kg^{-1}}$ for 2019 compared to $4.97\,\mathrm{m^2\,kg^{-1}}$ and $4.72\,\mathrm{m^2\,kg^{-1}}$ for 2017 and 2018 respectively. The model adequately predicts rapid SSA decay at EastGRIP within the temperature range, while for colder temperatures, the decay rate is the same but the intercept is significantly higher (Fig. 3a). Overall, for all included events during the three sampling years, behaviour of SSA decay is clearly captured by the model (Fig. 3c,d and e). Exploring temperature conditions alone we find that the model performs well when daily mean surface

temperatures are between -25°C and 0°C.

### 3.2.2   Environmental conditions during SSA decay events

Intuitively, environmental conditions would be considered to play a role for surface snow metamorphism and the rate of SSA decay. To explore this, hourly weather measurements from the PROMICE AWS and field report weather observations are analysed to provide environmental context to SSA decay events. Weather station data shows no systematic influence of basic

weather variables, relative humidity, surface temperature and wind speed on the model-data residuals, with linear regressions resulting in $r^2 < 0.1$ for all variables. An overview of event conditions using field observations are presented in Table A1. Temperatures below -25°C are characterised by the same slope defined by the model ($-0.54\,\mathrm{m^2\,kg^{-1}\,day^{-1}}$), but with a significantly higher intercept of $29\,\mathrm{m\,s^{-1}\,day^{-1}}$ compared to $14.7\,\mathrm{m\,s^{-1}\,day^{-1}}$ for temperatures above -25°C. Significant wind drift is expected when hourly mean wind speed exceeds $6\,\mathrm{m\,s^{-1}}$, which happens during 144 days out of the total 258 sampling

days from 2017, 2018 and 2019. Results indicate weather has no systematic influence on SSA decay during the first 2-5 days exposed at the surface, and that conditions vary for each event. The model is able to predict all defined decay events between -25°C and 0°C, indicating mechanisms of decay are the same. Daily mean values are more accurately predicted by the SSA decay model than individual sample sites due to snow surface variability. In-homogeneous surface snow is especially important to consider for isotopic composition, because there is potential for samples to contain snow from different precipitation and/or

deposition events.




### 3.2.3 Surface snow spatial variability

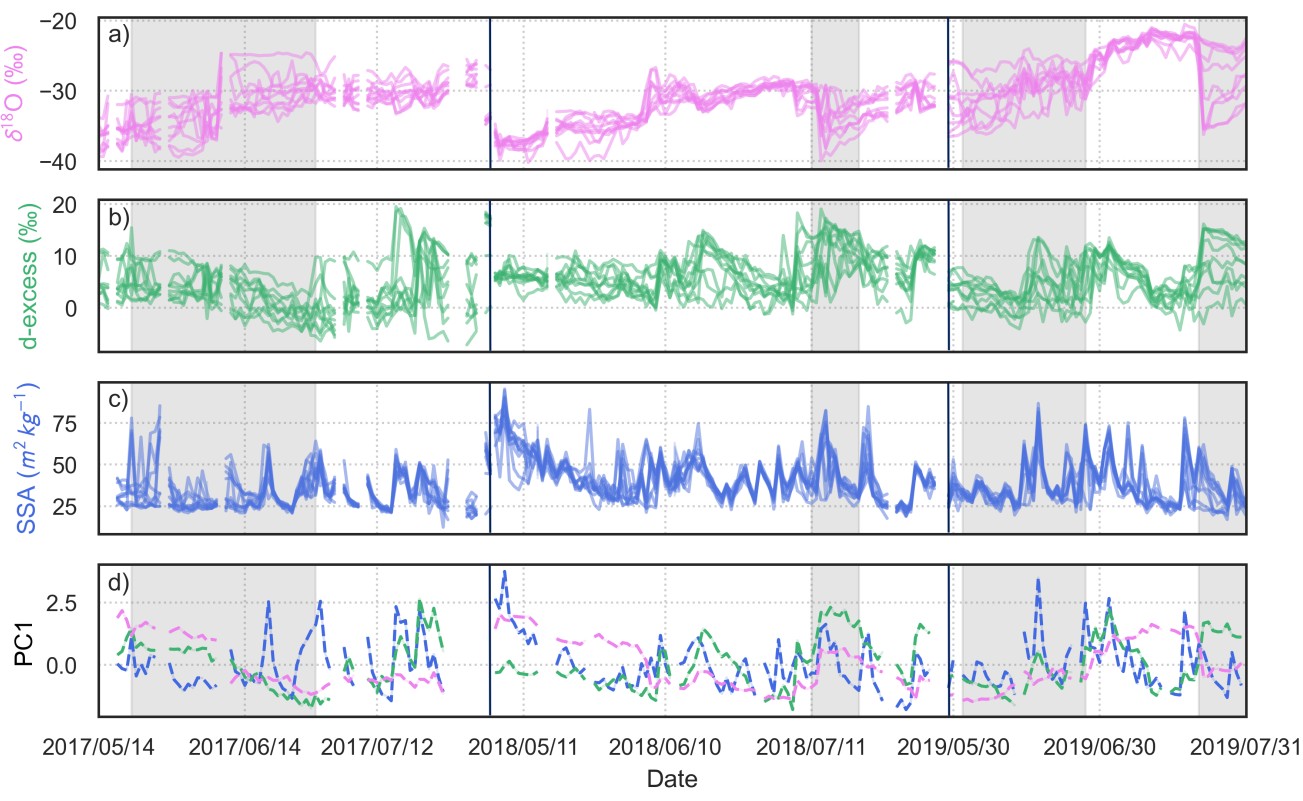

**Figure 4.** *Timeseries of snow isotopes and SSA*

Timeseries of $\delta^{18}$O (a), d-excess (b) and SSA (c) for 2017, 2018 and 2019 sampling seasons. d) shows the principle components of each parameter with colors corresponding to the color used to show absolute values. The black vertical lines indicate a break in the x-axis. Each faded line represents individual sample site values, and the thick line is the daily mean. Grey shaded regions indicate periods of high spatial variability in isotopic composition.

The characterization of the SSA decays provide a basis to explore how snow metamorphism of surface snow plays a role for the alteration of isotopic composition of Greenland snow after deposition. A recent study at EastGRIP has shown the significant in-homogeneity in surface snow due to post-depositional reworking of the snow (Zuhr et al., 2021). The focus for this manuscript is to identify signal coherence between physical properties and isotopic composition of surface snow subject to precipitation/deposition and post-depositional processes. Autocorrelation analysis shows that isotopic composition values are spatially decorrelated after 10 m ($r^2 < 0.3$ after 10 m). Therefore, to avoid attenuation of isotopic signal, each sample is treated as independent. Isotopic composition is measured from each SSA sample containing snow from the top 2.5 cm of the snow surface, potentially containing snow deposition layers from multiple precipitation events. Surface heterogeneity is considered





by using Empirical Orthogonal Function (EOF) analysis to determine the dominant mode of variance for each sampling year. Figure 4 shows timeseries of $\delta^{18}$O (a), d-excess (b) and SSA (c) with faded lines showing each sample site. The first principal components (PC1) of $\delta^{18}$O, d-excess and SSA are presented in Fig. 4d. All parameters continuously change throughout the field seasons of 2017, 2018 and 2019. Isotopic composition measurements (Fig. 4a, b) have larger spatial variability than SSA (Fig. 4c).

Inter-annual variability is observed in $\delta^{18}$O, with seasonal mean values of -31.6‰, -32.7‰ and -27.3‰ for 2017, 2018 and 2019 respectively (Fig. 4a). Note that the 2019 field season started approximately 15 days later than 2017 and 2018, resulting in a bias towards mid-summer conditions. Throughout the season $\delta^{18}$O follows a gradual increasing trend from May to August. Some cases of abrupt decreases (-10‰) are observed in the late summer, for example at July 12th in 2018 and July 25th in 2019. No clear seasonal trend is observed in d-excess (Fig. 4b) but with periods of gradual decreases. Total daily spread in
$\delta^{18}$O and d-excess is 15‰.

During 2017, 2018 and 2019 SSA has one dominant mode of variance (PC1) explaining 61%, 77% and 72% of the total variance in the respective datasets. PC1 of $\delta^{18}$O explains 69%, 83% and 75% of the total variance for the respective years. While PC1 of d-excess explains 47%, 51% and 60% for 2017, 2018 and 2019 respectively. PC1 of $\delta^{18}$O and d-excess show strong coherence from May to early June during 2017 and 2018, while for the second half of the season, and throughout 2019,
PC1 of d-excess corresponds to PC1 of SSA (Fig. 4d).

Surface variability due to post-depositional reworking of the snow is shown by a wide spread in SSA values during a given day. Time periods with low spatial variability indicate largely homogeneous snow cover over the transect, shown in Fig. 4 as shaded regions. High variability is defined by periods where the 5-day running-mean of spatial variance in $\delta^{18}$O is greater than one standard deviation. During periods of low spatial variability in isotopic composition, there is greater coherence between
PC1 of SSA and PC1 of d-excess, due to a reduction of noise in the dataset. PC1 of SSA and d-excess show a coherence during 2018 and 2019 seasons, while the signal is less clear during 2017 (Fig. 4b). However, the reduced signal coherence is concurrent with high spatial variability in isotopic composition.

A clear relationship between PC1 of SSA and PC1 of d-excess is observed when there is a relatively homogeneous snow layer over the sampling transect, defined by low spatial variance in $\delta^{18}$O.

### 3.2.4 Isotopic change during decay events

Using all 10 sample sites as independent values, the behaviour of isotopes during defined SSA decay events is analysed. To determine the isotopic change in the surface snow during rapid SSA decays, the rate of change in d-excess is plotted against the rate of change in SSA (Fig. 5). The change in SSA over a 2-day period is used. The daily mean change over the first 48h of each event is presented in Table 2.

In all events, the isotopic composition is observed to change, with $\delta^{18}$O increasing after 2-days but mostly limited to 1±1‰ mean increase, with the exception of E17 and E21 in 2019 (See Table 2). E17 is characterised by significant ground fog and snowfall during the event, while E21 has negative LHF (net-deposition) measured from the eddy-covariance system over the event. The percentage change of d-excess is an order of magnitude higher than $\delta^{18}O$ - expected due to the definition of d-excess





**Table 2.** *Table of isotopic change for decay events*

Behaviour of snow parameters during decay events are defined. The initial, 2-day and percentage change over a 2-day period are presented for $\delta^{18}O$, d-excess and SSA. All events used for the decay model are presented here, thus only the low-temperature event E7 (<-30 °C) is removed.

|  | $\delta^{18}O_0$ (‰) | | | d-excess$_0$ (‰) | | | SSA ($m^2\,kg^{-1}$) | | |
|---|---|---|---|---|---|---|---|---|---|
|  | Initial | Day-2 | 2-Day %change | Initial | Day-2 | 2-Day %change | Initial | Day-2 | 2-Day %change |
| E1 | -34.72 | -34.60 | 0.4% | 5.0 | 4.5 | -10.0% | 51.3 | 33.9 | -34.0% |
| E2 | -30.29 | -30.15 | 0.5% | 0.9 | -0.3 | -133% | 50.1 | 28.2 | -43.7% |
| E3 | -29.55 | -30.07 | -1.8% | -0.2 | -0.6 | -200% | 50.4 | 31.1 | -38.3% |
| E4 | -30.27 | -30.15 | 0.4% | 1.4 | -0.2 | -114% | 40.2 | 26.7 | -33.6% |
| E5 | -30.23 | -29.88 | 1.2% | 6.1 | 2.9 | -52.5% | 45.2 | 31.5 | -30.2% |
| E6 | -30.50 | -30.36 | 0.5% | 9.8 | 6.2 | -36.7% | 47.9 | 31.8 | -33.6% |
| E8 | -36.66 | NaN | NaN | 5.8 | NaN | NaN | 57.0 | NaN | NaN |
| E9 | -35.40 | -35.34 | 0.2% | 5.5 | 6.0 | 9.1% | 56.6 | 42.4 | -25.0% |
| E10 | -31.08 | -30.56 | 1.7% | 8.4 | 5.6 | -33.3% | 56.0 | 36.3 | -35.1% |
| E11 | -29.93 | -29.95 | -0.1% | 5.5 | 4 | -27.3% | 45.3 | 28.6 | -36.9% |
| E12 | -29.57 | -29.33 | 0.8% | 4.6 | 3.1 | -32.6% | 49.7 | 37.7 | -24.1% |
| E13 | -29.13 | -29.10 | 0.1% | 3.3 | 3.8 | 15.2% | 54.1 | 36.6 | -32.3% |
| E14 | -33.89 | -34.26 | -1.1% | 11.5 | 12.3 | 7.0% | 53.2 | 37.4 | -29.7% |
| E15 | -32.35 | -31.94 | 1.3% | 8.6 | 7.0 | -18.6% | 57.1 | 31.1 | -45.6% |
| E16 | -29.06 | -28.97 | 0.3% | 7.1 | 4.9 | -31.0% | 74.0 | 34.7 | -53.1% |
| E17 | -29.15 | -25.38 | 12.9% | 7.0 | 9.5 | 35.7% | 62.9 | 44.1 | -29.9% |
| E18 | -24.08 | -23.80 | 1.2% | 11.4 | 7.8 | -31.6% | 65.3 | 37.2 | -43.0% |
| E19 | -23.40 | -23.31 | 0.4% | 6.7 | 6.8 | 1.5% | 47.7 | 29.4 | -38.2% |
| E20 | -22.27 | -22.26 | 0.1% | 4.4 | 3.3 | -25.0% | 60.3 | 33.6 | -44.2% |
| E21 | -28.35 | -27.28 | 3.8% | 8.6 | 8.4 | -2.3% | 42.8 | 28.6 | -33.3% |

- and similar to SSA, with 14 out of 19 events showing a decrease in d-excess during the first 2-days of each event. Further

analysis looks specifically at the relationship between d-excess and SSA given the coherence observed between their PCs, and the significant change observed in Table 2.

SSA decreases by between 30 % and 53 % during the first 2-days, the largest change corresponding to the highest initial SSA value of $74\,m^2\,kg^{-1}$ as defined by the decay model. Using a significance level of 0.01, the relationship between change in d-excess after the second day of each event ($\Delta$d-excess) and change in SSA over the same time period ($\Delta$SSA) is assessed.

Events presented in Table 2 are shown in Fig. 5a. 72 % of decreases in SSA correspond to decrease in d-excess when treating each sample as an independent value. All large decreases in SSA correspond to high SSA values, as the model describes.




Increases in d-excess are observed at 12 samples sites, 6 of which are during 2017 and all correspond to initial d-excess values
< 5‰ (Fig. 5b). Thus suggests either low d-excess of deposited snow, or old snow that has been re-exposed. In addition, initial
d-excess is observed to significantly influence that magnitude of d-excess change over the subsequent 48h of rapid SSA decay
(Fig. 5a and b). The largest changes in d-excess corresponds to high initial d-excess values. Moreover, increases in d-excess
during rapid SSA decay follow very low initial d-excess values. In summary, in 72% (78 out of 108 samples) of cases decreases
in SSA correspond to a decrease in d-excess of the snow sample during the first 2-days. Moreover, the magnitude of change in
d-excess during rapid SSA decay shows a weak but significant dependence on the initial d-excess signal.

Significance of change in SSA and d-excess during events is tested by comparing the difference between the means of
daily changes for event and non-event periods using a t-test with 0.01 significance level. Background variability in d-excess
is $0.1\pm2.5‰$ for non-event periods, compared to $-0.4\pm2‰$ for events alone. Similarly for SSA, non-events daily change is
$0.04\,m^2\,kg^{-1}$ compared to $-7.7\,m^2\,kg^{-1}$ for events. SSA decay events exhibit significant difference in distribution to non-event

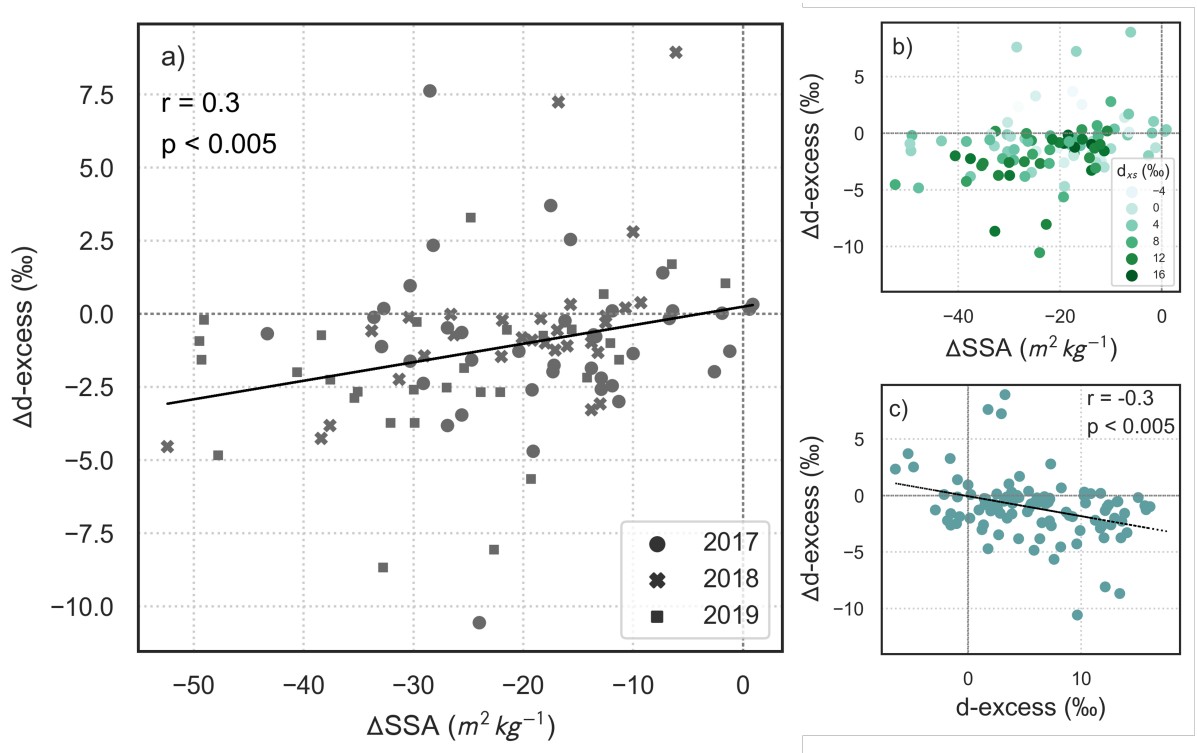

**Figure 5.** *Relationship between SSA and d-excess after the second day of each event*

The relationship between the rate of change in SSA ($\Delta$SSA $2days^{-1}$) and d-excess ($\Delta$d-excess $2days^{-1}$) over a 2-day period for a)
individual samples for events presented in Table 2 for 2017 (o), 2018, (x) and 2019 (□), b) the same values colour coded by initial d-excess
from each event. c) shows the relationship between change in d-excess after 2-days plotted against the initial d-excess value, with the linear
regression line in black.





daily changes (p < 0.01, t = 4.0070, df = 1715, Std. Err. = 0.125). Moreover, changes in d-excess during events are double the magnitude of background variability with a consistently negative sign for all years, supporting evidence that d-excess of

recently deposited snow has a 72 % chance of decreasing during surface snow metamorphism (SSA decay) during the first two days, according to our data.

Analysis shows that rapid SSA decay events correspond to decreases in d-excess over a 2-day period in 72 % of the samples. Results from EOF analysis during periods of low spatial variance in isotopic composition over the sampling transect reveals a coherence between the dominant mode of variance of SSA and d-excess, suggesting that processes driving change in SSA also

influence d-excess.

## 4   Discussion

Continuous daily SSA measurements at EastGRIP have enabled quantification of variations in snow physical properties due to precipitation and snow metamorphism during summer. Understanding the relationship between rapid decreases in SSA and corresponding change in isotopic composition require clearly defined events and environmental context. Using a multi-day SSA

decrease threshold, 21 events are defined from the summer field seasons of 2017, 2018 and 2019. All events are characterised by a peak and subsequent decay in SSA, the rate of which is proportional to the initial SSA value. SSA decay in precipitation free periods is driven by sublimation and vapour diffusion which is expected to influence the snow isotopic composition (Ebner et al., 2017; Hughes et al., 2021).

### 4.1   Decay model developments

In this study, we present an empirical SSA decay model for surface snow of polar ice sheets based on continuous daily SSA measurements. The model describes SSA decay under natural summer conditions on the ice sheet. The findings from this study agree with previous studies, that SSA decay is most accurately described by an exponential function (Cabanes et al., 2002), and indicates that the crystal structure of a new snow layer is a key driver of decay rate within the defined conditions over 2-5 day periods.

Comparison with weather station data showed that the SSA decay rate during events had no systematic influence from weather variables (wind speed, temperature and relative humidity). The only exception is for temperatures outside the set range for the model. Surface temperatures below -25 °C were characterised by a significantly higher background SSA (defined as the mean SSA value of the final day of decay events) (Fig. 3), indicating high background SSA due to reduced snow metamorphism in colder conditions. This observation is supported by theory and observation that sublimation and deposition

are thermally activated processes (Cabanes et al., 2003). Taillandier et al. (2007) (T07) developed an SSA decay model with a surface temperature parameter in addition to initial SSA which is able to capture the behaviour of decay during the cold event, E7, at EastGRIP suggesting temperature is important to consider when predicting SSA outside the defined temperature range. However, the influence of temperature on SSA decay rate within the defined temperature range is negligible. Model-





observation comparisons show equal performance for the SSA decay model from this study ($r^2$ = 0.89) compared to T07

temperature gradient metamorphism model ($r^2$ = 0.9).

The top 1 cm of the 2.5 cm SSA sample is measured by the Ice Cube device, and thus, is most likely to capture the precipitation signal (Gallet et al., 2009; Klein, 2014). Directly after precipitation, isothermal snow metamorphism is expected to be dominant due to to high surface curvature of fresh snow crystals (Colbeck, 1980). Alternative SSA decay models are proposed by Taillandier et al. (2007) to describe snow metamorphism under temperature gradient (temperature driven recrystallisation)

and isothermal (curvature driven recrystallisation) metamorphism, with the surface temperature and initial SSA being variable parameters. However, we find that all events are most accurately predicted using the temperature gradient decay equation, which accounts for the very low surface temperature observed in E7. The similarity in prediction for -25 °C to 0 °C suggests the EastGRIP SSA decays are not only driven by crystal curvature but by temperature gradient vapour diffusion as well.

The influence of snow metamorphism after precipitation during winter is expected to be reduced due to low temperatures

and negligible temperature gradients during polar night. Based on this, the model is only recommended to use for polar ice sheet summer conditions only. Within the defined conditions, the SSA decay model is a simple empirical model to describe SSA decay in the accumulation regions of the Greenland Ice Sheet, with dependence on the initial SSA alone.

## 4.2    Decay model applications

Conditions for the model are expected to be applicable over the Greenland Ice Sheet interior under mean summer conditions.

The model predicts decay events at EastGRIP with a $r^2$ of 0.89, compared to observation, within defined conditions. SSA estimates from satellites have previously been compared to ground observations and show a strong correlation between daily mean SSA and satellite retrieved SSA at EastGRIP (Kokhanovsky et al., 2019). The SSA decay model has the potential to predict SSA decay over the entire accumulation zone of the Greenland Ice Sheet using satellite data, the model can be evaluated for different sites to document the spatial variability in SSA over the entire ice sheet, and describe the summer SSA

decay. This has additional benefits for quantification of surface mass balance and surface energy budget due to the relationship between snow microstructure and surface albedo.

## 4.3    Rapid SSA decay and d-excess

In this study, processes driving snow metamorphism are documented to influence isotopic composition of the snow after precipitation, supporting experimental observations and theoretical understanding (Ebner et al., 2017; Wahl et al., 2021; Hughes

et al., 2021). Results from this study suggest that surface snow metamorphism following precipitation events corresponds to change in isotopic composition, most clearly observed in d-excess (Table 2). Based on our results, rapid decreases in SSA correspond to decreases in d-excess of a new snow layer in 72 % of cases during the first 2-days of rapid SSA decay.

Using the eddy-covariance latent heat flux measurements, we observed net sublimation during all decay events (with the exception of E21) used for isotopic analysis, which is in agreement with recent studies that document fractionation during

sublimation results in slight increases in $\delta^{18}$O and decreases in d-excess (Madsen et al., 2019; Hughes et al., 2021; Wahl et al., 2021). However, sublimation is not the only process occurring. Vapour pressure gradients due to surface curvature drive snow



metamorphism via vapour diffusion through the pore space and thus, kinetic fractionation is expected to influence the isotopic composition. A larger influence is expected for d-excess than $\delta^{18}O$ because kinetic fractionation influences $\delta D$ more than $\delta^{18}O$ ($d-excess = \delta D - 8 \cdot \delta^{18}O$) with a stronger influence on d-excess than $\delta^{18}O$, which can explain the covariance between

d-excess and SSA observed most clearly during 2019 (Cappa et al., 2003; Dadic et al., 2015). Our approach to the change over a 2-day period instead of daily change allows for increased propagation of the isotope signal during SSA decay to account for the 1 cm representation from Ice Cube SSA measurements, compared to the 2.5 cm bulk isotope measurements (Gallet et al., 2009; Klein, 2014). A significant relationship is observed between change in d-excess and change in SSA during the first 2-days compared to daily analysis (with an additional relationship observed during 2019 between daily change in d-excess and

daily change in SSA). Decreases in d-excess are observed during rapid SSA decay, driven by a combination of sublimation, deposition and vapour diffusion through the pore space.

Surface snow metamorphism is not confined to rapid SSA decreases, and thus isotopic composition change is observed continuously. However, results from this study indicate that d-excess changes during rapid SSA decay have significantly different distribution than the background non-event fluctuations. Our findings are in agreement with a study from Antarctica which

showed a significant relationship between d-excess and physical snow properties with depth, while negligible relationship was observed for $\delta^{18}O$ (Dadic et al., 2015). Our study has selected rapid SSA decays fitting to the decay model to address how changes in snow crystal morphology after precipitation relates to change in isotopic composition. Future studies would benefit from using isotope flux models to account for the influence of sublimation and deposition, to determine unexplained isotopic composition change.

An additional feature supporting the observation of processes driving surface snow metamorphism corresponds to a decrease in d-excess, is a clear relationship between substantial increases in SSA and increase in d-excess (Fig. 6). The upper 10th percentile of $\Delta$SSA increases ($14.7\,m^2\,kg^{-1}$) corresponds to positive $\Delta$d-excess in 70 % of cases (Fig. 5). Large increases in SSA are closely associated with precipitation, however, increases are observed in a number of other scenarios (Domine et al., 2009). Precipitation is expected to cause the largest SSA, suggesting that the d-excess of precipitation is most often higher than

existing surface snow. Our results therefore suggest that the precipitation isotopic composition signal is not always preserved after snow metamorphism due to (kinetic) fractionation during sublimation and other surface processes.

### 4.4 Influence of event conditions on isotopic change

Surface conditions prior to and during SSA decay events vary, with a number of events having no measured accumulation or observed snowfall (Fig. A). Removing events with non-homogeneous increases in surface height and events where additional

precipitation or significant snowdrift are observed, reveals that during rapid SSA decays following significant precipitation, there is increased likelihood of observing concurrent decrease in d-excess during the first day (Fig. 6). This observation combined with results presented in Fig. 5a strongly suggests that initial snow metamorphism after precipitation corresponds to a decrease in d-excess of in the surface snow.





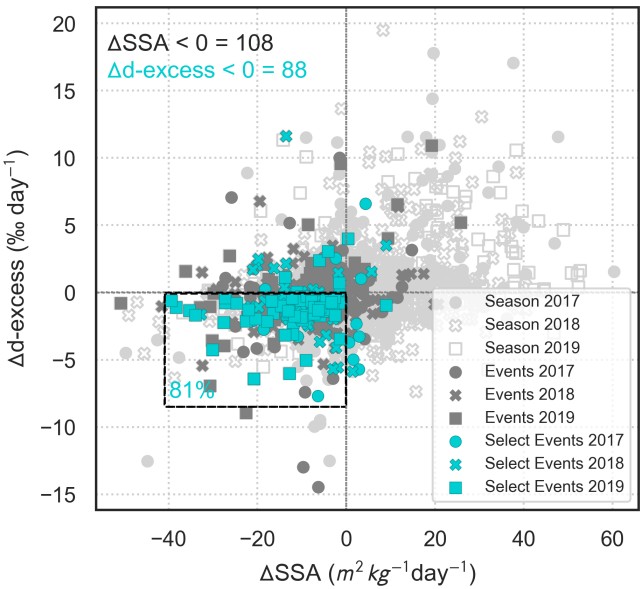

**Figure 6.** *Change in d-excess per day ($\Delta$d-excess $day^{-1}$) vs. change in SSA per day ($\Delta SSA day^{-1}$)*
The relationship between the rate of change in SSA per day ($\Delta SSA\,day^{-1}$) and d-excess ($\Delta$d-excess $day^{-1}$) for all summer seasons 2017-2019 (light grey), all events (dark grey) and selected events based on substantial accumulation (dark turquoise). The box indicates the values corresponding to daily decrease in d-excess during decrease in SSA, with 81 % of selected events in this quadrant.

## 4.5 Spatial variability of snow surface

Low accumulation rates at EastGRIP result in the potential for winter snow layers to influence the isotopic composition in the 2.5 cm surface snow. Accumulation heterogeneity causes uneven mixing of layers at each sample site, which is observed clearly in the large spatial variability in isotopic composition measurements in Fig. 4a and b. EOF analysis is used to account for spatial variability at each site, and a coherence is observed between the principal components of d-excess and SSA. PC1 is weaker when spatial variability is high, and during these periods the coherence between d-excess and SSA are muted. During the start

of 2017 and 2018 PC1 of d-excess is coherent with PC1 of $\delta^{18}O$, and decoupled from PC1 of SSA. At the start of the season, the 2.5 cm sample will contain winter snow layers which are less influenced by snow metamorphism (Libois et al., 2015; Town et al., 2008), and thus, a coherent signal between d-excess and $\delta^{18}O$ is observed. The transition to a coherence between PC1 of d-excess and PC1 of SSA can be explained by summer snow layers, influenced by snow metamorphism, causing d-excess to appear to become decoupled from $\delta^{18}O$, which is less influenced by kinetic fractionation than $\delta$D (Masson-Delmotte et al.,

2005) during snow metamorphism.



### 4.6 Implications to ice core interpretation

Documented changes in snow isotopic composition during surface snow metamorphism have potential implications for interpretation of stable water isotope records from ice cores, given that the current interpretation assumes the precipitation signal is preserved (Dansgaard, 1964). Seasonal transition from a coupling of PC1 of d-excess and PC1 of $\delta^{18}$O, to a coherence between PC1 of d-excess PC1 of SSA at the latter part of the season, suggest that summer snow metamorphism causes d-excess to appear to decouple from $\delta^{18}$O. Kinetic fractionation during sublimation is expected to be the cause a decrease in d-excess in the snow, given the different diffusivities of HDO and and $H_2{}^{18}$O (Masson-Delmotte et al., 2005).

Seasonal signals are influenced by millennial scale insolation variability (Masson-Delmotte et al., 2006; Laepple et al., 2011). An inverse relationship is observed between obliquity and d-excess over the past 250 ka years at Vostok which is attributed to the insolation gradient between high and low latitudes causing increases moisture transport from low latitudes relative to high latitudes (Vimeux et al., 2001, 1999). Results presented in our study document decreases in snow d-excess during surface snow metamorphism. Millennial scale local insolation variability has a strong influence on temperature gradients in the snow (Hutterli et al., 2009). Thus, it is possible that local insolation variability may also influence d-excess due to temperature gradients in the snow driving snow metamorphism at the surface.

Our results highlight the need to consider the influence of surface snow metamorphism on isotopic composition in stable water isotope records as the traditional interpretation of d-excess ice core signal does not account for any post-depositional signal. Future work to decouple the processes driving change in d-excess (sublimation from surface or interstitial vapour diffusion in the pore space) is vital for modelling the change in isotopic composition down to the close-off depth in the firn (Touzeau et al., 2018; Neumann and Waddington, 2004). In addition, it would be beneficial to obtain direct measurements of the isotopic composition and SSA of precipitation, to determine the fraction of precipitation in the SSA samples.

### 5 Conclusions

This study addresses the rapid SSA decay driven by surface snow metamorphism. In particular, the study aims to explore how rapid SSA decay relates to changes in isotopic composition of the surface snow in the dry accumulation zone of the Greenland Ice Sheet. Ten individual snow samples were collected on a daily basis at EastGRIP in the period between May and August of 2017, 2018 and 2019. Periods of snow metamorphism after precipitation events are defined using SSA measurements to extract periods of rapid decreases in SSA.

An exponential SSA decay model ($SSA(t) = (SSA_0 - 26.8)e^{-0.54t} + 26.8$) was constructed to describe surface snow metamorphism under mean summer conditions for polar snow, with surface temperatures between -25 °C and 0 °C and wind speeds below $6\,m\,s^{-1}$. The empirical model can be applied to remote areas of polar ice sheets and requires only initial SSA as the parameter, making it simple to use. The relationship between defined events of snow metamorphism and corresponding snow isotopic composition was then explored.

We observe changes in isotopic composition corresponding to post-depositional processes driving rapid SSA decay. Principal components from EOF analysis for SSA and d-excess indicate that under near-homogeneous surface snow conditions, d-excess





varies in phase with SSA throughout a large proportion of the sampling seasons. This suggests that post-depositional processes
and precipitation influence both physical snow structure and isotopic composition concurrently. Over the first 2-days of rapid
SSA decay events, d-excess is observed to decrease significantly from the initial value for most events, at the same time
we observe net sublimation. Significant changes in surface snow d-excess are observed during days following a precipitation
event, suggesting that precipitation d-excess signal is altered after deposition, together with changes in physical snow properties
(SSA).

In summary, our results suggest that the precipitation isotopic composition signal is not always preserved due to isotopic
fractionation during the processes driving surface snow metamorphism. Observations of post-depositional decrease in d-excess
during rapid SSA decay hints to local processes influencing the d-excess signal and therefore an interpretation as source region
signal is implausible.





## Appendix A

**Table A1.** *SSA Decay Event Conditions*

Duration and conditions for all 21 events defined by the threshold. 'Initial Conditions' refers to the conditions during the day ( 24h) before the event, while 'Event Conditions' describes the dominant conditions for the event duration, based on field observations. 'Surface Temperature' is the mean surface temperature during the event. 'Comments' highlight any significant weather behaviour during the event.

|      | Date          | Event No. | Surface Temperature | Initial Conditions | Event Conditions | Comments          |
| ---- | ------------- | --------- | ------------------- | ------------------ | ---------------- | ----------------- |
| 2017 | 27/05 - 01/06 | E1        | -17.3               | No clear driver    | Clear-sky        |                   |
|      | 19/06 - 24/06 | E2        | -13.6               | Snowfall           | Clear-sky        |                   |
|      | 30/06 - 02/07 | E3        | -14.0               | Snowfall           | Overcast         | Snow drift Day-0  |
|      | 10/07 - 15/07 | E4        | -13.2               | Snowfall           | Clear-sky        |                   |
|      | 18/07 - 19/07 | E5        | -11.7               | Snowfall           | Overcast         |                   |
|      | 21/07 - 23/07 | E6        | -11.2               | Snowfall           | Overcast         |                   |
| 2018 | 07/05 - 10/05 | E7        | -33.7               | Drift and fog      | Clear/ice-fog    | Snowfall Day-2    |
|      | 14/05 - 15/05 | E8        | -19.8               | Snowfall           | Clear-sky        |                   |
|      | 16/05 - 18/05 | E9        | -21.5               | Snowfall and fog   | Overcast         |                   |
|      | 09/06 - 11/06 | E10       | -14.9               | Ground fog         | Overcast         |                   |
|      | 27/06 - 29/06 | E11       | -15.3               | Ground fog         | Clear-sky        |                   |
|      | 30/06 - 03/07 | E12       | -11.2               | Wind drifted snow  | Clear-sky        |                   |
|      | 04/07 - 06/07 | E13       | -10.2               | Snowfall           | Clear-sky        |                   |
|      | 16/07 - 21/07 | E14       | -14.3               | No clear driver    | Clear-sky        | Dusting of snow   |
|      | 23/07 - 27/07 | E15       | -14.1               | Ground fog         | Clear-sky        |                   |
| 2019 | 17/06 - 20/06 | E16       | -11.4               | Snowfall           | Clear-sky        |                   |
|      | 27/06 - 30/06 | E17       | -9.5                | No clear driver    | Overcast         | Fog and snow      |
|      | 02/07 - 05/07 | E18       | -7.0                | Snowfall           | Overcast         |                   |
|      | 06/07 - 08/07 | E19       | -10.0               | No clear driver    | Clear-sky        |                   |
|      | 18/07 - 20/07 | E20       | -7.6                | Ground fog         | Overcast         |                   |
|      | 28/07 - 31/07 | E21       | -6.5                | No clear driver    | Clear-sky        |                   |



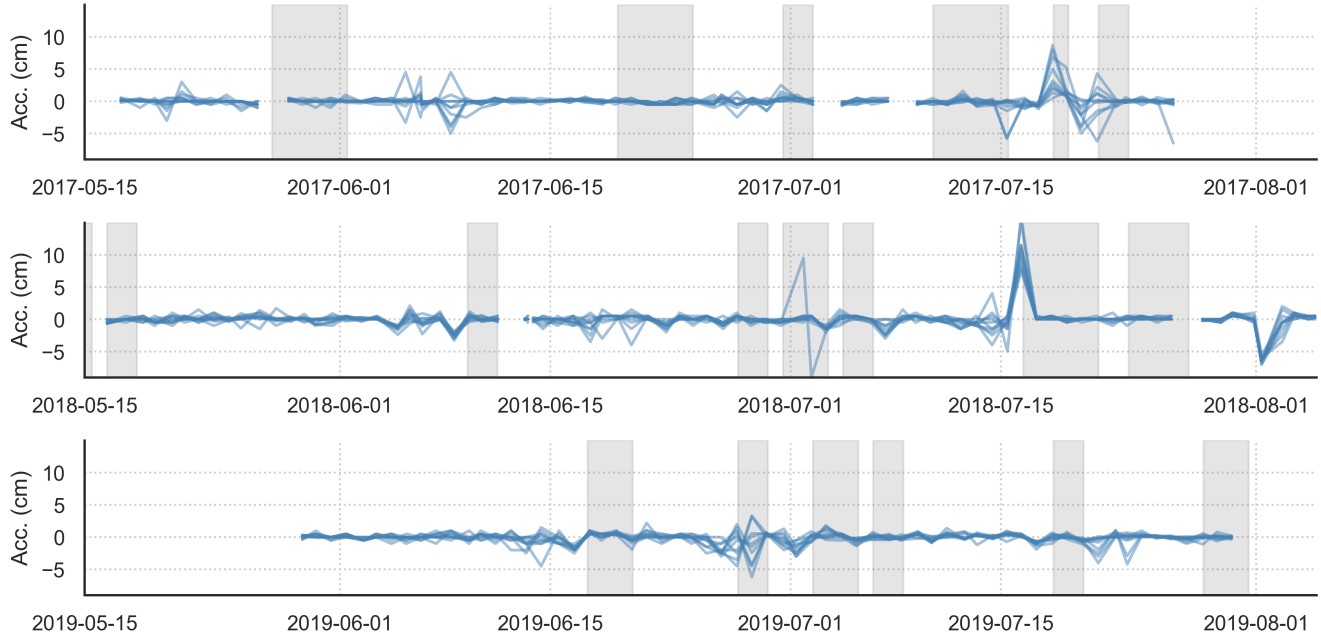

**Figure A1.** *Accumulation at each sample site*

Accumulation measurements from each sample site over the 90 m sampling transect is shown here for 2017, 2018 and 2019 respectively. Each line represents an individual site. Negative values indicate a decrease in surface height, and positive values suggest precipitation or deposition adding to the surface height. The grey bars show the individual events defined in Section 3.1

*Data availability.* The SSA, density and accumulation data for all sampling years is available on the PANGAEA database with the DOI:***. Snow isotope data is also available on the PANGAEA database with the DOI:***. Data from the Programme for Monitoring of the Greenland Ice Sheet (PROMICE) 400 were provided by the Geological Survey of Denmark and Greenland (GEUS) at http://www.promice.dk. Eddy Covaraniance Tower measurement are available on the PANGAEA database with the DOI: https://doi.org/10.1594/PANGAEA.928827.

*Author contributions.* HCSL, AKF and RHS designed the study together. AKF, SW, MH, MB, AZ, SK and HCSL carried out the data
collection and measurements. RHS, AKF and HCSL worked directly with the data. RHS, AKF and HCSL prepared the manuscript with contributions from all co-authors. AKF contributed largely to the manuscript text and structure. HCSL designed and administrated the SNOWISO project.

*Competing interests.* The authors declare that they have no conflict of interest.



*Acknowledgements.* This project has received funding from the European Research Council (ERC) under the European Union's Horizon
2020 research and innovation program: Starting Grant-SNOWISO (grant agreement 759526). EastGRIP is directed and organized by the
Centre for Ice and Climate at the Niels Bohr Institute, University of Copenhagen. It is supported by funding agencies and institutions
in Denmark (A. P. Møller Foundation, University of Copen-hagen), USA (US National Science Foundation, Office of Polar Programs),
Germany (Alfred Wegener Institute, Helmholtz Centre for Polar and Marine Research), Japan (National Institute of Polar Research and
Arctic Challenge for Sustainability), Norway (University of Bergen and Bergen Research Foun-dation), Switzerland (Swiss National Science
Foundation), France (French Polar Institute Paul-Emile Victor, Insti-tute for Geosciences and Environmental research), and China (Chinese
Acad-emy of Sciences and Beijing Normal University).





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
