# Peer review of "Exploring the role of snow metamorphism on the isotopic composition of the surface snow at EastGRIP"

_The Cryosphere, 2021_

## Referee Comment (RC2)

**Overview**

This paper investigates the relationship between changes in snow specific surface area (SSA) and its isotopic composition, focused on d-excess, at EastGRIP. The Authors focus on precipitation events, after which rapid SSA decays are observed, coupled to a decrease in d-excess. The Authors propose an exponential rate law for SSA decay, which is temperature independent between 0 and -25°C. The Authors then discuss the interplay between snow metamorphism and d-excess, and the possible impact of their findings on the interpretation of the ice core isotopic record.

**General comments**

The idea underlying this research is very nice: snow metamorphism results in sublimation-condensation cycles which should lead to isotopic fractionation. SSA decay is taken as a proxy for the intensity of metamorphism, and the expected correlation between SSA decay and isotopic fractionation is found, and is readily visible in d-excess. Such a study is clearly relevant to the interpretation of the ice core isotopic record and the data presented therefore deserves attention.

However, my opinion is that the experimental protocol is partly flawed, and this unfortunately casts doubt on the validity of the data obtained and on the conclusions derived. The first point is that SSA is measured on a 1 cm thick layer while isotopes are measured on a 2.5 cm thick layer. Furthermore, no detailed observations of surface snow are mentioned to ensure that the thicker 2.5 cm sample was the same snow layer as the top 1 cm snow. In many cases, the authors may then be measuring 2 little-related snow samples, which would in fact completely invalidate their study.

Many processes can affect the very surface snow layer. These include fog deposition, the formation of surface hoar or sublimation crystals, and wind drifting. All this is hardly mentioned, so that I am not even sure that adequate observations were systematically made. These are absolutely necessary for any careful snow physics investigation. If a 0.5 cm-thick fog deposit or surface hoar formation takes place, then clearly the SSA value will mostly reflect this deposit while the isotopic measurement will mostly characterize the underlying snow layer. Relating both measurements will then be totally meaningless. It is clear to me that the authors should have sampled only the top layer for isotopic measurements. If not enough material was present in their ICE CUBE sample holder, then they should simply collect more surface sample nearby.

Wind drifting is another important process, which is not detailed. The threshold of 6 m/s for the mean daily wind speed is simply not adequate. Hourly values must be considered, and in fact ideally maximum, not average values, are most useful to evaluate wind speed effect on drifting. But the best data on this aspect is observations. Wind drifting can easily be detected by observations. I appreciate that such observations cannot be done 24 hours a day, but the consequences of wind drifting are easily observable by looking at changes in the snow scene.

Drifting can remove newly precipitated snow or accumulate it is some places. This must be recorded when sampling. It is fairly easy to recognize snow layers from careful observations. All these mandatory observations do not appear to have been done.

I very strongly recommend that the authors detail whatever observations were done and clearly say what has not been done. In their analysis, they should only keep data for which they are certain that SSA and isotopic measurements were on the same layer. All data with surface hoar, fog or sublimation crystals should be eliminated. Drifting events resulting in non-homogeneous layers that were sampled must likewise be eliminated. If there are not sufficient observations to sort the data, then I fear the study may be invalid.

The organization of the paper must also be modified. Data appear in the discussion. All results should be reported in the results section and extra figures showing wind speed and snow surface conditions must be drafted.

Regarding the SSA decay rate law, I am not sure this is the best formula. Since sublimation is thermally activated, the absence of a temperature effect is strange. Perhaps when data is sorted, such an effect will appear. The authors quote (Cabanes et al., 2003) to support their choice of analytical expression, but those Authors had a temperature-dependent rate law. Furthermore, subsequent studies on SSA decay rate laws proposed other analytical expressions, and their exploration should be discussed when the rate law is investigated, not line 309 in the discussion.

In summary, this potentially interesting study may be partially of totally invalidated by an inadequate experimental protocol, at least based on the information supplied in the paper. If the authors have made observations not reported in this version, they should report all relevant information in a revised version. I then recommend sorting the data and removing all data where there is a reasonable suspicion that SSA and isotopic measurements were not on the same snow layer. I also strongly recommend a more logical organization of the paper. The discussion is often unfounded speculation and must be considerably shortened. I propose below numerous specific comments that I hope will be useful to the Authors in preparing an extensively revised version, for which I recommend a second round of review. These comments were written before the general evaluation, so there is some repetition. And finally, I kindly request that *all* Authors involved in this work make a careful reading of the revised version. This does not seem to have been done for the version I read, which is not very respectful for the reviewers.

**Specific comments**

Line 35. Spell out SSA=specific surface area, which is the surface are of the ice-air interface per unit mass of snow, expressed in $m^2 \, kg^{-1}$. It is not assumed to be linked to the optical grain size $d_{opt}$, as mentioned by the Authors, it is rigorously and simply linked by a geometric relationship

SSA=6/$\rho_{ice}$ $d_{opt}$, as shown in equation (1) of (Gallet et al., 2009), which is probably a more relevant reference than Linow 2012. In fact this relationship was already implicitly mentioned by (Grenfell and Warren, 1999), although they did not use the term specific surface area.

Lines 41-43. The reasons for SSA decrease (of dry snow) are not explained well and even erroneously. Wind fragmentation in fact increases SSA since smaller crystals are formed (Domine et al., 2009). Sublimation does not necessarily lead to SSA decrease as it reduces crystal size; and likewise vapor diffusion does not necessarily lead to SSA decrease. What actually leads to SSA decrease is the disappearance of small structures, often by sublimation, and the growth of larger crystals, often but not only by vapor diffusion in the pore space.

Line 47. It is erroneous to state that "While current versions of the so-called decay models exist, these are mostly based on lab-experiments and non-polar snow observations". The works of Cabanes and Taillandier are mostly based on Arctic and subarctic observations. Granted, none of these studies used data obtained on ice sheets, and this could be mentioned, if there are reasons to believe that ice sheet processes involved in SSA decrease are in general different from those on seasonal Arctic snowpacks. By the way, (Carmagnola et al., 2014) tested various SSA decay models against data from Summit, Greenland, and this may me relevant to the authors' topic.

Line 78. What is meant by surface temperature? Is this the skin temperature measured by IR emission? Or is it the air temperature near the surface? Mentioning a reference is not sufficient. A paper must be self -standing and must not require looking up references for understanding, especially for such a central variable. If this is skin temperature, all relevant details must be given here, including the instrument used, the wavelength range and the emissivity value used. Furthermore, validation of the skin temperature measurements would be desirable. IR sensors require very careful calibration to be accurate.

Line 85 ff. Sampling procedure. It is essential to note when there is a change in the snow layer sampled, i.e. when there was wind drift or precipitation. I guess precipitation events were readily identified, but what about wind drift? Did the authors note when the layer being sampled changed because of wind erosion of wind accumulation? This is critical for data interpretation.

Line 100. "Light penetration depth in snow of 200 kg m$^{-3}$ is approximately 1 cm". Light penetration does not just depend on density, but also on SSA. Thus for 200 kg m$^{-3}$, a penetration depth of 1 cm corresponds to a precise SSA value. Furthermore, penetration depth is not very meaningful. Do the authors mean e-folding depth? Note that if the e-folding depth is 1 cm, still 27% of the reflected light intensity will be due to depths >1 cm.  Also did the authors make detailed observations of detailed surface processes such as surface hoar, sublimation crystals or rime events (these are frequent at Summit, perhaps also at EastGrip)? This is important because these thin surface deposits will greatly impact measured SSA, while they will be diluted in isotopic measurements. To evaluate penetration depth and the impact of surface deposits on

SSA measurements, the Authors can use the TARTES model. https://snow.univ-grenoble-alpes.fr/snowtartes/ . This will allow them to make valid quantitative statements, and to explore the impact of surface deposits on measured SSA.

Line 117-118. It is strange the Authors did not sample the top 1 cm for isotopic measurements, to ensure better correspondence with the SSA measurements.

Table 1. Usually Table captions are concise and explanation are in footnotes. Most of the caption is in fact unnecessary and can be deleted.

Lines 132-133. Eq. (1) was indeed proposed by Cabanes et al. as the most empirically accurate, but this was just to fit their limited data set. Legagneux (2005) proposed a theoretically correct equation (his Req. 2). That equation was also used by (Flanner and Zender, 2006). Taillandier et al. (2007) used an approximation of that equation to fit experimental data and their equation has a log form. I believe the expression of Taillandier is more suitable. From the discussion, the Authors tested it, but this should be detailed here, not in the discussion.

Lines 162-164. Ground temperatures are not very relevant to the explanation of crystal shapes, as these form in clouds at a different temperature. And by the way Domine et al. (2008) is not the most suitable reference for this. I recommend (Kuroda and Lacmann, 1982) and references therein.

Line 165.The upper threshold for wind speed used here is a daily mean value of 6 m s$^{-1}$. When the daily mean value is 6 m s$^{-1}$, It is very likely that gust speeds were much higher and that wind drifting took place, with major modifications in SSA. Perhaps transport even brought other layers. I think combining events with and without snow drift is not adequate to derive SSA decay rate laws. At the minimum, events with and without drifting should be treated separately to investigate wind effects. Regarding isotopes, the sampling of blowing snow would have been interesting. Was that performed?

Line 178. What is the RMSE? This is mentioned line 194 but would be better mentioned here in context.

Line 190. The authors indicate intermittent snowfall during day 2 of E14. Why did they not remove this presumably thin new layer to avoid this artefact? The thin layer greatly affected the SSA measurement but probably had little impact on the 2.5 cm-thick isotope sample.

Line 197. Why is not an equation proposed and tested for the lower temperatures?

Lines 204-205. No influence of basic environmental variables. How about cloudiness? A very important variable for SSA decay is the temperature gradient in the snowpack. Near the surface, this is going to be greatly affected by cloudiness. In the absence of clouds, there will be a much stronger temperature gradient near the surface than under cloudy conditions. This probably deserves a bit of exploration. Various proxies for cloudiness can be tested, in particular the longwave budget.

Line 208. Are the units correct here?

Lines 242-243. Shaded regions in Fig 4 are said to indicate largely homogeneous snow cover. But The caption to Figure 4 says "Grey shaded regions indicate periods of high spatial variability in isotopic composition." I am confused.

Lines 241-249. This discusses the correlation between SSA and d-excess. The coherence is better when the snow layer is homogeneous. Could that just be due to wind effects? When the wind speed is low and there is no wind drifting, the snow remains unperturbed and *a priori* homogeneous. On the contrary, under greater wind speeds, drifting takes place, heterogeneity is generated and SSA and d-excess become decorrelated. Furthermore, since SSA measurements probe about the top 1 cm while isotopic measurements probe the top 2.5 cm, it is clear that when wind drifting takes place, both measurements may measure highly different layers, explaining the decorrelation. How about limiting data analysis to those events without wind speed?

Lines 256-257. Here the authors mention fog and negative LHF, i.e. likely surface hoar formation. Thus the authors may have observed snow conditions. All these observations must be mentioned when results are first presented. Data analysis must consider which processes were involved for each event. By the way, the standard abbreviation for latent heat fluxes is LE, not LHF.

Line 268. The authors invoked re-exposed old snow to explain some d-excess values. Careful observations during sampling can answer this question. If there was 1 cm of recent snow over old snow, the SSA measurement will have measured recent snow while isotopic measurements will have measured predominantly old snow. This will affect the quality of the SSA-d-excess correlation analysis. Again, inadequate samples must be removed from the analysis.

Line 287-288. Changes in snow physical properties observed are probably not due to precipitation and metamorphism *sensu stricto* (i.e. involving only water vapor transport within the snow layer). Processes involved also include wind drift, fog deposition, surface hoar deposition, and also possibly sublimation crystal formation. This last process is due to vapor transport within the snow, but since the growth of completely new crystals is involved, I suspect their isotopic composition would be very different from that of the snow layer they originate from. Sublimation crystals are in fact very frequent on cold snow under intense sunlight, even though reports are few (Weller, 1969;Gallet et al., 2014).

Line 291-292. For older snow also, sublimation and vapor diffusion are not the only processes involved. In particular, wind drifting is probably important.

Line 297. The correct reference is Cabanes 2003, not 2002.

Line 309-310. The comparison with the equation of Taillandier should be indicated in results. In fact, the choice of Cabanes' equation should be justified earlier on. Its interest as well. By the way, (Cabanes et al., 2003) used a temperature-dependent exponential coefficient.

Lines 311-318. This paragraph is not physically very sound and is not based by any quantitative analysis. Since the temperature gradient near the snow surface is not evaluated, there is no basis to say that isothermal metamorphism is dominant after precipitation. Then, since the Authors do not find any significant effect of temperature, they assume their observations are explained by the temperature gradient, implicitly implying that the temperature gradient show little variations between events. This paragraph should just be removed. All the statements are unsubstantiated. Furthermore, what is important in TG metamorphism is not the magnitude of the temperature gradient, but the magnitude of the water vapor flux, which is temperature-dependent. Lastly, it can be affected by wind speed through wind pumping and also by convection (Trabant and Benson, 1972;Benson and Trabant, 1973;Johnson et al., 1987;Sturm and Johnson, 1991). All these aspects would need to be discussed and quantified to engage in the discussion proposed in this paragraph.

Lines 319-322. Here again, the authors make unfounded statements. How do they know the temperature gradient is negligible during polar night? Under clear sky conditions, radiative cooling will on the contrary induce strong temperature gradients near the surface of the snow. The authors may just conclude that since their model is empirical it only applies under the conditions where data were obtained. In fact, it may not even be valid at this site in summer during other years.

Lines 324-331. Could not the authors compare their model to data obtained using the algorithms developed in (Kokhanovsky et al., 2019)? It seems possible to determine precipitation events using Sentinel data, as indicated by high-SSA periods, and then investigate the decay to test whether the model developed here indeed applied to the accumulation zone of the GIS. This paragraph lacks convincing arguments and sound a bit like just wishful thinking, while tests are possible.

Lines 336-337. Why would this correlation between SSA and d-excess be observed in only 72% of cases? I think it would be interesting to explore which events actually monitored a constant layer, rather than a layer perturbed by wind drift, the formation of surface hoar or sublimation crystals, or fog deposition.

Lines 339-351. This discussion of snow metamorphism could be significantly improved. I am not sure surface curvature effects played a detectable role. In any case, the authors need to substantiate this with quantitative calculations, they cannot just make such statements without a demonstration. I would think water vapor fluxes caused by temperature gradients and wind pumping, and perhaps thermal convection, can explain most observations.

Lines 360-366. This paragraph discusses the relationship between SSA increases and concomitant d-excess increases. However, this seems very misleading to me. This paper is

focused on SSA decrease of a given snow layer over time. Here, the approach is different. The authors consider changes in the SSA of surface snow, regardless of whether these changes involve the same layer. In fact, their SSA increases seems to always involve a change in layer, e.g. due to precipitation. Therefore, plotting data obtained by the evolution of a given identified layer together with data involving a change of layer seems meaningless to me.  What I understand from this paragraph is that new layers with high SSA have a higher d-excess value than older (and different) layers with low SSA. This may be interesting, but is different from the main topic of this paper, and should therefore not presented as the same topic.

Lines 368-373. It is surprising to see data presented in the discussion. This should be in the results section. So in fact there seems to have been observations of snow surface conditions and changes. Wind drifting, a key process for data interpretation, may have been observed after all. We need to see those data. Fig. A1 needs to also show mean hourly wind speed, and ideally maximum hourly wind speed if available, as well as observations of drifting. In fact, all surface snow observations, including fog deposition, the formation of surface hoar or sublimation crystals, and any other relevant information, must be shown in a Figure.

Lines 393-399. The speculation between insolation, temperature gradient and d-excess may be potentially interesting, but lacks a clear basis. Since the authors did not measure T gradients and did not adequately discuss their role on d-excess, I think this paragraph is not very useful. Please substantiate or remove.

The section on ice core implications could perhaps be strengthened a bit by treating specific examples. For examples, how is the d-excess signal affected by more frequent precipitation that metamorphize without wind perturbation, in comparison to precipitation events that rapidly form a wind slab with time-stable SSA? How does that relate to climate scenarios (e.g. glacial vs. interglacial). This is just a suggestion. I am sure the Authors can present other interesting cases. This is where I expected more in-depth discussions.

**References**

Benson, C. S., and Trabant, D.: Field Measurements on the flux of water vapour through dry snow, The Role of Snow and Ice in Hydrology, , 1973Banff, 1973, 291-298, 1973.
Cabanes, A., Legagneux, L., and Domine, F.: Rate of evolution of the specific surface area of surface snow layers, Environ. Sci. Technol., 37, 661-666, 10.1021/es025880r, 2003.
Carmagnola, C. M., Morin, S., Lafaysse, M., Domine, F., Lesaffre, B., Lejeune, Y., Picard, G., and Arnaud, L.: Implementation and evaluation of prognostic representations of the optical diameter of snow in the SURFEX/ISBA-Crocus detailed snowpack model, The Cryosphere, 8, 417-437, 10.5194/tc-8-417-2014, 2014.
Domine, F., Taillandier, A.-S., Cabanes, A., Douglas, T. A., and Sturm, M.: Three examples where the specific surface area of snow increased over time, The Cryosphere, 3, 31-39, 10.5194/tc-3-31-2009, 2009.

Flanner, M. G., and Zender, C. S.: Linking snowpack microphysics and albedo evolution, J. Geophys. Res., 111, D12208, 10.1029/2005jd006834, 2006.

Gallet, J.-C., Domine, F., Zender, C. S., and Picard, G.: Measurement of the specific surface area of snow using infrared reflectance in an integrating sphere at 1310 and 1550 nm, The Cryosphere, 3, 167-182, https://doi.org/10.5194/tc-3-167-2009, 2009.

Gallet, J.-C., Domine, F., Savarino, J., Dumont, M., and Brun, E.: The growth of sublimation crystals and surface hoar on the Antarctic plateau, The Cryosphere, 8, 1205-1215, 10.5194/tc-8-1205-2014, 2014.

Grenfell, T. C., and Warren, S. G.: Representation of a nonspherical ice particle by a collection of independent spheres for scattering and absorption of radiation, J. Geophys. Res., 104, 31697-31709, 10.1029/1999JD900496, 1999.

Johnson, J. B., Sturm, M., Perovich, D. K., and Benson, C.: Field observations of thermal convection in a subarctic snow cover, in: Avalanche Formation, Movement and Effects (Proceedings of a Symposium held at Davos, September 1986) IAHS pub. 162, edited by: Salm, B., and Gubler, H., IAHS, 105-118, 1987.

Kokhanovsky, A., Lamare, M., Danne, O., Brockmann, C., Dumont, M., Picard, G., Arnaud, L., Favier, V., Jourdain, B., Le Meur, E., Di Mauro, B., Aoki, T., Niwano, M., Rozanov, V., Korkin, S., Kipfstuhl, S., Freitag, J., Hoerhold, M., Zuhr, A., Vladimirova, D., Faber, A.-K., Steen-Larsen, H. C., Wahl, S., Andersen, J. K., Vandecrux, B., van As, D., Mankoff, K. D., Kern, M., Zege, E., and Box, J. E.: Retrieval of Snow Properties from the Sentinel-3 Ocean and Land Colour Instrument, Remote Sensing, 11, 2280, 10.3390/rs11192280, 2019.

Kuroda, T., and Lacmann, R.: Growth-kinetics of ice from the vapor-phase and its growth forms, J. Cryst. Growth, 56, 189-205, 10.1016/0022-0248(82)90028-8, 1982.

Sturm, M., and Johnson, J. B.: Natural-convection in the sub-arctic snow cover, Journal of Geophysical Research-Solid Earth and Planets, 96, 11657-11671, 10.1029/91JB00895, 1991.

Trabant, D., and Benson, C. S.: Field experiments on the development of depth hoar, Geol. Soc. Am. Mem., 135, 309-322, https://doi.org/10.1130/MEM135-p309, 1972.

Weller, G.: The heat and mass balance of snow dunes on the central Antarctic plateau, J. Glaciol., 8, 277-284, 1969.

---

## Author Comment (AC1)

**Response to Review 1**:

We thank the reviewer for their time and effort to evaluate and develop our study. We acknowledge the constructive criticism and comments from the reviewer and propose the following revisions. We appreciate the comments by the reviewer which have resulted in a significantly strengthened manuscript.

The original comments from the reviewer are in black and in blue are the author's responses, with blue italics to show the in-text changes. We want to point out that due to many useful comments and suggestions, the major revisions have been implemented resulting in significant changes to the manuscript, as can be seen in the document attached below.

**Introduction**

In my opinion, the introduction needs a more stringent train of thought to lead readers into the topic more smoothly. At the start, a broader introduction to the importance of proper ice core proxy use, and especially the relevance of this study in this wider context, would help to gain the readers' attention for this work. In this regard, L28-29, L46-50, L58-61, L67-69 are already interesting hooks, on which you could expand, so that the importance of your work is explicitly stated. I would further recommend a broader climate description of the study site, because this is something the authors rely on later during the interpretation of results (e.g. L375).

We take onboard this comment by adding an additional explanation of the interpretation of d-excess in ice cores (linked to a later comment starting L22). To expand the context and the importance of this study we propose to add the following text, which creates a stronger coherence between the introduction and the discussion on isotopic fractionation.

Introduction: *"... Decreasing SSA is predominantly the result of Ostwald Ripening, where large grains grow at the cost of smaller grains (Lifshitz and Slyozov,1961; Wagner 1961; Legegneux et al., 2004), and vapour diffusion driven by sublimation from convex surfaces, and deposition onto low energy regions (Pinzer et al., 2012; Flin and Brzoska, 2008; Sokratov and Golubev, 2009). The latter is dependent on temperature (Cabanes et al., 2002), temperature gradients between the air (Ebner et al.,2017), surface and subsurface, and wind conditions (Neumann et al., 2004; Town et at., 2008). Under natural conditions SSA decrease is driven by a combination of these processes (Pinzer and Schneebeli, 2009), each potentially modifying the isotopic composition of the snow (Ebner et al., 2017)."*

Regarding the study site, we have added some additional information about the accumulation rate and synoptic conditions at EastGRIP, while an extensive description of the meteorological conditions over the three sampling seasons has been added to the results. An overview of the results section restructuring can be found in the responses in the results section of the reviewer's comments.

*"The accumulation rate is approximately 14 cm w.eq. yr−1 (Schaller et al., 2017).*

*Westerly winds prevail during 2017 and 2018 with a wind direction of 227◦N, while 2019 had a prevailing south westerly wind (239◦N), corresponding to opposing phases of the North Atlantic Oscillation (NAO).*

*Significant weather conditions such as ground fog, drifting snow and snowfall, were documented each day."*

L5: The phrase 'after precipitation/deposition events' used here gives me the opportunity to point out the unclear use of either term in this manuscript. Given that you refer to surface snow, which stayed at the surface for an unknown period (L38-39), I would prefer the term 'deposition' event defined more clearly somewhere in the introduction/method section and used consistently throughout the manuscript, replacing 'precipitation'.

The term 'deposition event' is now used throughout the manuscript unless specifically referring to a precipitation. We propose to add the text below to define deposition events. We also take this opportunity to add an explanation of the influence of surface hoar and or sublimation crystal formation on the surface snow, based on SSA measurements of these deposition features from previous studies.

*"The term deposition events is used to describe rapid increases in SSA, expected to be from precipitation or drifted snow. It does not explicitly include surface hoar and sublimation crystal-like grain growth at the surface, given that previous studies indicate these depositional features have an SSA value around 54 $m^2$ $kg^{-1}$ using SSA of hoar frost (Dominé et al., 2009)."*

To account for the possibility of deposition via surface hoar we use field observations, latent heat flux measurements and temperature gradient data when analysing the isotopic change.

L22: Since the interpretation of deuterium excess as a proxy of moisture source conditions is a key background for this study, I suggest expanding on this point of the introduction. What kind of conditions were thought to be reflected by d-excess

We have expanded on the interpretation of d-excess in ice cores and what environmental conditions control d-excess.

*"The second-order parameter deuterium excess (d-excess) is defined by the deviation from the near-linear relationship between δ18O and δD due to non-equilibrium (kinetic) fractionation (d-excess = δD - 8 · δ18O), and is understood to reflect moisture source conditions (Dansgaard, 1964; Merlivat and Jouzel, 1979; Johnsen et al., 1989), snow crystal formation in clouds (Ciais and Jouzel, 1994; Sodemann et al., 2008), and changes in moisture source (Masson-Delmotte et al., 2005). …*

*Post-depositional processes at the surface involve additional kinetic effects adding complexity to the interpretation of d-excess (Casado et al., 2016; Hughes et al., 2021; Casado et al., 2021)."*

This is followed by recent studies evidencing kinetic fractionation during sublimation, as well as the supersaturated conditions leading to hoar frost which would cause large increases in d-excess (Stenni et al., Feher et al., 2021, Hughes et al., 2021; Casado et al., 2021). The revised discussion more stringently links the results from our study to previous work.

L34: As far as I can see, this is the first time, you use SSA as an abbreviation, so that an explanation of the full term and a slightly more detailed definition would be appropriate here.

We apologise for this mistake; this sentence has now been modified to the following:

"*Snow metamorphism works to reduce the snow-air interface, which can be quantified using the parameter snow specific surface area (SSA) (Legagneux et al., 2005). SSA of a snow sample is dependent on optical grain radius and density of ice (SSA=6/rhoice\*dopt) (Gallet et al., 2009)*".

L43: You use the term snow 'crystals' here. I suggest replacing it with snow 'grains', here and throughout the manuscript, because you are not specifically talking about the crystallographic term but rather the ice matrix, which is mostly composed of multi- crystal ice grains.

We agree, now 'snow crystals' has been changed to 'snow grains' throughout the text.

**Methods**

In section 2.5.1., the definition of a decay event is not entirely clear to me.

We add the following text to improve clarity:

"*To systematically identify rapid decreases in SSA, which we use as a proxy for events of snow metamorphism after deposition (identified based on the high mean SSA values), a threshold is set using the bottom 10$^{th}$ percentile of SSA decreases over a two-day period. This was found to result in the most equal number of events from each sampling year compared to 1- and 3-day changes. SA decay events are defined as by the initial peak, identified by the threshold, through to the next increase in SSA (rather than decrease).*"

L94-96: When did you take the samples exactly? How many were afternoon samples, and does this affect the results discussed here?

The samples were all done in the daytime, and primarily done in the morning. In the submitted paper we resampled the meteorological data to the SSA sampling time-periods to ensure consistent comparison. Moreover, the decay model and model intercomparison are now based on the exact sampling time, given that the existing models are hourly resolution.

It is slightly more complicated to include the exact sampling time when assessing the relationship between change in SSA and the absolute SSA values (Fig. 1), given that we do not want to interpolate SSA as we understand that the change will not be linear throughout the day. Instead, we propose to keep the "daily" change, and clearly state the different sampling times as limitation to the model. The corresponding description has been changed to:

"*The samples were all taken in the daytime, primarily in the morning. The meteorological data is re-sampled to the SSA sampling time-periods to ensure consistent comparison.* "

L97-98: I suggest amending the title of this section, because you are not strictly talking about the calibration of the Ice Cube device but the SSA measurements using the Ice Cube.

The section title has been changed to "*SSA measurement protocol*".

L101: is 294 kg m-3 the density averaged over all three seasons, or do the seasons differ

significantly in value?

This is an average over all seasons. Annual means and standard deviations have also been added "(2017 = 307±40 kg m$^{-3}$, 2018 =278±47 kg m$^{-3}$, 2019 = 294±50 kg m$^{-3}$)".

L108: Did you use the identical sample for SSA and stable water isotopic measurements or neighbouring material? Depending on this, the sentence in L110 needs amending: 'sealed in a polyethylene bag' or are several bags used for one sample?

The SSA samples were subsequently measured for water isotopes composition, we apologise that this was unclear. To clarify this sentence has been added to the text:

*"Individual SSA samples were put in separate bags and subsequently sampled for water isotopic composition. Thus, every day the 10 SSA samples have a corresponding isotopic composition. The resultant isotope value is the average composition over the top 2.5 cm of snow."*

L135: Why did you choose the threshold 6 m s-1? If I am not mistaken, blowing snow is already an issue at 5 m s-1, so that this could be a better threshold? It could also be helpful to know how many data points are in the upper spectrum of wind speeds still considered.

We agree with hindsight that this threshold is insufficient to reduce the likelihood of surface perturbation, and to address this we now use the 10-minute data from PROMICE. It is important to note here that 209 out of the total 237 sampling days have daily maximum wind speed exceeding 5 m s$^{-1}$ and no events had wind-speed consistently below 5 m s$^{-1}$ (two had 5.1 m s$^{-1}$). In addition, snowdrift events were documented in the EastGRIP field diary and correspond to wind-speeds above 7 m s$^{-1}$. Several events have maximum wind-speed between 6- 7 m s$^{-1}$, and no snowdrift documented. Based on this analysis and observations from the literature, we define two wind categories, as briefly suggested by the reviewer in a later comment, we have added a secondary wind-speed category for comparison of SSA decay when wind-speed is <6 m s$^{-1}$ (low-wind events), and when maximum wind-speed is between 6- 7 m s$^{-1}$ (moderate-wind events). The following text is added to the document:

*"A set of criteria are required to reduce the potential of analysing events with wind-perturbed surfaces, resulting in the removal of surface snow. In Antarctica, unconsolidated surface snow has been observed to drift at wind speeds as low as 5 m s−1 measured at 2 m height (Birnbaum et al., 2010). However, a study from Greenland documented snowdrift starting at 6 m s−1 (Christiansen, 2001), likely due to warmer temperatures allowing for the surface snow to become more bonded (Li and Pomeroy, 1997). At EastGRIP, calm conditions correspond to wind speeds from 0 –5.2 m s−1 according to field diary observations. The mean daily maximum wind speed for the three sampling seasons was 6.8 m s−1, while blowing snow was documented only when wind speeds exceeded 7 m s−1. Based on this assessment, we define two wind-speed categories for comparison of the effects of wind-speed on SSA decrease. The first includes events with wind-speed consistently below 5.2 m s−1, hereafter referred to as low-wind events, to ensure no surface perturbation. Secondly, we consider events where the maximum wind-speed is between 6 –7 m s−1, hereafter referred to as the moderate-wind events. The inclusion moderate-wind events allow an assessment of the influence of wind-speed on SSA decrease."*

Two SSA decay events are below the 6ms-1 threshold, both of which are from 2018. The remaining SSA decay events, E10 and E11, have maximum values of 5.1 m s$^{-1}$, and 5.07 m s$^{-1}$, respectively and last for 3-days each.

Out of the 21 initially defined events, only 2 are below the wind-speed threshold with maximum values of 5.1 m s$^{-1}$ in both events. We expect negligible snowdrift for these two events allowing us to confidently argue that the surface is unperturbed and isotopic change is the result of snow metamorphism. The likelihood of drifting snow during moderate-wind events is considered using the equation defined from Li and Pomeroy (1998), where the threshold wind-speed for snowdrift is defined as a function of temperature.

Following the same structure as in the original manuscript, we construct the SSA decay model with parameter values set for the two wind-regimes. We add the revised figure to this response. Intuitively, the SSA decay rate is higher for moderate-wind events (-0.53 m$^2$ kg$^{-1}$ day$^{-1}$) compared to low-wind events (-0.41 m$^2$ kg$^{-1}$ day$^{-1}$). As the reviewer will see later in this response, we add the results from the comparison of our data and SSA decay model to existing models from Flanner and Zender (2006) and Taillandier et al. (2007).

The wind-speed distributions for the daily maximum and 10-minute mean values are also added to the supplement. For isotopic analysis we now focus on the low-wind events alone and describe the latent heat flux and temperature gradients during the two events.

L162-164: I think that you are making an important point here. Could you clarify this sentence so that it becomes obvious why you chose -25°C and not -22°C as the boundary of your SSA decay model, given that this is where snow crystal shape changes? And maybe this is better placed in the methods section.

We agree that the temperature boundaries are clearly an interesting feature of the work. Nonetheless, due to the updated wind-speed criteria, we have ultimately removed this temperature boundary condition given that the low-temperature event coincides with high-winds.

Figure 1: Firstly, the layout of this figure does a good job at visualising the sampling procedure. Unfortunately, I cannot read the site labels and the legend in panel (a). Personally, I can recommend the open-source software QGIS for designing maps. Table 1: How is the data coverage of the AWS for the seasons 2017-2019? How many data points are missing? I think that '.2' should be amended to '0.2'. The term EC needs explanation here, as it is the first time, this is mentioned.

We thank the reviewer for the suggestion to use QGIS. We have simply replaced the map as the legend in the previous figure was not relevant to the study, but QGIS will definitely be beneficial for future work. '.2' has been changed to '0.2' and added information about the eddy-covariance tower and the measurement instruments from PROMICE to the table.

**Results**

When it comes to the presentation of results, certain parts of the description appear repetitive (e.g. L270-273), while major conclusions are only mentioned once and are not stated clearly

enough (e.g. L283-285). Your work is really interesting, so that I would like to see (1) a concise description of all records of relevance, (2) a step-by-step line of interpretation, in which your outcomes become more visible. At the moment, the measured results and their interpretation are often mixed and the sub-division into chapters not very clear. Moreover, the chosen language is sometimes vague, leaving out important details which allow the reader to know exactly which parameter you are talking of (e.g. L240-249). I would recommend making the descriptions as precise and specific as possible, e.g. in L87 'the specific sampling dates' would be better.

Based on the reviewers' comments, we have restructured the results to improve clarity and strengthen our arguments. The new format is broadly as follows and we have included the updated figures.

1) Description of all used datasets (time-series and basic statistics between years) has been added in the first results section. Here we look mostly at the inter-annual variability and highlight the significantly lower accumulation for 2017.

2) EOF analysis follows the meteorological description to show the relationship between the dominant modes of variance of SSA, d18O and d-excess. The three parameters, SSA, d18O and d-excess are described before presenting the EOF analysis. By moving this section, the motivation for exploring the relationship between isotopic composition and SSA later in the paper becomes clearer. Here we also add the revised accumulation data for each sampling season.

3a) Description of SSA decay events extracted by the threshold, with clear explanation that events with snowfall/fog/snow drift (from field-diary) and high wind speeds are removed from further analysis. The identification of similar decay shapes in SSA time series are then noted.

3b) The empirical model to describe the SSA decrease behaviour during periods of rapid SSA decay is described for the previously described wind-speed categories. A linear regression between dSSA and SSA for the two wind-speed categories is noted to describe the decay rate and decay constant in the decay model. Based on the reviewer's comment, we first show the mean decays for each event, and in the second panel the modelled decay. In addition, we compare the model from this study to previous models from the literature. The model from Flanner and Zender (2006) is based on theoretical grain growth, and thus we can compare the observed behaviour - and our empirical model - to their physical-based model.

4a) As previously mentioned the isotopes are measured from each SSA sample, giving us a direct comparison for analysis of SSA decay events. In the same format as the original manuscript, the isotopic changes over a single day and 2-days are documented for both the low- and moderate-wind events.

4b) We then look in more detail at the two low-wind events given that we can be more certain of minimal surface perturbation. Latent heat flux and temperature gradients during these events are presented to explain the direction of fluxes.

(If repeated, precipitation isotopic composition would be measured to determine the proportion of precipitation isotope signal in a snow sample, and thus observe the change in

isotope signal from the exact precipitation signal).

L223-224: Can you provide an estimate of the probability that the top 2.5 cm sample contains material from several precipitation events? This is one part, where a more detailed climate description upfront including accumulation event frequency would be helpful.

The reviewer is indeed here asking an interesting question. Accumulation data is now added to Fig. 2 to address this point. The accumulation data shows that all the SSA samples on the first day of each decay event contained snow from more than one deposition event, given that no daily increase exceeded 2.5 cm. The uncertainty for the accumulation measurements is up to 1 cm, and therefore we cannot confidently use these measurements to approximate the number of precipitation events in one sample of snow. In addition, based on the data presented in this paper, we cannot be certain that a precipitation layer is not subsequently removed by the wind. This is added as a limitation to this study.

To account for this, we approximate the accumulation for the low-wind events, and use the field observations to identify the conditions preceding these events. The following text to the paper:

*"Both E10 and E11 had consistent clear sky conditions. We note here that E11 was preceded by significant ground fog, not snowfall, indicating that the peak value of 46 m2 kg-1 was likely the result of surface hoar, and thus, the SSA decay follows an SSA peak not caused by precipitation."*

*"As mentioned in Section 3.2, ground fog preceded the SSA peak in E11, corresponding to negligible accumulation. In contrast, approximately 1 cm of snow was accumulated during the day prior to E10, corresponding to observation of snowfall."*

L292-293: This appears to be a sentence with crucial interpretation of your records, which I think you should expand on. I am aware that the line between results and discussion section can be drawn before or after the interpretation of results, but I would like to see a clearer structure and separation from the discussion in the context of previous research.

We agree with this suggestion and propose to firstly include these previous observations more explicitly in the introduction. By focussing on the two low-wind events and their associated fluxes, the comparison to previous studies becomes more fluid. Quantification of sublimation flux during this period can potentially be added to quantify the fractionation effect. Uncertainty regarding the 2.5 cm bulk isotope measurement would hinder this approach.

*"Documentation of strong sublimation during the day and weak deposition during the night corresponds to decreases in d-excess (up to 6‰) and increases in d18O of up to 1.8‰. This observation agrees with previous observation of equilibrium fractionation during sublimation (Hughes et al., 2021; Wahl et al., 2021; Casado et al., 2021). "*

L304: We may be well familiar with this, but could you give a reference to back this statement, which is the result of earlier studies?

Apologies, the references Cabanes et al. (2002, 2003), Legegneux et al. (2003, 2004), Taillandier et al. (2007) and Flanner and Zender (2006) have now been added here which

documents the decreased rate of snow metamorphism with lower temperatures.

Figure 3: In my opinion, this is the most important figure when it comes to describing SSA observations and the model performance, and it nicely highlights the rapid decay at the start of SSA decay events. For some parts of your model performance discussion, it would be helpful to see observations and model outcome in one panel, so I suggest amending the current panel layout or splitting this figure into two figures about (a) observation and (b) model performance. Furthermore, I recommend using one term, i.e. 'rapid SSA decay events' or similar, throughout the manuscript to be more specific than 'events' here. And since you point out the higher intercept for temperatures <-25°C (L198) and same regression slope (L207), I suggest adding a linear regression line to panel (a) and noting somewhere that the x-axis doesn't extend to 0.

We agree with the reviewer here and have changed the figure to show a column with observed decays for b) moderate-wind events and c) low-wind events. A second column shows the modelled outputs for these events (d) and e)).

The model evaluation section is extended to include a comparison to existing models. This enables us to determine what is 'meant to be' according to physical models and compare this to observations and our model with parameters set for low-wind and moderate-wind events. The additional comparison to the moderate-wind events allows for a general assessment of the additional influence of wind on the decay rate.

The lower row of Fig. A2. now shows two events - E2 from 2017 and E18 from 2019 - that have maximum daily wind speed of 6.26 m s$^{-1}$ and 6.28 m s$^{-1}$ respectively, and no observed snowdrift. Based on the drift threshold defined in Li and Pomeroy (1998), E2 has potential influence from snowdrift, but not E18 (U(10) = 7.09 m/s and 8.17 m s$^{-1}$ for E2 and E18 respectively), which agrees with an underestimation of decrease from FZ06 compared to observation during E2. Interestingly, we get the lowest RMSE values for FZ06 and the moderate wind events. Possible explanations include the initial snow conditions and event duration, which are included in the discussion.

Figure 4: Is there a way to enhance the contrast between the thick line as mean and thinner individual records? Personally, I find it a challenge to see the thick line.

Yes of course, the plot has been changed to show the individual samples as crosses ('+').

**Discussion**

The discussion could benefit from a wider literature context.

The discussion has been largely rewritten and more relevant recent literature has been included in the discussion. Furthermore, changes have also been made to account for our changes in the structure of the results section and our methods related to event criteria. In this revised version of the manuscript, we first discuss the different regimes resulting from the EOF analysis, referring directly to papers such as Casado et al. (2021) where they identify the relative influence of precipitation and snow metamorphism on the isotopic signal in Antarctica.

Our approach to the decay model in the study is subsequently discussed, with a focus on the comparison to previous models. We highlight the limitations relating to the fact that the two low-wind events are from 2018 while the strong coherence between d-excess and SSA from EOF analysis is observed in 2019. We highlight that the purpose of SSA decay models to predict change in SSA of a snow sample through time is not readily applied to exposed surface snow, and that there is potentially an alternative direction for future studies to focus on the multiple mechanism of surface snow reworking, that would be useful for surface energy budget calculations using remote sensing.

The structure of the isotopic analysis has been modified for more stringent comparison of our results to expectations of isotopic change from previous studies. Some sections have been merged to be more concise.

L317-318: Here, it would be good to clarify that this is in agreement with the study Taillandier et al. (2007). Are there other studies that you could compare your approach/results to?

Of course, we develop this section based on results from the model intercomparison. This enables us to discuss our empirical model with respect to the physical based model from Flanner and Zender (2006) based on theoretical grain growth. The RMSE values in Table 2 indicate that FZ06 best predicts the decay of both low- and moderate-wind events.

L333: This sentence is actually the first time you state a causal connection between SSA and d-excess development, and I recommend including this section 4.3 earlier as part of the interpretation section.

We agree with the reviewer and follow the reformatted results structure to first discuss the results from EOF analysis. We focus on the inter-annual difference in regimes, where a very strong relationship is observed in 2019, while d18O and d-excess are decoupled, compared to the opposite relationships in 2018. Our results are compared to recent work from Casado et al., 2021, who document a similar inter-annual variability in Antarctica.

L351: It would be good to see references of earlier research on these factors, i.e. sublimation, deposition and vapour diffusion, cited here.

Of course, these references have been added to show the previous work that, specifically referring to Casado et al. (2021). In addition, we propose to add more structure to the isotopes discussion by primarily identifying the expectation of isotopic change during precipitation resulting from different processes (as mentioned in the response to the previous comment).

L372: While you identify 'initial snow metamorphism' after deposition as driver of d- excess, I think that you should be more specific and discuss the importance of deposition-free phases, here described as overcast and clear-sky conditions (Table A1), for d-excess.

This point from the reviewer is appreciated and we have added a more extensive description of conditions for the events used to assess isotopic change.

A detailed description of temperature gradients and latent heat flux data during the two low-wind SSA decay events allow us to identify the processes controlling the change in isotopic

composition. The text added in response to the comment starting L393 addresses.

L381: I understand that winter snow layers have undergone more isothermal metamorphism, which is less efficient than temperature-gradient metamorphism acting especially during spring and autumn. Therefore, I recommend rephrasing 'winter layers which are less influenced by snow metamorphism'.

We agree with the reviewer and propose the following text instead:

*"Snow metamorphism is thermally activated given the dominant influence of sublimation and deposition (Cabanes et al., 2002, 2003; Legegneux et al., 2004). During winter, the temperatures are very low (<-30C) and minimal insolation reduces the diurnal near-surface snow temperature gradients, resulting in isothermal metamorphism being dominant which reduces the rate of snow metamorphism, or SSA decay, compared to temperature gradient snow metamorphism (Dadic et al., 2008)."*

L393: Section 4.6 is a very interesting and important one. The first sentence of the second paragraph appears to be a major jump in the train of thought, which I struggle to follow. Especially, the last paragraph of the conclusion contains important findings. I wish to see the last statement (L426-428) put for discussion with the same clarity earlier in the manuscript. Then, the conclusion will become a summary.

We have explained the point in L426-428 with analysis of latent heat flux and temperature gradients corresponding to isotopic change during low-wind events. As the reviewer will see, the revised structure facilitates a more concise discussion with regard to processed driving isotopic change. The following text has been added to the discussion from which we compare our observations*:*

*"Three key mechanisms are expected to drive the rapid SSA decays; 1) large grains growing at the expense of small grains (Legagneux et al., 2004; Flanner and Zender, 2006), 2) diffusion of interstitial water vapour (Ebner et al., 2017; Touzeau et al., 2018; Colbeck, 1983), 3) sublimation due to the wind ventilating the saturated pore air, known as 'wind-pumping' (Neumann and Waddington, 2004; Town et al., 2008). The dominant mechanisms can theoretically be identified by a combination of the change in isotopic composition - indicating the fractionation effect - and the LE and temperature gradient data.*

*In theory, mechanism 1) causes minimal change in the bulk isotopic composition of a snow layer under isothermal conditions (Ebner et al., 2017). Therefore, observations of SSA decay corresponding to negligible isotopic composition change could be explained by this mechanism. We observe no events with consistent isotopic composition throughout. In the instance of 2) interstitial diffusion, light isotopes are preferentially diffused, while the heavy isotopes will be preferentially deposited onto the cold snow grains (Ebner et al., 2017; Touzeau et al., 2018; Colbeck, 1983). Thus, diffusion of water vapour in the pore space causes a decrease in d-excess and slight increases in δ18O due to kinetic fractionation (Casado et al., 2021). 3) Sublimation has been widely documented to cause an increase in δ18O of the remaining snow mass due to equilibrium fractionation, and a significant decrease in d-excess due to kinetic fractionation (Ritter et al., 2016; Madsen et al., 2019; Hughes et al., 2021; Wahl et al., 2021; Casado et al., 2021).*

*An overall increase in δ18O and decrease in d-excess during E10 can be attributed to a combination of 2) and 3) based on observation of net-sublimation and high amplitude diurnal temperature gradient variability indicating vapour transport within the pore space. The period between 9th June at 15:18 UTC and 10th June 10:40 UTC recorded net deposition corresponding to an overall decrease in δ18O during the first day and minimal decrease in d-excess, potentially due deposition of atmospheric water vapour (Stenni et al., 2016; Feher et al., 2021; Casado et al., 2021).*

*A 30% decrease in d-excess corresponds to negligible change in δ18O during E11. Net-sublimation double that of E10 is measured, but with reduced amplitude in both TGs. Moreover, the largest decrease in d-excess occurs after the first day when the surface-subsurface TG is consistently negative. This indicates that vapour diffusion is controlling the isotopic composition, and the effect of equilibrium fractionation during sublimation from the surface only weakly influences the bulk isotopic composition (Casado et al., 2021)."*

L414: In my opinion, simply applying this model at other sites goes a bit too far, because site-specific accumulation seasonality/frequency plays a major role for the near-surface metamorphism. I therefore suggest elaborating on the potential and limitations of the SSA decay model for other sites in greater detail in the discussion section.

We fully acknowledge this point from the reviewer, and instead compare to physical based models from the literature. The observed influence of wind-speed on the SSA decay rate is also discussed, with reference to the limitations of such field-based studies. For example, we are limited by persistent moderate winds potentially perturbed the snow surface, as evidenced by 209 out of the 287 sampling days (2017-2019) having maximum wind-speed above 5 ms-1 based on 10-minute mean values. Although there is a low probability of snowdrift up to 6 ms-1 based on the equation below defined by Li and Pomeroy (1998), we acknowledge the potential and highlight this as a limitation.

New text: *"The SSA decay model described in this study is intended as an investigation into the in-situ behaviour of surface snow SSA through time."*

L419: While you state earlier that d-excess varies with d18O at the beginning of the season and with SSA later during summer, you state that mainly SSA and d-excess are coupled. Please be consistent with your earlier interpretation here.

We apologise for the inconsistency here. The text has been edited to reflect the different regimes between the sampling years, where for 2019 the SSA and d-excess are coupled, while the δ18O and d-excess are coupled during 2018. We propose to remove any generalisation and focus instead on the potential causes for the opposing regimes. We discuss the EOF results in the context of the isotopic change during the SSA decay events to improve the coherence of the discussion.

When it comes to supplementary material, I could imagine a more detailed presentation of the AWS data to be helpful for readers as a meteorological background for this study. Figure A1 is already a good start, which a bit more of a description would help.

The supplementary information has been extended, with a number of plots to support

statements in the paper, specifically the covariance between principal components and wind-speed distributions for 10-minute data and the daily maximum values. In-text references to these figures have been updated and we hope this helps with clarity for explanations.

**Technical details**

L20: 'first order parameters' - here and in other parts of the manuscript, a hyphen is required ('first-order').

The text has been changed to "first-order parameters".

L105: As you are describing a value range here, an en-dash is needed for 2 -15–130 m kg . Same applies for value ranges throughout the manuscript.

All ranges presented in the manuscript have been corrected for this mistake.

L116: To avoid any misreading, I suggest that the equation is placed in a separate line and to replace the en-dash in 'd-excess' on the left side of the equation with a hyphen. This could also be a good place to give the d18O equation.

The d-excess equation has been moved to a separate line, we have ultimately decided not to include the d18O equation given the primary focus on SSA and d-excess.

L121: Since you are talking about events in time, 'where' should be replaced with 'when'.

This has been changed to:" This study focuses on the events when the SSA measurements decrease rapidly".

L132: Equation 1 requires a multiplication sign.

A multiplication sign has been added to equation 1 and the additional equation 2.

$$SSA(t) = SSA_0 \cdot e^{-\alpha t} \qquad SSA(t) = B - A \cdot ln(t + \Delta t),$$

L180: I think it would be helpful to reference that Table A1 is part of the Appendix. This applies here and for all other references to the Appendix/Supplementary Material. L190: 'snow fall' should be corrected to 'snowfall'.

The accumulation plot has been corrected and incorporated into the description of meteorological conditions at the start of the results. The supplementary material now includes the Table describing event conditions based on field diary observations, the relationships between the principal components from the EOF analysis, and the spatial variance of each relevant parameter (SSA, d18O and d-excess).

Please go through the entire manuscript once more and check:

The proper use and non-use of articles to achieve concise language; Inserting spaces between values and units; Introducing abbreviations when first used, both in the text and in figure captions.

We apologise for errors in the text. The revised manuscript will be thoroughly checked for all the above.

[revised manuscript text omitted]

65 depending on surface conditions (Cabanes et al., 2003; Pinzer and Schneebeli, 2009a), each potentially modifying the isotopic composition of the snow (Ebner et al., 2017).

Models can provide a quantitative description of the rapid SSA decrease after deposition. Previous studies have proposed SSA decay models using a combination of field measurements and controlled laboratory experiments (Cabanes et al., 2002, 2003; Legagneux et al., 2003, 2004; Flanner and Zender, 2006; Taillandier et al., 2007).

70  Exponential models to describe SSA decay are documented to be the best fitting to in-situ data (Cabanes et al., 2003). However, the lack of a physical basis led Legagneux et al. (2003) to construct a theoretical equation to describe SSA decay based on grain growth theory, which was then developed by Flanner and Zender (2006) who defined parameters based on surface temperature, temperature gradient and snow density.

75 Existing SSA decay models have not yet been extensively applied to polar ice sheet surface snow. Conditions for surface snow on polar ice sheets  are not necessarily comparable to other alpine  and Arctic regions regarding negligible melt and the high-latitude radiation budget

 . Moreover, while continuous surface SSA measurements exist from

[revised manuscript text omitted]

Mean and standard deviation for weather variables, surface temperature (calculated from upwards and downwards long-wave radiation with long-wave em... set to 0.97), relative humidity, wind speed and latent heat flux during the three sampling seasons. Surface temperature, relative humidity and wind sp... PROMICE weather station based on 10-minute measurements. Latent heat flux an upwards flux from the eddy-covariance tower.

sample with a known volume. At the start of each season, sticks were placed at each site and snow height was determined by the distance between the snow surface and top of the stick. Accumulation was calculated using the cumulative sum of the daily difference between measurements of snow height from each site. The resultant datasets consist of 10 daily measurements of three parameters, SSA, density and accumulation, over a  89-, 94- and 66-day period for 2017, 2018 and 2019 respectively.

 The samples were all taken in the day time, primarily in the morning.  The meteorological data is re-sampled to the SSA sampling time-periods to ensure consistent comparison.

**2.3 SSA measurements**

**2.4**

Each snow sample is placed into the Ice Cube sampling container below an Infra-Red (IR) laser diode (1310 nm), where the SSA is calculated based on IR hemispherical reflectance, explained in Gallet et al. (2009), while information on the Ice Cube device can be found in Zuanon (2013).  The *e*-folding depth of 1310 nm radiation in snow of 200 kg m$^{-3}$ is approximately 1 cm  2017, 2018 and 2019 is 293 kg m$^{-3}$ (307±40 kg m$^{-3}$, 278±47 kg m$^{-3}$, 294±50 kg m$^{-3}$ )for 2017, 2018 and 2019 respectively), resulting in each measurement being heavily weighted to the top <1 cm of the 2.5 cm sample. The light reflected from the snow samples is converted into inter-hemispheric IR reflectance using a calibration curve based on methane absorption methods (Gallet et al., 2009). A radiative-transfer model is used to

retrieve SSA from inter-hemispherical IR reflectance. To avoid influence from solar radiation, SSA was measured inside a ventilated white tent kept at temperatures between -5 °C and -10 °C. SSA measurements have an uncertainty of 10 % for values between  5–130 $m^2\,kg^{-1}$ (Gallet et al., 2009).

**2.4 Surface snow isotopes**

 Individual SSA samples were put in separate bags and subsequently sampled for water isotopic composition. Thus, every day the 10  SSA samples have a corresponding isotopic composition. The resultant isotope value is the average composition over the top  2.5  cm of snow. Each sample was sealed in polyethylene bags to avoid any air to equilibrate with the snow and affect the isotopic composition. All samples were kept frozen during transportation and storage.

After melting, each bag was shaken to ensure the isotopic composition of the sample is representative. $1.25\,\mu l$ of each sample was then pipetted into a vial ready for isotopic analysis. The snow samples were then analysed at Alfed Wegener Institute in Bremerhaven using a cavity ring-down spectroscopy instrument model Picarro L-2120-i and L-2140-i following the protocol of Van Geldern and Barth (2012). This technique is used to obtain measurements of $\delta^{18}O$ and $\delta D$ with an uncertainty of 0.15‰ and 0.8‰ respectively.  The calcualted values for $d$-excess have an uncertainty of 1‰. Observing relationships between our SSA and isotope data requires consideration for the depth offset between the SSA measurements and the isotopic composition measurement which measures the entire 2.5 cm snow layer.

**2.5 Data analysis**

**2.6**

**2.5.1 Defining SSA decay events**

 To systematically identify rapid decreases in SSA, which we use as a proxy for events of snow metamorphism after deposition (identified based on the high mean SSA values), a threshold is set using the bottom 10th  percentile of SSA decreases over a two-day period (-13 $m^2\,kg^{-1}$  2-$day^{-1}$). This was found to result in the most equal number of events from each sampling year compared to 1- and 3-day changes. SSA decay events are defined as by the initial peak, identified by the threshold, through to the next increase in SSA (rather than decrease).

We here use the term deposition events to describe rapid increases in SSA, expected to be from precipitation, drifted snow

or hoar formation. Previous studies have indicated that surface hoar and sublimation crystal-like grain growth features at the surface have an SSA value around $54\,\mathrm{m^2\,kg^{-1}}$, based on    SSA of hoar frost (Domine et al., 2009). Accumulation data and field observations are used to identify the initial conditions.

A set of criteria are required to reduce the potential of analysing events with wind-perturbed surfaces, resulting in the removal of surface snow. In Antarctica, unconsolidated surface snow has been observed to drift at wind speeds as low as $5\,\mathrm{m\,s^{-1}}$ measured at $2\,\mathrm{m}$ height (Birnbaum et al., 2010). However, a study from Greenland documented snowdrift starting at $6\,\mathrm{m\,s^{-1}}$ (Christiansen, 2001), likely due to warmer temperatures allowing for the surface snow to become more bonded (Li and Pomeroy, 1997). At EastGRIP, calm conditions correspond to wind speeds from $0$–$5.2\,\mathrm{m\,s^{-1}}$ according to field diary observations. The mean daily maximum wind speed for the three sampling seasons was $6.8\,\mathrm{m\,s^{-1}}$, while blowing snow was documented only when wind speeds exceeded $7\,\mathrm{m\,s^{-1}}$.

Based on this assessment, we define two wind-speed categories for comparison of the effects of wind-speed on SSA decrease. The first includes events with wind-speed consistently below $5.2\,\mathrm{m\,s^{-1}}$, hereafter referred to as low-wind events, to ensure no surface perturbation. Secondly, we consider events where the maximum wind-speed is between $6$–$7\,\mathrm{m\,s^{-1}}$, hereafter referred to as the moderate-wind events. The inclusion moderate-wind events allows an assessment of the influence of wind-speed on SSA decrease.

**2.5.2  Modelling surface snow metamorphism**

The first empirical SSA decay model was proposed by (Cabanes et al., 2003) who described a temperature-dependent exponential decay based on snow samples collected from the Alps (Cabanes et al., 2002) and Arctic Canada (Cabanes et al., 2003). A following logarithmic equation (Eq. (2)) fit controlled to laboratory experiments was proposed by (Legagneux et al., 2003), where parameters A and B were found to be arbitrarily related to the decay rate and initial SSA of each sample, and are linearly correlated at -15 °C.

$$SSA(t) = SSA_0 \cdot e^{-\alpha t} \tag{1}$$

$$SSA(t) = B - A \cdot ln(t + \Delta t) \tag{2}$$

To improve the physical basis of the model, the theory of Ostwald Ripening, describing grain growth driven by a physical need to reduce surface energy, was implemented into the model (Legagneux et al., 2004). The equation (Eq. (3)) has two parameters $\tau$ and $n$; $\tau$ is the decay rate and $n$ relates to theoretical grain growth. The physical model was developed by Flanner and Zender (2006) to incorporate more specific physical quantification to the parameters to include information about

temperature, temperature gradient, and density. Based on these three conditions, they created a look-up table for $\tau$ and $n$.

$$SSA(t) = SSA_0 \left( \frac{\tau}{t + \tau} \right)^{1/n} \tag{3}$$

Taillandier et al. (2007) proposed two equations based on the logarithmic model, defined by Legagneux et al. (2004), to define the decay rate under isothermal and temperature gradient conditions where they were able to directly incorporate a surface temperature parameter.

An empirical decay model is constructed building upon previous studies (Cabanes et al., 2002, 2003; Flanner and Zender, 2006; Legagneux et al., 2002, 2003; Taillandier et al., 2007). This model uses continuous daily SSA measurements from EastGRIP to describe the behaviour of surface snow SSA in polar summer conditions. All samples of defined SSA decay events are used to quantify surface snow metamorphism.

$$SSA(t) = SSA_0\, e^{-\alpha t}$$

**3  Results**

**3.1  EastGRIP conditions**

Meteorological variables over the three sampling seasons vary substantially. Figure 2 shows the 10-minute mean values of air temperature, wind-speed, relative humidity and latent heat flux (LE). The accumulation in Fig. 2d are daily mean values (see Section 2.2). Air temperatures were below 30 °C between May 5th and May 8th, such low temperatures were not recorded for 2017 and 2019. However, when comparing the period from May 27th (start of 2019 season) to August 5th of each year, 2018 air temperatures (-13.3 °C) were still 0.5 °C lower than 2017 and 3.2 °C lower than 2019. Two days during 2019 recorded air temperature above 0 °C.

The 2017 season was characterised by high wind intrusions of $>13\,\mathrm{m\,s^{-1}}$  at approximately 20-day intervals. Considering all three sampling years, 2017, 2018 and 2019, the average daily maximum wind speed is $7\,\mathrm{m\,s^{-1}}$, with 209 out of the total 237 sampling days having maximum wind speed above $5\,\mathrm{m\,s^{-1}}$. The distributions of daily maximum wind-speed compared to 10-minute mean values are found in the Supplemental Fig. A1. Relative humidity is consistent throughout the years with mean values around 95% and similar variability of $\sim$7%.

[Figure]

**Figure 2.** *Meteorological data from 2017, 2018 and 2019*

Data is presented for the specific sampling periods for each year. The 10-minute mean data from PROMICE is shown for air temperature (a), wind-speed (b) and relative humidity (c). The bold lines indicate the mean values, based on the snow sampling time interval, for air temperature and relative humidity, and the maximum value for wind-speed. The Relative humidity is determined from vapour pressure and saturation vapour pressure. Latent heat flux (d) is 10-minute averages from the eddy-covariance tower, with the bold grey line showing the daily sum. Accumulation is presented in panel d).

**4 Results**

**3.1 SSA decay events**

There was a total of 5 cm accumulated snow over the 89-day season of 2017, half the amount of 2018 and 2019. The field season for 2018 started on the 5th of May, 9-days earlier than 2017 (14th May), and 22 days earlier than 2019 (27th May). Substantially more sublimation was recorded in 2019, where the daily sum was approximately double that of 2018.

**3.0.1 Spatial and temporal surface variability**

 A recent study at EastGRIP has shown the significant
in-homogeneity in surface snow due to post-depositional  reworking of the snow

 (Zuhr et al., 2021). To avoid attenuation of isotopic signal, each
sample is treated independently. Using a confidence interval of 95% (p<0.05), the relationship between SSA and isotopic

[Figure]

**Figure 3.** **Time-series of SSA  (a) d18O (b),
*d*-excess (c)  and the  principal components (PC1) of each variable (d). For each plot,
the  markers indicate the individual sampling  sites and the  link shows the daily mean The secondary
y-axis in panel a) shows the accumulation.  The grey bars  indicate the
SSA  decay events
.

composition is tested using Empirical Orthogonal Function (EOF) analysis. The purpose of EOF analysis is to identify the dominant modes of variance in both the temporal and spatial dimensions for each parameter - SSA,  and $d$-excess - which are all measured from the same sample.

All parameters continuously change throughout the field seasons of 2017, 2018 and 2019  (Fig.3), with large spatial variability in isotopic composition. SSA is characterised by peaks, often corresponding to large spatial variability, followed by gradual decreases over a number of days, a feature which is most prominent during 2017 and corresponds to negligible accumulation. The amplitude of SSA variability is largest in 2019. The start of the 2018 season has very high SSA values (daily mean $88\,\mathrm{m^2\,kg^{-1}}$) corresponding to low and homogeneous $\delta^{18}$O. Maximum SSA values  of individual samples for 2017, 2018 and 2019 are 85.3$\,\mathrm{m^2\,kg^{-1}}$  95.3$\,\mathrm{m^2\,kg^{-1}}$ $^{-1}$ and $86.7\,\mathrm{m^2\,kg^{-1}}$ respectively.

 Inter-annual variability is observed in $\delta^{18}$O, with seasonal mean values of -31.6‰, -32.7‰ and -27.3‰ for 2017, 2018 and 2019 respectively (Fig. 3a). Note that the 2019 field season started approximately 15 days later than 2017 and 2018, resulting in a bias towards mid-summer conditions. Throughout the season $\delta^{18}$O follows a gradual increasing trend from May to August. Some cases of abrupt decreases (-10‰) are observed in the late summer, for example, on July 12th in 2018 and July 25th in 2019. No clear seasonal trend is observed in $d$-excess (Fig. 3b) but with periods of gradual decreases. Total daily spread in $\delta^{18}$O and $d$-excess is approximately 15‰.

The spatial and temporal principal components of each variable are presented in Fig.  3d. During 2017, 2018 and 2019 all variables have one dominant mode of variance, or principle component (PC1). PC1 of SSA (PC1$_{\mathrm{SSA}}$) explains 61%, 77% and 72% of variance for the respective years, PC1 of $\delta^{18}$O (PC1$_{\delta^{18}\mathrm{O}}$) explains 69%, 83% and 75% of the total variance respectively, while PC1 of $d$-excess (PC1$_{d-\mathrm{excess}}$) explains 47%, 51% and 60%.

Distinct differences are observed between the sampling years, most prevalent is the opposing regime from 2018 to 2019. During 2018 PC1$_{\delta^{18}\mathrm{O}}$ and PC1$_{d-\mathrm{excess}}$ exhibit a significant relationship, with a strong negative correlation for the spatial component of PC1$_{\delta^{18}\mathrm{O}}$ and PC1$_{d}-\mathrm{excess}$. A significant relationship is also observed for the temporal component of PC1$_{\mathrm{SSA}}$ and PC1$_{\delta^{18}\mathrm{O}}$. In contrast, data from 2019 are characterised by significant relationships between PC1$_{\mathrm{SSA}}$ and of PC1$_{d}-\mathrm{excess}$ in both the spatial (r=0.75) and temporal dimensions. No relationship is observed between PC1$_{\delta^{18}\mathrm{O}}$ and PC1$_{d-\mathrm{excess}}$ during 2019. For 2017, significant relationships (p<0.05, 95% confidence) are observed between the temporal component of PC1$_{\mathrm{SSA}}$ and PC1$_{d-\mathrm{excess}}$, and the temporal and spatial component of PC1$_{\delta^{18}\mathrm{O}}$ and PC1$_{d-\mathrm{excess}}$. A shift is observed after July 15th where PC1$_{d-\mathrm{excess}}$ changes from co-varying with PC1$_{\delta^{18}\mathrm{O}}$ to PC1$_{\mathrm{SSA}}$.

**3.1 SSA decay events**

290 ## 3.2

Continuous SSA measurements allow for the construction of an empirical model to describe SSA decay at EastGRIP through time while exposed to surface processes. A visual inspection of the SSA decay events highlighted in Fig. 3a indicates a relationship between initial SSA and subsequent magnitude of decrease. Prior to analysis,

295 we assess the meteorological conditions and field observations to remove SSA decay events with potentially perturbed surface snow. This includes all events coinciding with observations of ground fog, snowdrift, and snowfall (indicated in Fig. 3), and events where the wind-speed exceeds the thresholds defined in Section 2.5.1.

From the years 2017, 2018 and 2019 a total of 21 events are identified that fulfil the SSA decay criteria (as defined in Section 2.5.1). These events are named E1, E2 etc (see Table A for more information on the individual events). Exploring

300 weather conditions for these events reveals that 12 out of the 21 events are influenced by either snowdrift, snowfall, or ground fog according to field diary observations. Of the remaining 9 events, two are in the low-wind category (E10 and E11$=5.1\,\mathrm{m\,s^{-1}}$), and 7 in the moderate-wind category. Both E10 and E11 had consistent clear sky conditions. We note here that E11 was preceded by significant ground fog, not snowfall, indicating that the peak value of $46\,\mathrm{m^2\,kg^{-1}}$  was likely the result of

305 surface hoar, and thus, rapid SSA decay follows an SSA peak not caused by precipitation.

SSA samples are treated individually to quantify SSA decay rate for the different categories. The rate of SSA decay is closely linked to the SSA value at the start of each event (initial SSA vs. magnitude of decrease during the decay period  (Fig. 4), suggesting the rate of change is proportional to the absolute value, as described by exponential decay law (r=-0.71 and r=-0.84 for low- and moderate-wind events respectively) (Cabanes et al., 2003).

310 The mean air temperature for all SSA decay events was between -20.8 °C and -7 °C. The first day of each event is characterised by the largest change in SSA, followed by a decrease in magnitude over the subsequent days. This feature is most apparent for the longer events (E1, E2 and E4), where SSA has minimal change below $25\,\mathrm{m^2\,kg^{-1}}$.

**3.2 EastGRIP SSA decay model**

315 SSA decay rate is quantified by plotting the rate of change in SSA per day against the absolute SSA value for all 10 sampling sites for  low- and moderate- wind events (Fig. 4a). We observe a linear relationship between the rate of change in SSA per day (ΔSSA) and SSA.

320  An overview of event conditions using field observations are presented in Table A.

[Figure]

**Figure 4.** *Decay Model Construction and Predictions*

Linear regressions for change in SSA against the SSA for the low-wind (blue) and moderate-wind (purple) SSA decay events (a). Filled markers indicate the daily mean values and transparent markers show the individual samples sites. The observed SSA decays are show for the moderate-wind events (b), and the low wind events (c), followed by the modelled SSA decays for the respective events in d) and c). The legend in d) and e) indicates the SSA decay event number, presented in Table A

 The SSA decay model for EastGRIP is  constructed using the differential equation for the linear relationship between ΔSSA and absolute SSA Solving the differential with respect to time (t), produces the SSA decay model defined as Eq. (4), which follows the equation structure  of Eq. (1).

325
$$\frac{dSSA}{dt} \quad SSA(t) = -0.54 \, (SSA_0 - C) \, e^{-\alpha \cdot t} + 14.69 \, C \tag{4}$$

$$SSA(t) = (SSA_0 - 26.8) \, e^{-0.54t} + 26.8$$

Where SSA(t) is the SSA measurement at a given time, $SSA_0$ is the initial SSA value,  $\alpha$ is the decay rate, and $C$ is the constant. The decay rate, determined by the slope of the linear regressions in Fig. 4, is higher for moderate-wind SSA
330 decay events (-0.53 m$^2$ kg$^{-1}$ day$^{-1}$ ) than for low-wind SSA decay events (-0.41 m$^2$ kg$^{-1}$ day$^{-1}$). To account for a non-zero decay constant,  $C$ describes the 'background' SSA state which is defined by the value of $x$ when the linear regression crosses the  $y$-axis in Fig. 4a

**3.2.1**

335   **~~All samples for all events are included in plot a) showing the relationship between the rate of change in SSA per day ($\Delta$SSA day$^{-1}$) against the daily absolute values. Points are coloured by the daily mean surface temperature. The linear regression is based on values for surface temperatures between -25°C and 0°C, and daily mean wind speeds below $6\,\mathrm{m\,s}^{-1}$. b) shows a comparison between the model predicted SSA values using Eq. (4), against the SSA observations. The marker colour represents the day of the events (DOE). Marker style represents the sampling year to~~
340

    , equal to $21\,\mathrm{m}^2\,\mathrm{kg}^{-1}$ and $24\,\mathrm{m}^2\,\mathrm{kg}^{-1}$ for low- and moderate-wind
345   events respectively. Note that events are here named E1, E2  etc. consistent with Table A.

350

   have an extent of 2-5 days.  E9 in 2018 is poorly represented by the moderate-wind SSA decay model from this study. The mean air temperature for this event was -20.8°C, 5°C less than the next coldest (E11 at -15.3°C). Fitting the model for E9 alone gives a decay rate of
355   $0.44\,\mathrm{m}^2\,\mathrm{kg}^{-1}\,\mathrm{day}^{-1}$, similar that of the low-wind events.  We therefore observe a temperature dependence of SSA decay similar to Cabanes et al. (2003). Based on limited number of events, we document low-winds having a similar effect to
360   air temperatures below -20°C on the SSA decay rate. Our results indicate a slower rate of decay under decreased wind-speed conditions. A similar effect is observed for low temperature, as the single SSA decay event in  moderate-wind category but with mean air temperature below -20°C followed the decay rate of  the
365

    low-wind events.

**3.2.1 Model evaluation**

**Table 2.** *RMSE - Model comparison*

| | Low-wind | | Moderate-wind | | |
|---|---|---|---|---|---|
| | Mean | Individual | Mean | Individual | |
| | $m^2\,kg^{-1}$ | $m^2\,kg^{-1}$ | $m^2\,kg^{-1}$ | $m^2\,kg^{-1}$ | |
| This Study | 3.64 | 4.76 | 2.48 | 3.50 | |
| FZ06 | 3.45 | 7.08 | 1.28 | 2.92 | |
| T07 | 6.34 | 7.11 | 5.63 | 6.10 | |

This Study uses the respective $\alpha$ and $C$ for the low- and moderate-wind events, using daily (mean) and individual samples. FZ06 parameters $\tau$ and $n$ are defined by the look-up table from Flanner and Zender (2006). T07 uses the mean surface temperature for each event.

Model performance is tested by 1) comparing daily predicted decrease to the 10 daily observations, and 2) comparing results
370 from this study to previous models from Flanner and Zender (2006) and Taillandier et al. (2007). Model-data residuals are normally distributed, suggesting no systematic errors in model predictions. The root mean squared error (RMSE) between model predictions and observed SSA is $4.76\,m^2\,kg^{-1}$ and $3.50\,m^2\,kg^{-1}$ for the low-wind and moderate-wind SSA decay events.

375 Using the physical-based decay model from Flanner and Zender (2006), hereafter referred to as FZ06, the influence of wind-speed on observed SSA decay rate can be assessed. Low-wind SSA decay events (E10 and E11) are most accurately predicted by FZ06 using the parameter values of $\tau =$ 4.5 and $n$
380  6.1 from the look-up table (Flanner and Zender, 2006). Both the empirical model from this study, and the model from Taillandier et al. (2007), hereafter T07, underestimate the rate of decrease for low-wind decay events, most apparent during the first day of the event E10 (see Fig. A3).

385 ### 3.2.2

weather variables, relative humidity, surface temperature and wind speed on the model-data residuals, with linear regressions resulting in r² < 0.1 for all variables. An overview of event conditions using field observations are presented in Table A. Temperatures below -25°C are characterised by the same slope defined by the model (-0.54The data indicates that in natural conditions, wind-speed (between 6 m² kg⁻¹ day⁻¹), but with a significantly higher intercept of 29 m s⁻¹ day⁻¹ compared to 14.7and 7 m s⁻¹ day⁻¹ for temperatures above -25) increases the surface SSA decay rate ($\alpha$°C. Significant wind drift is expected when hourly mean wind speed exceeds 6= m-0.53, $\alpha$ s⁻¹, which happens during 144 days out of the total 258 sampling days from 2017, 2018 and 2019. Results indicate weather has no systematic influence on SSA decay during the first 2-5 days exposed at the surface, and that conditions vary for each event. The model is able to predict all defined decay events between -25=°C and 0°C, indicating mechanisms of decay are the same. Daily mean values are more accurately predicted by the SSA decay model than individual sample sites due to snow surface variability. In-homogeneous surface snow is especially important to consider for isotopic composition, because there is potential for samples to contain snow from different precipitation and/or deposition events. -0.41). RMSE values presented in Table 2 indicate that FZ06 predicts decay with the least error, for both wind-speed categories. Moreover, all models have lowest errors when predicts events in the moderate-wind category.

**3.2.2 Surface snow spatial variability**

*Timeseries of snow isotopes and SSA*Timeseries of $\delta^{18}$O (a), d-excess (b) and SSA (c) for 2017, 2018 and 2019 sampling seasons. d) shows the principle components of each parameter with colors corresponding to the color used to show absolute values. The black vertical lines indicate a break in the x-axis. Each faded line represents individual sample site values, and the thick line is the daily mean. Grey shaded regions indicate periods of high spatial variability in isotopic composition.

**3.3 Isotopic change decay during events**

The characterization of the SSA decays provide a basis to explore how snow metamorphism of surface snow plays a role for the alteration of isotopic composition of Greenland snow after deposition. A recent study at EastGRIP has shown the significant in-homogeneity in surface snow due to post-depositional reworking of the snow (Zuhr et al., 2021). The focus for this manuscript is to identify signal coherence between physical properties and isotopic composition of surface snow subject to precipitation/deposition and post-depositional processes. Autocorrelation analysis shows that isotopic composition values are spatially decorrelated after 10 m (r² < 0.3 after 10 m). Therefore, to avoid attenuation of isotopic signal, each sample is treated as independent. Isotopic composition is measured from each SSA sample containing snow from the top 2.5 cm of the snow surface , potentially containing snow deposition layers from multiple precipitation events. Surface heterogeneity is considered by using Empirical Orthogonal Function (EOF) analysis to determine the dominant mode of variance for each sampling year. Figure **??** shows timeseries of

The rate of change in SSA during low- and moderate-wind events is explored with respect to the rate of change in isotopic composition, given the covariance identified from EOF analysis. The rate of change in *d*-excess is plotted against the rate of change in SSA (Fig. 5), considering 1- and 2-day time intervals. We here include analysis of 2-day to allow isotopic

[Figure]

**Figure 5.** Isotopic change during all the analysed events are shown, with each point indicating a specific sampling site. The daily change in *d*-excess (d$_{xs}$) and SSA is presented in a), with 0 indicated with the grey dotted lines. The change in *d*-excess and SSA over a 2-day period is shown in b), while the change in *d*-excess is plotted against the absolute *d*-excess values is shown in c). Linear regressions are presented from daily change (light green) and 2-day change (dark green).

equilibration between the existing surface snow and snow deposited in the day preceding the event. The change after 2-days is presented in Table 3 for each low- and moderate-wind event.

All events have an overall change in isotopic composition, with the percentage change in *d*-excess being an order of magnitude higher than that of $\delta^{18}$O
425  . Increasing $\delta^{18}$O

 corresponds to decreasing *d*-excess in 5 out of 8 events. E9, E11 and E13 deviate from this pattern. E9 and E13 both exhibit increases in $\delta^{18}$O
430  and  *d*-excess, whereas E11 is characterised by a slight decrease in $\delta^{18}$O and 27% decrease in *d*-excess.

Using a significance level of 0.05, the relationship between change in *d*-excess ($\Delta d$-excess) and change in SSA ($\Delta$SSA)
435 is assessed. The results are presented in Fig. 7. Firstly, the $\Delta d$-excess over 1-day are normally distributed around a mean of

$\Delta d$-excess values < 4%  tend to correspond to smaller $\Delta$SSA ( -15
440 m$^2$ kg$^{-1}$ –0

 Fig. **??**d)$m^2\,kg^{-1}$), suggesting that large decreases in $d$-excess occur after an extended period of exposure. This feature is highlighted in Fig.

~~Surface variability due to post-depositional reworking of the snow is shown by a wide spread in SSA values during a given day. Time periods with low spatial variability indicate largely homogeneous snow cover over the transect, shown in Fig. **??** as shaded regions. High variability is defined by periods where the 5-day running-mean of spatial variance in $\delta^{18}$O is greater than one standard deviation. During periods of low spatial variability in isotopic composition, there is greater coherence between PC1 of SSA and PC1 of d-excess, due to a reduction of noise in the dataset. PC1 of SSA and d-excess show a coherence during 2018 and 2019 seasons, while the signal is less clear during 2017However, the reduced signal coherence is concurrent with high spatial variability in isotopic composition.~~ 7c), with high initial $d$-excess corresponding to the largest decreases in $d$-excess.

**Behaviour

**Table 3.** Table of  isotopic change

| | | | | | | | | | |
|---|---|---|---|---|---|---|---|---|---|
| | | | | | **E1** | | | | |
| | | | | | **E2** | | | | |
| | | | **E3** -29.55 -30.07 -1.8% -0.2 -0.6 -200% 50.4 31.1 -38.3% | | **E4** | | | | |
| | **E5** -30.23 -29.88 1.2% 6.1 2.9 -52.5% 45.2 31.5 -30.2% | **E6** -30.50 -30.36 0.5% 9.8 6.2 -36.7% 47.9 31.8 -33.6% | **E8** | | | | | | |
| | | | | | **E10** | | | | |
| | | | | | **E11** | | | | |
| | | | **E12** -29.57 -29.33 0.8% 4.6 3.1 -32.6% 49.7 37.7 -24.1% | | **E13** | | | | |
| **E14** -33.89 -34.26 -1.1% 11.5 12.3 7.0% 53.2 37.4 -29.7% | **E15** -32.35 -31.94 1.3% 8.6 7.0 -18.6% 57.1 31.1 -45.6% | **E16** -29.06 -28.97 0.3% 7.1 4.9 | | | | | | | |
| | **E19** -23.40 -23.31 0.4% 6.7 6.8 1.5% 47.7 29.4 -38.2% | **E20** -22.27 -22.26 0.1% 4.4 3.3 -25.0% 60.3 33.6 -44.2% | **E21** | | | | | | |

Behaviour of snow parameters during decay events are defined. The initial, 2-day and percentage change over a 2-day period are presented for $\delta^{18}$O, $d$-e

**3.3.1 Low-wind event analysis**

**3.3.2**

[Figure]

**Figure 6.** Isotopic change analysis for low-wind events, E10 and E11. Panel a) shows daily change in *d*-excess against change in d18O for E10 and E11 with corresponding linear regressions, b) shows change in *d*-excess against change in SSA, and c) shows change in d18O and change in SSA. The r- and p-value for each regression are indicating in the corresponding colours.

[Figure]

**Figure 7.** Latent heat flux (LE) (grey), relative humidity with respect to ice (purple), air-surface temperature gradient (TG) (red) and surface-10 cm subsurface TG (red) for the low-wind SSA decay events, E10 (a) and E11 (b) (Table Dark grey shading in LE indicates sublimation and light grey shows deposition.

~~Using all 10 sample sites as independent values, the behaviour of isotopes during defined SSA decay events is analysed. To determine the isotopic change in the surface snow during rapid SSA decays, the rate of change in d-excess is plotted against the rate of change in SSA (Fig. 5). The change in SSA over a 2-day period is used. The daily mean change over the first 48h of each event is presented in Table 3.~~

 As mentioned in section 3.1, ground fog preceded the SSA peak in E11, conccurent with negligible accumulation recorded. In contrast, approximately ~~1±1‰ mean increase, with the exception of E17 and E21 in 2019 (See Table 3). E17 is characterised by significant ground fog and snowfall during the event, while E21 has negative LHF (net-deposition) measured from the eddy-covariance system over the event. The percentage change of d-excess is an order of magnitude higher than $\delta^{18}O$ - expected due to the definition of d-excess - and similar to SSA , with 14 out of 19 events showing a decrease in d-excess during the first 2-days of each event. Further analysis looks specifically at the relationship between d-excess and SSA given the coherence observed between their PCs, and the significant change observed in Table 3.~~ cm of snow was accumulated during the day prior to E10, corresponding to observation of snowfall.

 Figure 6 shows the relationship between  the daily change (Δd-excess) and change in SSA over the same time period (ΔSSA) is assessed.  in isotopic composition and SSA. For E10, both $\Delta\delta^{18}O$ and $\Delta d$-excess and $\Delta$SSA and $\Delta d$-excess have significant negative correlations (r  =‰ (Fig. 7b) -0.5, r=-0.8). ~~Thus suggests either low d-excess of deposited snow, or old snow that has been re-exposed. In addition, initial d-excess is observed to significantly influence that magnitude of d-excess change over the subsequent 48h of rapid SSA decay (Fig. 7a and b). The largest changes in d-excess corresponds to high initial d-excess values. Moreover, increases in d-excess during rapid SSA decay follow very low initial d-excess values. In summary, in 72% (78 out of 108 samples) of cases decreases in SSA correspond to a decrease in d-excess of the snow sample during the first 2-days. Moreover, the magnitude of change in d-excess during rapid SSA decay shows a weak but significant dependence on the initial d-excess signal.~~

 =‰ 0.6), while no significant relationship is observed between the Δ-parameters during E11. All samples exhibit negligible change (<0.7‰) in $\delta^{18}O$ during E11.

The dominant direction of vapour flux is assessed using air, surface and subsurface (10cm depth) temperature data and LE between the snow and atmosphere. Net-sublimation is observed during both E10 and E11, with a total sum of 33.9 $Wm^{-2}$ and 55.8  -7.7 $Wm^{-2}$ for the respective events. The LE is controlled primarily by the temperature gradient (TG)

between the air and the surface, with strong sublimation (>

495
10 % chance of decreasing during surface snow metamorphism (SSA

W % of the

500
 m⁻²), corresponding a negative TG between the air and surface of 2.5°C on June 10th. A
concurrent upwards vapour flux is indicated based on the TG between the subsurface and surface snow. Downwards LE flux up
to 4 W m⁻¹ is observed each night corresponding to the transition from a negative to positive TG between the air and surface.
The period between sampling on 9th June at 15:18 UTC and 10th June 10:40 UTC recorded net deposition, corresponding to
505 significant increase in $\delta^{18}$O and decrease in *d*-excess.

The amplitude of all parameters is during for E11 compared to E10. A negative surface-subsurface TG persists throughout
the first day of E11, indicating a downwards vapour flux.

**4 Discussion**

Continuous daily SSA measurements at EastGRIP during the summer season of 2017, 2018 and 2019 have enabled quan-
510 tification of variations in snow physical properties due to  deposition and snow metamorphism.
Understanding the relationship between rapid decreases in SSA and corresponding change in isotopic composition require
clearly defined events and environmental context. Using a

 set of criteria, nine SSA decay events during precipitation-free
515 periods are defined and used to construct an empirical decay model. We firstly discuss the behaviour of SSA

 decay at EastGRIP compared to existing models. The isotopic change
associated with low-wind SSA decay events is then considered, in the context of sublimation, vapour diffusion and wind effects
(Ebner et al., 2017; Hughes et al., 2021).

520

 SSA decay  at EastGRIP]

525  SSA decay  at EastGRIP

The empirical decay model defined in this study accurately predicts the SSA decay of surface snow at EastGRIP over a limited time-period. We find that rapid SSA decay events are best described by an exponential

530  decay function, in agreement with observations from Cabanes et al. (2003). The expected temperature dependence on the SSA decay rate ~~during events had no systematic influence from weather variables (wind speed, temperature and relative humidity). The only exception is for temperatures outside the set range for the model. Surface temperatures below -25 °C were characterised by a significantly higher background SSA (defined as the mean SSA value of the final day of decay events) (Fig. 4), indicating high background SSA due to reduced snow metamorphism~~ is apparent

535 during E9, where the mean air temperature is less than -20 °C, which is in agreement with the accepted knowledge that snow metamorphism is slower in colder conditions  due to sublimation and deposition  being thermally activated processes

540  (Cabanes et al., 2003; Legagneux et al., 2003; Flanner and Zender, 2006; Taillandier et al., 2007). The narrow temperature range of SSA decay events does not facilitate a conclusive definition of a temperature-dependent decay rate.

In addition, we focus on the influence of  wind-speed of the SSA decay rate

545  and observe a more rapid SSA decay with increased wind-speed, potentially due to increased ventilation of saturated pore air acting as a catalyst for snow metamorphism (Cabanes et al., 2003; Flanner and Zender, 2006; Neumann and Waddington, 2004). Wind erosion cannot be definitively ruled out due to dis-continuous documentation of surface conditions. However, in some cases, high wind speeds are documented to increase SSA due to fragmentation and sublimation of suspended snow grains, which are then re-deposited and effectively

550 sieved into the pore spaces of the surface snow layer (Domine et al., 2009).

~~The top 1 cm of the 2.5 cm SSA sample is measured by the Ice Cube device, and thus, is most likely to capture the precipitation signal (Gallet et al., 2009; Klein, 2014). Directly after precipitation, isothermal snow metamorphism is expected to be dominant due to to high surface curvature of fresh snow crystals (Colbeck, 1980). Alternative SSA decay models are proposed by Taillandier et al. (2007) to describe snow metamorphism under temperature gradient (temperature driven~~

555  Comparison to existing physical models allows for the assessment of the additional influence of wind-speed, not considered previously (Flanner and Zender, 2006). However, we find that

560

 FZ06 most predicts the moderate-wind events with the lowest error. This is potentially due to the  initial conditions for low-wind event E10 likely corresponding to surface hoar, while the models from the literature tend to describe SSA decay  from precipitation. The initial SSA value of $46\,\mathrm{m^2\,kg^{-1}}$ for E10 is in agreement with documented SSA of surface hoar (Domine et al., 2009).

**4.1**

~~Conditions for the model are expected to be applicable over the Greenland Ice Sheet interior under mean summer conditions. The model predicts decay events at EastGRIP with a $r^2$ of 0.89, compared to observation, within defined conditions. SSA estimates from satellites have previously been compared to ground observations and show a strong correlation between daily mean SSA and satellite retrieved SSA at EastGRIP (Kokhanovsky et al., 2019). The SSA decay model has the potential to predict SSA decayGreenland Ice Sheet, the model can be evaluated for different sites to document the spatial variability in SSA over the entire ice sheet, and describe the summer SSA decay. This has additional benefits for quantification of surface mass balance and surface energy budget due to the relationship between snow microstructure and surface albedo.~~ following the methods in Kokhanovsky et al. (2019) would be an interesting future study, but is outside the scope of this manuscript.

**4.1  Inter-annual variability**

~~In this study, processes driving snow metamorphism are documented to influence isotopic composition of the snow after precipitation, supporting experimental observations and theoretical understanding (Ebner et al., 2017; Wahl et al., 2021; Hughes et al., 202 . Results from this study suggest that surface snow metamorphism following precipitation events corresponds to change in isotopic composition, most clearly observed in d-excess (Table 3)~~ The surface snow over the 90 m sampling transect is often non-homogeneous due to uneven distribution of accumulation. EOF analysis is used to account for spatial variability at each site, and to determine covariance between the parameters SSA, $\delta^{18}O$ and $d$-excess. The positive mode of $PC1_{SSA}$ is associated with depositional events, such as precipitation, surface hoar formation, and wind-fragmented snowdrift, causing an increase in SSA (Domine et al., 2009), while the negative mode is associated with snow metamorphism or wind scouring (Cabanes et al., 2002, 2003; Legagneux et al., 2003, 2004; Taillandier et al., 2007; Flanner and Zender, 2006). Based on  this interpretation, correlations between $PC1_{SSA}$ and $PC1_{d-excess}$ or $PC1_{\delta^{18}O}$ suggests the aforementioned mechanisms controlling SSA variability also influence the isotopic composition.

595

Accumulation intermittency and temperature conditions are proposed as a potential explanation for the change in regime from a coherence between $PC1_{\delta^{18}O}$ and $PC1_{d-excess}$ in 2018 and $PC1_{SSA}$ and $PC1_{d-excess}$ in $\delta^{18}O$ and decreases in d-excess

600 ~~kinetic fractionation is expected to influence the isotopic composition. A larger influence is expected for d-excess than $\delta^{18}O$ because kinetic fractionation influences $\delta D$ more than $\delta^{18}O$ ($d-excess = \delta D - 8 \cdot \delta^{18}O$) with a stronger influence on d-excess than $\delta^{18}O$, which can explain the covariance between d-excess and SSA observed most clearly during 2019 (Cappa et al., 2003; Da . Our approach to~~ 2019. Casado et al. (2021) show that during low precipitation periods in Antarctica, the isotopic signal is strongly modified during snow metamorphism. Approximately 10 cm of accumulation is recorded in both 2018 and 2019, but

605 a gradual increase during 2018 suggests multiple small deposition events, whereas 2019 is characterised by step-like increases. Therefore, the strong correlation between $PC1_{SSA}$ and $PC1_{d-excess}$ can be attributed to increased surface exposure and warmer temperatures facilitating snow metamorphism, in agreement with findings from Casado et al. (2021).

Low accumulation during 2017 presents a caveat to this interpretation, with results from 2017 showing $PC1_{d-excess}$ to be

610 influence by both $PC1_{SSA}$ and $PC1_{\delta^{18}O}$ during different periods. The period from May 15th to June 10th follows the regime observed during 2018 and corresponds to a negligible temperature gradient between the air, surface, and 10  cm

615

620  subsurface (Fig. A4). In contrast, the period from July 1st onwards is characterised by a near-constant upwards vapour flux, indicated by a negative temperature gradient between the

625  air, surface, and subsurface. $PC1_{d-excess}$ covaries with $PC1_{SSA}$ during this period, much like 2019, suggesting that vapour diffusion driven by temperature gradients modifies the

snow isotopic composition. This agrees with previous studies documenting kinetic effects during snow grain growth resulting from pore space diffusion (Neumann and Waddington, 2004; Casado et al., 2016; Ebner et al., 2017; Casado et al., 2021).

An additional feature supporting the observation of processes driving surface snow metamorphism corresponds to a decrease in d-excess, is a clear relationship between substantial increases in SSA and increase in d-excess (Fig. ??) . The upper 10th percentile of $\Delta$SSA increases $(14.7\,\mathrm{m^2\,kg^{-1}})$ corresponds to positive $\Delta$d-excess in 70% of cases (Fig. 7). Large increases in SSA are closely associated with precipitation, however, increases are observed in The opposing phases of the North Atlantic Oscillation (NAO) between the years can explain the different meteorological conditions. The NAO is in a number of other scenarios (Domine et al., 2009). Precipitation is expected to cause the largest SSA, suggesting that the d-excess of precipitation is most often higher than existing surface snow. Our results therefore suggest that the precipitation isotopic composition signal is not always preserved after snow metamorphism due to (kinetic) fractionationduring sublimation and other surface processes. positive phase during 2018 and the majority of 2017 bringing below-average temperatures, as observed at EastGRIP (Hanna et al., 2015). The opposite is observed during 2019, corresponding to a positive phase in the NAO.

*Change in d-excess per day ($\Delta$d-excess $day^{-1}$) vs. change in SSA per day ($\Delta SSA day^{-1}$)*The relationship between the rate of change in SSA per day ($\Delta SSA day^{-1}$) and d-excess ($\Delta$d-excess$day^{-1}$) for all summer seasons 2017-2019 (light grey), all events (dark grey) and selected events based on substantial accumulation (dark turquoise). The box indicates the values corresponding to daily decrease in d-excess during decrease in SSA, with 81% of selected events in this quadrant. Conclusive results from EOF analysis are limited by wind-effects, especially in the negative phase, corresponding to decrease in SSA, where wind scouring potentially removes the surface layer (Domine et al., 2009; Flanner and Zender, 2006; Hachikubo et al., 2014) . Decoupling snow metamorphism from wind scouring is considered in the following section on isotopic change during low-wind SSA decay events.

**4.2 Influence of event conditions on isotopic change**

Surface conditions prior to and

**4.2 Isotopic change during SSA decay events**

Three key mechanisms are expected to drive the rapid SSA decays, 1) large grains growing at the expense of small grains (Legagneux et al., 2004; Flanner and Zender, 2006), 2) diffusion of interstitial water vapour (Colbeck, 1983; Ebner et al., 2017; Touzeau et , 3) sublimation due to the wind ventilating the saturated pore air, known as 'wind-pumping' (Neumann and Waddington, 2004; Town et al., . The dominant mechanisms can theoretically be identified by a combination of the change in isotopic composition - indicating the fractionation effect - and the LE and temperature gradient data.

In theory, mechanism 1) causes minimal change in the bulk isotopic composition of a snow layer under isothermal conditions (Ebner et al., 2017). Therefore, observations of SSA decay corresponding to negligible isotopic composition change could be explained by this mechanism. We observe no events with consistent isotopic composition throughout. In the instance of 2) interstitial diffusion, light isotopes are preferentially diffused, while the heavy isotopes will be preferentially deposited onto the cold snow grains (Colbeck, 1983; Ebner et al., 2017; Touzeau et al., 2016). Thus, diffusion of water vapour in the

660    pore space causes a decrease in $d$-excess and slight increases in $\delta18O$ due to kinetic fractionation (Flanner and Zender, 2006) . 3) Sublimation has been widely documented to cause an increase in $\delta^{18}O$ of the remaining snow-mass due to equilibrium fractionation, and a significant decrease in $d$-excess due to kinetic fractionation (Ritter et al., 2016; Madsen et al., 2019; Hughes et al., 2021 ).

     An overall increase in $\delta18O$ and decrease in $d$-excess during E10 can be attributed to a combination of 2) and 3) based on
665    observation of net-sublimation and high amplitude diurnal temperature gradient variability indicating vapour transport within the pore space. The period between 9th June at 15:18 UTC and 10th June 10:40 UTC recorded net deposition corresponding to an overall decrease in $\delta18O$ during
670     the first day  and minimal decrease in $d$-excess, potentially due a deposition of atmospheric water vapour (Stenni et al., 2016; Feher et al., 2021; Casado et al., 2021) .

     A 30% decrease in $d$-excess corresponds to a  negligible change in $\delta18O$ during
675    E11. Net-sublimation double that of E10 is measured, but with reduced amplitude in both TGs. Moreover, the largest decrease in $d$-excess occurs after the first day when the surface-subsurface TG is consistently negative, indicating that vapour diffusion plays a role in modifying the isotopic composition, and the effect of equilibrium fractionation during sublimation from the surface only weakly influences the bulk isotopic composition over the 3-day period (Casado et al., 2021). Decoupling the influence of atmosphere-surface exchange and diffusion from subsurface snow requires additional measurements of isotopic
680    composition of atmospheric water vapour and precipitation isotopes, which is outside the scope of this study.

**4.3**

685    ~~for spatial variability at each site, and a coherence is observed between the principal components of d-excess and SSA. PC1 is weaker when spatial variability is high, and during these periods the coherence between d-excess and SSA are muted. During the start of 2017 and 2018 PC1 of d-excess is coherent with PC1 of $\delta^{18}O$, and decoupled from PC1 of SSA. At the start of the season, the 2.5 cm sample will contain winter snow layers which are less influenced by snow metamorphism (Libois et al., 2015; Town et al., 2008), and thus, a coherent signal between d-excess and $\delta^{18}O$ is observed. The transition~~
690

**4.3**

**4.3 Implications and perspectives**

695  Our results suggest that processes driving snow metamorphism modify the isotopic composition

700  the snow while exposed at the surface, supporting experimental observations and theoretical understanding (Ebner et al., 2017; Wahl et al., 2021; Hughes et al., 2021). We find that $d$-excess is mostly influenced by vapour fluxes in the pore space, driven by temperature gradients. Net-sublimation appeared to have less influence on the isotopic composition, but this is expected to be  due to the depth of the sample and the short duration of both low-wind events.

705

~~Seasonal signals are influenced by millennial scale insolation variability (Masson-Delmotte et al., 2006; Laepple et al., 2011). An inverse relationship is observed between obliquity and d-excess over the past 250 ka years at Vostok which is attributed to the insolation gradient between high and low latitudes causing increases moisture transport from low latitudes relative to high latitudes (Vimeux et al., 2001, 1999). Results presented in our study document decreases in snow d-excess during surface~~

710

715  The findings of this exploratory study reiterates the importance of quantifying the isotopic fractionation effects associated with processes driving snow metamorphism during precipitation free periods. Moreover, the inter-annual variability observed at EastGRIP between 2018 and 2019 suggests that precipitation intermittency and temperature (gradients) play a role in isotopic change, which is not so readily identified in the surface snow SSA data. Future work to decouple the processes driving change in  $d$-excess (sublimation from surface or interstitial vapour diffusion in the pore space) is vital for modelling the

720 change in isotopic composition down to the close-off depth in the firn (Touzeau et al., 2018; Neumann and Waddington, 2004).  Future studies would benefit from obtaining direct measurements of the isotopic composition  of precipitation and surface hoar, to determine the fraction of  such deposits in the SSA samples. Furthermore, a quantitative representation of vapour fluxes in the surface snow would provide a basis from which to quantify the the relative influence of fractionation during sublimation and interstitial diffusion.

**5 Conclusions**

This study addresses the rapid SSA decay driven by surface snow metamorphism. In particular, the study aims to explore how rapid SSA decay relates to changes in isotopic composition of the surface snow in the dry accumulation zone of the Greenland Ice Sheet. Ten individual snow samples were collected on a daily basis at EastGRIP in the period between May and August of 2017, 2018 and 2019. SSA and isotopic composition was measured for each sample. Periods of snow metamorphism after  deposition events are defined using SSA measurements to extract periods of rapid decreases in SSA.

An exponential SSA decay model ($SSA(t) = (SSA_0 - 26.8)e^{-0.54t} + 26.8$ $SSA(t) = (SSA_0 - C)e^{-\alpha \cdot t} + C$) was constructed to describe surface snow metamorphism under mean summer conditions for polar snow, with surface temperatures above -30 °C  Two categories were defined to assess the influence of wind-speed on the SSA decay rate. The relationship between defined events of snow metamorphism and corresponding snow isotopic composition was then explored.

 Changes in isotopic composition corresponding to post-depositional processes driving  SSA decay is observed in all events. Over the first  2-days of SSA decay events,  $d$-excess is observed to decrease  from the initial . Analysis of SSA decay events with consistent low wind speed indicates that the combined effects of vapour diffusion and diurnal LE variability causes isotopic fractionation of the surface snow in the absence of precipitation.

[revised manuscript text omitted]

840 Hachikubo, A., Yamaguchi, S., Arakawa, H., Tanikawa, T., Hori, M., Sugiura, K., Matoba, S., Niwano, M., Kuchiki, K., and Aoki, T.: Effects of temperature and grain type on time variation of snow specific surface area, Bulletin of Glaciological Research, 32, 47–53, https://doi.org/10.5331/bgr.32.47, 2014.

Hanna, E., Cropper, T. E., Jones, P. D., Scaife, A. A., and Allan, R.: Recent seasonal asymmetric changes in the NAO (a marked summer decline and increased winter variability) and associated changes in the AO and Greenland Blocking Index, International Journal of
845 Climatology, 35, 2540–2554, https://doi.org/10.1002/joc.4157, 2015.

[revised manuscript text omitted]

---

## Author Comment (AC2)

**Response to Review 2:**

We thank the reviewer for their helpful comments and their insight on this subject. The authors acknowledge the constructive criticism and comments from the reviewer and propose the following revisions. We appreciate the comments by the reviewer which have resulted in a significantly strengthened manuscript.

The original comments from the reviewer are in black and in blue are the author's responses, with blue italics to show the in-text changes. The authors want to point out that due to many useful suggestions, the major revisions have been implemented resulting in significant changes to the manuscript, as can be seen in the document attached below.

**Overview**

This paper investigates the relationship between changes in snow specific surface area (SSA) and its isotopic composition, focused on d-excess, at EastGRIP. The Authors focus on precipitation events, after which rapid SSA decays are observed, coupled to a decrease in d-excess. The Authors propose an exponential rate law for SSA decay, which is temperature independent between 0 and -25°C. The Authors then discuss the interplay between snow metamorphism and d-excess, and the possible impact of their findings on the interpretation of the ice core isotopic record.

**General comments**

The idea underlying this research is very nice: snow metamorphism results in sublimation-condensation cycles which should lead to isotopic fractionation. SSA decay is taken as a proxy for the intensity of metamorphism, and the expected correlation between SSA decay and isotopic fractionation is found, and is readily visible in d-excess. Such a study is clearly relevant to the interpretation of the ice core isotopic record and the data presented therefore deserves attention.

However, my opinion is that the experimental protocol is partly flawed, and this unfortunately casts doubt on the validity of the data obtained and on the conclusions derived. The first point is that SSA is measured on a 1 cm thick layer while isotopes are measured on a 2.5 cm thick layer. Furthermore, no detailed observations of surface snow are mentioned to ensure that the thicker 2.5 cm sample was the same snow layer as the top 1 cm snow. In many cases, the authors may then be measuring 2 little-related snow samples, which would in fact completely invalidate their study.

The reviewer addressed a major concern related to the sampling protocol for SSA and isotopes. The authors primarily want to clarify that the isotopic composition was *directly* measured from each SSA sample. Thus, each SSA sample has a corresponding isotopic composition. The offset we refer to comes from the measurement resolution of SSA due to the e-folding depth of 1310 nm radiation in high density snow. We apologise that this was not made clear enough in the original manuscript, and we hope that this clarification gives the reviewer increased confidence in the sampling protocol. We add the following text to the manuscript:

*"Individual SSA samples were put in separate bags and subsequently sampled for water isotopic composition. Thus, each day the 10 SSA samples have a corresponding isotopic composition."*

Many processes can affect the very surface snow layer. These include fog deposition, the formation of surface hoar or sublimation crystals, and wind drifting. All this is hardly mentioned, so that I am not even sure that adequate observations were systematically made. These are absolutely necessary for any careful snow physics investigation. If a 0.5 cm-thick fog deposit or surface hoar formation takes place, then clearly the SSA value will mostly reflect this deposit while the isotopic measurement will mostly characterise the underlying snow layer. Relating both measurements will then be totally meaningless. It is clear to me that the authors should have sampled only the top layer for isotopic measurements. If not enough material was present in their ICE CUBE sample holder, then they should simply collect more surface sample nearby.

The reviewer here addresses important comments related to other relevant processes for surface snow. Daily observations were recorded for snowfall, snowdrift, and ground fog, although there was no consistent documentation of surface hoar/sublimation crystal surface features. There is no doubt that fog deposition and surface hoar etc are processes that are important for SSA studies as documented by Domine et al., 2009; Gallet et al., 2014; Fergyresy et al., 2018. However, the observed SSA value for surface hoar is ~54 $m^2$ $kg^{-1}$ (Domine et al., 2009), is similar to the values on the initial day of our events. We would therefore most often expect an increase in SSA in the instance of surface hoar and snow drift (Kuhn et al., 1977; Grenfell et al., 1994; Domine et al., 2009; Libois et al., 2014).

We add the following text to highlight the importance of addressing potential of such surface features. In the methods, to clarify the potential that SSA increase is the result of precipitation, snowdrift or surface hoar:

*"We here use the term deposition events to describe rapid increases in SSA, expected to be from precipitation, drifted snow or hoar formation. Previous studies have indicated that surface hoar and sublimation crystal-like grain growth features at the surface have an SSA value around 54 $m^2$ $kg^{-1}$, based on the SSA of hoar frost (Domine et al., 2009)."*

In the discussion:

*"However, we consider potential increases in SSA in the absence of precipitation under the following conditions: 1) surface hoar formation on an aged snow surface (SSA < 50 $m^2$ $kg^{-1}$), 2) the effective sieving of small, fragmented grains into the pore space via wind, and 3) from sublimation and subsequent fragmentation of snow grains while suspended by the wind (Domine et al., 2009). Selecting only rapid decreases in SSA reduces the probability of capturing these processes in our analysis."*

Regarding the depth of the sample, we add that each sample had 2.5 cm of snow. However, we can only say for certain that the top 1 cm of each sample was measured given the e-folding depth (now edited in manuscript from light penetration depth) for 200 kg m$^{-3}$ is 1 cm, which is lower than the mean density for EastGRIP surface snow. We state this as a limitation.

Wind drifting is another important process, which is not detailed. The threshold of 6 m/s for the mean daily wind speed is simply not adequate. Hourly values must be considered, and in fact ideally maximum, not average values, are most useful to evaluate wind speed effect on drifting. But the best data on this aspect is observations. Wind drifting can easily be detected by observations. I appreciate that such observations cannot be done 24 hours a day, but the consequences of wind drifting are easily observable by looking at changes in the snow scene.

We agree with hindsight that this threshold is insufficient to reduce the likelihood of surface perturbation, and to address this we now use the 10-minute data from PROMICE. It is important to note here that 209 out of the total 237 sampling days have daily maximum wind speed exceeding 5 m s$^{-1}$ and no events had wind-speed consistently below 5 m s$^{-1}$ (two had 5.1 m s$^{-1}$). In addition, snowdrift events were documented in the EastGRIP field diary and correspond to wind-speeds above 7 m s$^{-1}$. Several events have maximum wind-speed between 6- 7 m s$^{-1}$, and no snowdrift documented. Based on this analysis and observations from the literature, we define two wind categories, as briefly suggested by the reviewer in a later comment, we have added a secondary wind-speed category for comparison of SSA decay when wind-speed is <6 m s$^{-1}$ (low-wind events), and when maximum wind-speed is between 6- 7 m s$^{-1}$ (moderate-wind events). The following text is added to the document:

"A set of criteria are required to reduce the potential of analysing events with wind-perturbed surfaces, resulting in the removal of surface snow. In Antarctica, unconsolidated surface snow has been observed to drift at wind speeds as low as 5 m s−1 measured at 2 m height (Birnbaum et al., 2010). However, a study from Greenland documented snowdrift starting at 6 m s−1 (Christiansen, 2001), likely due to warmer temperatures allowing for the surface snow to become more bonded (Li and Pomeroy, 1997). At EastGRIP, calm conditions correspond to wind speeds from 0 −5.2 m s−1 according to field diary observations. The mean daily maximum wind speed for the three sampling seasons was 6.8 m s−1, while blowing snow was documented only when wind speeds exceeded 7 m s−1.Based on this assessment, we define two wind-speed categories for comparison of the effects of wind-speed on SSA decrease. The first includes events with wind-speed consistently below 5.2 m s−1, hereafter referred to as low-wind events, to ensure no surface perturbation. Secondly, we consider events where the maximum wind-speed is between 6 −7 m s−1, hereafter referred to as the moderate-wind events. The inclusion moderate-wind events allow an assessment of the influence of wind-speed on SSA decrease."

Out of the 21 initially defined events, only 2 are below the wind-speed threshold with maximum values of 5.1 m s$^{-1}$ in both events. We expect negligible snowdrift for these two events allowing us to confidently argue that the surface is unperturbed and isotopic change is the result of snow metamorphism. The likelihood of drifting snow during moderate-wind events is considered using the equation defined from Li and Pomeroy (1998), where the threshold wind-speed for snowdrift is defined as a function of temperature.

Following the same structure as in the original manuscript, we construct the SSA decay model with parameter values set for the two wind-regimes. We add the revised figure to this response. Intuitively, the SSA decay rate is higher for moderate-wind events (-0.53 m$^2$ kg$^{-1}$ day$^{-1}$) compared to low-wind events (-0.41 m$^2$ kg$^{-1}$ day$^{-1}$). As the reviewer will see later in this response, we add the results from the comparison of our data and SSA decay model to

existing models from Flanner and Zender (2006) and Taillandier et al. (2007).

Drifting can remove newly precipitated snow or accumulate it some places. This must be recorded when sampling. It is fairly easy to recognize snow layers from careful observations. All these mandatory observations do not appear to have been done.

I very strongly recommend that the authors detail whatever observations were done and clearly say what has not been done. In their analysis, they should only keep data for which they are certain that SSA and isotopic measurements were on the same layer. All data with surface hoar, fog or sublimation crystals should be eliminated. Drifting events resulting in non-homogeneous layers that were sampled must likewise be eliminated. If there are not sufficient observations to sort the data, then I fear the study may be invalid.

We refer back to our previous response regarding the documentation of snowfall, snowdrift, and ground fog. We remove the events with snowfall and wind drifted snow and Table A with event overview is kept in the Appendix. In addition, we add the following text to the Methods section "Defining SSA decay events".

*"We here use the term deposition events to describe rapid increases in SSA, expected to be from precipitation, drifted snow or hoar formation. Previous studies have indicated that surface hoar and sublimation crystal-like grain growth features at the surface have an SSA value around 54 $m^2$ $kg^{-1}day^{-1}$, based on the SSA of hoar frost (Domine et al., 2009). If snowfall/snowdrift/ground fog was documented during the SSA decay, this event is removed from analysis due to perturbation of the surface layer."*

We wish to highlight that the one of the low-wind events was preceded by ground fog, not snowfall. We see value in including these events given that we have ensured negligible wind-perturbation during the event. It is interesting to compare the isotopic change during these two events. We now explicitly include this in the results section "3.2 SSA decay events":

*"Both E10 and E11 had consistent clear sky conditions. We note here that E11 was preceded by significant ground fog, not snowfall, indicating that the peak value of 46 $m^2$ $kg^{-1}$ was likely the result of surface hoar, and thus, rapid SSA decay follows an SSA peak not caused by precipitation."*

To further accommodate this comment, we present the latent heat flux and temperature gradient data from the two low-wind events, and extend the discussion of isotopic change with regard to the near surface fluxes. "4.3 Rapid SSA decay and isotopic composition". Here we state that a lack of consistent observation of surface hoar in the SSA samples as a limitation to the study, but we take every precaution to ensure we are analysing unperturbed surface snow.

The organization of the paper must also be modified. Data appear in the discussion. All results should be reported in the results section and extra figures showing wind speed and snow surface conditions must be drafted.

The structure of the paper has been modified to address this comment. Most restructuring is applied to the results, and the discussion then follows suit. Meteorological conditions are

presented in the first section of the results, highlighting the inter-annual variability in temperature, accumulation, and latent heat flux between the sampling years. A description of SSA and isotopic composition is then presented alongside the EOF analysis, before focussing on the SSA decay events. Having outlined the modified event criteria in the methods, the suitable events are defined, and the decay model is presented with parameter values best fit to the two wind-regimes. Comparison to physical based models from the literature is included in the model evaluation. The final results section on isotopic change primarily considers events from both the low- and moderate-wind regimes, before focussing in detail on the two low-wind events. The latent heat flux and temperature gradients are assessed to infer processes driving isotopic change. The revised results facilitate a more concise discussion.

Regarding the SSA decay rate law, I am not sure this is the best formula. Since sublimation is thermally activated, the absence of a temperature effect is strange. Perhaps when data is sorted, such an effect will appear. The authors quote (Cabanes et al., 2003) to support their choice of analytical expression, but those Authors had a temperature-dependent rate law. Furthermore, subsequent studies on SSA decay rate laws proposed other analytical expressions, and their exploration should be discussed when the rate law is investigated, not line 309 in the discussion.

We appreciate that the reviewer has pointed this out. The temperature-dependence is now stated from Cabanes et al. (2003) in the introduction, along with the subsequent models proposed by Legagneux et al. (2004), Flanner and Zender (2006) and Taillandier et al. (2007). We have added the following text to the introduction:

*"Previous studies have proposed SSA decay models using a combination of field measurements and controlled laboratory experiments (Cabanes et al., 2002, 2003; Legagneux et al., 2003, 2004; Flanner and Zender, 2006; Taillandier et al., 2007). Exponential models to describe SSA decay are documented to be the best fitting to in-situ data from Arctic Canada (Cabanes et al., 2003). However, the lack of physical basis led Legagneux et al. (2003) to construct a new equation based on laboratory experiments to describe a temperature dependent SSA decay."*

The rarity of consistent low-wind conditions limits all in-situ studies regarding the duration of SSA decay events. However, we feel the documentation of SSA decay at the surface is valid and useful for planning of future campaigns, where more detailed observations would be beneficial, and for remote sensing studies.

In summary, this potentially interesting study may be partially of totally invalidated by an inadequate experimental protocol, at least based on the information supplied in the paper. If the authors have made observations not reported in this version, they should report all relevant information in a revised version. I then recommend sorting the data and removing all data where there is a reasonable suspicion that SSA and isotopic measurements were not on the same snow layer. I also strongly recommend a more logical organization of the paper. The discussion is often unfounded speculation and must be considerably shortened. I propose below numerous specific comments that I hope will be useful to the Authors in preparing an extensively revised version, for which I recommend a second round of review. These

comments were written before the general evaluation, so there is some repetition. And finally, I kindly request that *all* Authors involved in this work make a careful reading of the revised version. This does not seem to have been done for the version I read, which is not very respectful for the reviewers.

We are grateful for the time and effort taken by the reviewer to comment on this manuscript. The edited manuscript follows a more logical format and the edits made based on the reviewer's comments have improved the quality of the study. We apologise for mistakes in the original manuscript, we will ensure the revised document is carefully checked for errors.

**Specific revisions required**:

Line 35. Spell out SSA=specific surface area, which is the surface are of the ice-air interface per unit mass of snow, expressed in m2 kg-1. It is not assumed to be linked to the optical grain size dopt, as mentioned by the Authors, it is rigorously and simply linked by a geometric relationship SSA=6/ρice dopt, as shown in equation (1) of (Gallet et al., 2009), which is probably a more relevant reference than Linow 2012. In fact this relationship was already implicitly mentioned by (Grenfell and Warren, 1999), although they did not use the term specific surface area.

We apologise for missing this, we have changed this to:

*"The snow-air interface can be described by the widely used parameter snow specific surface area (SSA), where the SSA of a snow sample is dependent on optical grain radius and density of ice (SSA = 6 / rho$_{ice}$*d$_{opt}$) (Gallet et al., 2009), and can be utilised as a measure for snow metamorphism (Cabanes et al., 2002, 2003; Legagneux et al., 2002)."*

Lines 41-43. The reasons for SSA decrease (of dry snow) are not explained well and even erroneously. Wind fragmentation in fact increases SSA since smaller crystals are formed (Domine et al., 2009). Sublimation does not necessarily lead to SSA decrease as it reduces crystal size; and likewise vapor diffusion does not necessarily lead to SSA decrease. What actually leads to SSA decrease is the disappearance of small structures, often by sublimation, and the growth of larger crystals, often but not only by vapor diffusion in the pore space.

We appreciate the reviewer's insight here and have made changes to the text to correct this mistake.

Introduction: *"Freshly deposited snow has a high SSA which decreases with time under both isothermal (<10 °C m−1) and temperature gradient (>10 °C m−1) conditions (Cabanes et al., 2002; Legagneux et al., 2004; Domine et al., 2007; Genthon et al., 2017). Decrease in SSA is predominantly the result of Ostwald Ripening, where large grains grow at the cost of smaller grains (Lifshitz and Slyozov,1961; Legegneux et al., 2004), vapour diffusion in the pore space driven by sublimation and deposition (Flin and Brzoska, 2008; Sokratov and Golubev, 2009; Pinzer et al., 2012), and wind effects (Picard et al., 2019). Under natural conditions SSA decrease is driven by a combination of these processes depending on surface conditions (Cabanes et al., 2003; Pinzer and Schneebeli, 2009a), each potentially modifying the isotopic composition of the snow (Ebner et al., 2017)."*

An additional sentence or two are proposed for the discussion to explain the influence of wind, specifically relating to the results from EOF analysis to mention the potential for SSA increase due to sieving of fragmented grains (Domine et al., 2009), and wind-pumping potentially reducing SSA via sublimation (Town et al., 2008). This is particularly of interest when we observe the covariance between the SSA and isotopic parameters, given that some increases in SSA could be due to this effect, and the corresponding isotopic change would be the result of fractionation and not from precipitation or wind-blown snow. To account for the ambiguity, we focus on decreases in SSA where grain growth is likely happening and refer to latent heat fluxes and temperature gradients when assessing isotopic change.

Line 47. It is erroneous to state that "While current versions of the so-called decay models exist, these are mostly based on lab-experiments and non-polar snow observations". The works of Cabanes and Taillandier are mostly based on Arctic and subarctic observations. Granted, none of these studies used data obtained on ice sheets, and this could be mentioned, if there are reasons to believe that ice sheet processes involved in SSA decrease are in general different from those on seasonal Arctic snowpacks. By the way, (Carmagnola et al., 2014) tested various SSA decay models against data from Summit, Greenland, and this may me relevant to the authors' topic.

We thank the reviewer for pointing this out and have corrected this error. The paper Carmagnola et al. (2014) is a useful reference for the comparison to models. Like Linow et al. (2012), they look at the snow properties over a vertical profile as opposed to looking at the temporal evolution of the exposed surface snow. We therefore maintain that our continuous SSA data from EastGRIP is a valid approach to quantify the in-situ SSA decay under natural conditions. Even in the case of elevated wind-speed, we believe it is useful to document how the surface SSA is influenced with regard to remote sensing, as the reviewer also pointed out. The analysis for remote sensing was outside the scope for this paper unfortunately.

The following edit is proposed to acknowledge the previous SSA studies for polar snow, and highlight that we are referring here to SSA studies in the accumulation area of ice sheets:

"While continuous surface SSA measurements exist from Antarctica (Gallet et al., 2011; Gallet et al., 2014; Picard et al., 2014), those from Greenland focus on the depth evolution of SSA (Linow et al., 2012; Carmagnola et al., 2013). A continuous dataset of daily SSA and corresponding isotopic composition measurements from the accumulation zone of the Greenland Ice Sheet can contribute to understanding the relevance of snow metamorphism for surface energy budget and for ice core studies."

Line 78. What is meant by surface temperature? Is this the skin temperature measured by IR emission? Or is it the air temperature near the surface? Mentioning a reference is not sufficient. A paper must be self -standing and must not require looking up references for understanding, especially for such a central variable. If this is skin temperature, all relevant details must be given here, including the instrument used, the wavelength range and the emissivity value used. Furthermore, validation of the skin temperature measurements would be desirable. IR sensors require very careful calibration to be accurate.

We apologise for the oversight here and have added instrument specifics to Table 1. The surface temperature is calculated from upwards and downwards longwave radiation with long wave emissivity set to 0.97 and is added to the text.

*"Surface temperature from PROMICE is calculated from upwards and downwards long-wave radiation (measured using Kipp & Zonen CNR4 radiometer) with long-wave emissivity set to 0.97."*

Line 85 ff. Sampling procedure. It is essential to note when there is a change in the snow layer sampled, i.e. when there was wind drift or precipitation. I guess precipitation events were readily identified, but what about wind drift? Did the authors note when the layer being sampled changed because of wind erosion of wind accumulation? This is critical for data interpretation.

Wind drift was documented in the field diary as well as snowfall and ground fog. However, detailed observations of surface features were not measured consistently over the 3 sampling years. High spatial variability in SSA and accumulation gives us an indication of a heterogeneous surface. Moreover, we consider each sample site individually to avoid attenuation of signals by using the mean. The field observation protocol is added in the methods, and a description of the surface conditions has been added in the results.

Line 100. "Light penetration depth in snow of 200 kg m−3 is approximately 1 cm". Light penetration does not just depend on density, but also on SSA. Thus for 200 kg m-3, a penetration depth of 1 cm corresponds to a precise SSA value. Furthermore, penetration depth is not very meaningful. Do the authors mean e-folding depth? Note that if the e-folding depth is 1 cm, still 27% of the reflected light intensity will be due to depths >1 cm. Also did the authors make detailed observations of detailed surface processes such as surface hoar, sublimation crystals or rime events (these are frequent at Summit, perhaps also at EastGrip)? This is important because these thin surface deposits will greatly impact measured SSA, while they will be diluted in isotopic measurements. To evaluate penetration depth and the impact of surface deposits on SSA measurements, the Authors can use the TARTES model. https://snow.univ-grenoble- alpes.fr/snowtartes/ . This will allow them to make valid quantitative statements, and to explore the impact of surface deposits on measured SSA.

We appreciate the reviewer's insight here and clarify that we are referring to e-folding depth which has been corrected in the manuscript. As previously mentioned, significant fog, snow drift and snowfall were documented in the field diary. However, no consistent detailed observations of surface features such as surface hoar/rime/sublimation crystals were made. We propose to use the eddy-covariance LE measurements to identify the potential of these deposits during the low-wind events used to observe concurrent isotopic change. The reviewer mentions the TARTES model which is a valuable tool. Nonetheless we are constrained by the nature of our Greenland surface observations and unfortunately this limits us from getting accurate additional information. For future work, TARTES is surely very useful. The following text has been updated in the methods:

*"The e-folding depth of 1310 nm radiation in snow of 200 kg m$^{-3}$ is approximately 1 cm (Gallet et al., 2009). At EastGRIP, the mean snow density from 2017, 2018 and 2019 is 293 kg m$^{-3}$*

*resulting in each measurement being heavily weighted to the top <1 cm of the 2.5 cm sample (307 ± 40 kg m$^{-3}$, 278 ± 47 kg m$^{-3}$ 294 ± 50 kg m$^{-3}$ for 2017, 2018 and 2019 respectively)."*

Line 117-118. It is strange the Authors did not sample the top 1 cm for isotopic measurements, to ensure better correspondence with the SSA measurements.

In our responses above we have clarified the sampling procedure in greater detail than provided in the original manuscript. To ensure that the isotopes correspond to the SSA samples directly, this procedure was preferentially used, instead of taking two separate snow samples for SSA and isotopes. We hope these answers fulfil the reviewers request on these matters.

Table 1. Usually Table captions are concise and explanation are in footnotes. Most of the caption is in fact unnecessary and can be deleted.

All the Table captions have been edited to be more concise.

Lines 132-133. Eq. (1) was indeed proposed by Cabanes et al. as the most empirically accurate, but this was just to fit their limited data set. Legagneux (2005) proposed a theoretically correct equation (his Req. 2). That equation was also used by (Flanner and Zender, 2006). Taillandier et al. (2007) used an approximation of that equation to fit experimental data and their equation has a log form. I believe the expression of Taillandier is more suitable. From the discussion, the Authors tested it, but this should be detailed here, not in the discussion.

We acknowledge the usefulness in presenting the results of our inter-model comparison and have added the results in Section 3.3 Model Evaluation. Prior to this, the following text has been added to the methods section '2.2.1 Modelling SSA decay" to accommodate this suggestion:

*"The first empirical SSA decay model was proposed by Cabanes et al. (2003) who described a temperature-dependent exponential decay based on snow samples collected from the Alps (Cabanes et al., 2002) and Arctic Canada (Cabanes et al., 2003). A following logarithmic equation (Eq. log) fit controlled to laboratory experiments was proposed by Legegneux et al. (2004), where parameters A and B are arbitrarily related to the decay rate and initial SSA of each sample and are linearly correlated at -15°C. To improve the physical basis of the model, the theory of Ostwald Ripening, describing grain growth driven by a physical need to reduce surface energy, was implemented into the model (Legagneux et al., 2005). The equation (Eq. 4) has two parameters τ and n; T is the decay rate and n relates to the grain growth. The physical model was developed by Flanner and Zender (2006) to incorporate more specific physical quantification to the parameters to include information about temperature, temperature gradient and density. Based on these three conditions, they created a look-up table for τ and n.*

*Taillandier et al. (2007) proposed two equations based on the logarithmic model first proposed by Legagneux et al. (2004) to define the decay rate under isothermal and temperature gradient conditions where they were able to directly incorporate a surface temperature*

*parameter.*

*An empirical decay model is constructed upon previous studies (Cabanes et al., 2002, 2003; Flanner and Zender, 2006; Legagneux et al., 2002, 2003; Taillandier et al., 2007). This model uses continuous daily SSA measurements from EastGRIP to describe the behaviour of surface snow SSA in polar summer conditions. All samples of defined SSA decay events are used to quantify surface snow metamorphism."*

Lines 162-164. Ground temperatures are not very relevant to the explanation of crystal shapes, as these form in clouds at a different temperature. And by the way Domine et al. (2008) is not the most suitable reference for this. I recommend (Kuroda and Lacmann, 1982) and references therein.

This is a valid point which we overlooked; however, this explanation has ultimately been removed from the revised manuscript given that the SSA decay during the single 'cold' event is likely to have been influenced by snowdrift. The reference to Kuroda and Lacmann (1982) is appreciated for general understanding, and we apologise for the inaccurate referencing here.

Line 165. The upper threshold for wind speed used here is a daily mean value of 6 m s-1. When the daily mean value is 6 m s-1, It is very likely that gust speeds were much higher and that wind drifting took place, with major modifications in SSA. Perhaps transport even brought other layers. I think combining events with and without snow drift is not adequate to derive SSA decay rate laws. At the minimum, events with and without drifting should be treated separately to investigate wind effects. Regarding isotopes, the sampling of blowing snow would have been interesting. Was that performed?

To address this comment, we refer back to our response to an early comment in the 'General comments' section. As we previously noted, the SSA decay rate for moderate-wind events (max. wind-speed 6- 7 m s$^{-1}$) is substantially higher than for low-wind events (< 6 m s$^{-1}$). Here, we make use of the physical based model from Flanner and Zender (2006) and Taillandier et al. (2007) by comparing their predictions to those of our data and empirical model. These comparisons are presented in the results section "Model evaluation", and then discussed in the section "SSA decay at EastGRIP". Unfortunately, there was no sampling of blowing snow, but we mention this, as well as sampling of surface hoar, as a suggestion for future studies.

We add an additional figure (Figure A2) to show the results of the model comparison for the two low-wind events (E10 and E11), and for examples of moderate-wind events (E2 from 2017 and E18 from 2019). The two moderate-wind events have maximum 3 m wind-speeds of 6.26 m s$^{-1}$ and 6.28 m s$^{-1}$. Based on the drift threshold defined in Li and Pomeroy (1998), E2 has potential influence from snowdrift, but not E18 (U(10) = 7.09 m s$^{-1}$ and 8.17 m s$^{-1}$ for E2 and E18 respectively), which agrees with an underestimation of decrease from FZ06 compared to observation during E2. Interestingly, we get the lowest RMSE values for FZ06 and the moderate wind events. Possible explanations include the initial snow conditions and event duration, which are included in the discussion.

Line 178. What is the RMSE? This is mentioned line 194 but would be better mentioned here

in context.

Yes of course, this has now been added earlier in the text for all models used. Based on the revised analysis, the RMSE based on low-wind events is 3.64 m$^2$ kg$^{-1}$ for the exponential model from this study, 3.45 m$^2$ kg$^{-1}$ for FZ06 and 6.34 m$^2$ kg$^{-1}$ for T07 based on the individual sample sites. For the moderate-wind events the RMSE is actually smaller, the values are 2.48 m$^2$ kg$^{-1}$, 1.28 m$^2$ kg$^{-1}$ and 5.63 m$^2$ kg$^{-1}$ for this study, FZ06 and T07 respectively.

Line 190. The authors indicate intermittent snowfall during day 2 of E14. Why did they not remove this presumably thin new layer to avoid this artefact? The thin layer greatly affected the SSA measurement but probably had little impact on the 2.5 cm-thick isotope sample.

Events with intermittent snowfall/snowdrift/ground fog are now removed from further analysis. Removing surface artefacts would likely result in a degree of compaction in the sampling holder, and therefore to avoid any disturbance to the samples, they were handled as little as possible.

Line 197. Why is not an equation proposed and tested for the lower temperatures?

During our sampling period, there was only one event with mean temperatures below -30°C. As previously mentioned, the wind-speed during this event is higher than the threshold. During the initial analysis, we grouped the events by temperature ranges, however, we did not observe a clear temperature dependence of the decay rate. After the removal of events likely to have surface perturbations, we observe a single event in the moderate-wind category which is poorly predicted by the equation for the wind-speed category. As the additional text below explains, this event had the lowest mean air temperatures and thus we do observe the expected temperature dependence.

*"Event 9 in 2018 is poorly represented by the moderate-wind SSA decay model from this study. The mean air temperature for this event was -20.8°C, 5°C less than the next coldest (E11 at -15.3°C). Fitting the model for E9 alone gives a decay rate of 0.44 m$^2$ kg$^{-1}$ day$^{-1}$, similar to that of the low-wind events. We therefore observe a temperature dependence of SSA decay like Cabanes et al. (2003). Based on the limited number of events used here, we document low-winds having a similar effect to air temperatures below -20°C on the SSA decay rate."*

Line 208. Are the units correct here?

Apologies, these have been changed.

Lines 204-205. No influence of basic environmental variables. How about cloudiness? A very important variable for SSA decay is the temperature gradient in the snowpack. Near the surface, this is going to be greatly affected by cloudiness. In the absence of clouds, there will be a much stronger temperature gradient near the surface than under cloudy conditions. This probably deserves a bit of exploration. Various proxies for cloudiness can be tested, in particular the longwave budget.

We had explored this in the original manuscript and found that there is no significant relationship between the SSA decay rate and cloudiness based on linear regression analysis. However, to clarify, we are not suggesting that these variables do not affect the SSA and the decay rate, but that based on our data alone, we do not observe a significant relationship. We do observe an interesting relationship between the principal components of SSA, d-excess and $\delta^{18}O$, and cloudiness/longwave radiation over the entire sampling period. The purpose of this analysis was to identify any systematic influence of the decay rate for the defined events, and therefore, we decide to focus on the dominant influences on the events we are analysing.

We evaluate cloudiness when assessing the isotopic change during low-wind events. However, both events correspond to near constant clear skies.

Lines 242-243. Shaded regions in Fig 4 are said to indicate largely homogeneous snow cover. But The caption to Figure 4 says "Grey shaded regions indicate periods of high spatial variability in isotopic composition." I am confused.

Apologies for the mistake. To fix this inconsistency and to improve the coherence of the manuscript, we move the EOF analysis prior to the SSA decay model results. The principal components of each variable (SSA, $\delta^{18}O$ and d-excess) are assessed for statistical significance, and we find that there are opposing regimes between the years. In 2019, $\delta^{18}O$ and d-excess covary in the spatial and temporal dimensions, contrasted with the strong significant relationship between the principal components of SSA and d-excess in 2019. It is apparent that the two years differed significantly in overall temperature conditions, which is clear is the mean $\delta^{18}O$ values, which is potentially related to the opposing NAO phase in 2017/2018 and 2019. Even in the SSA decay events the behaviour is different. The specific SSA decay shape, which is clearly identifiable in 2017 and 2019 is less obvious in 2018. Furthermore, this is relevant for the discussion of processes driving isotopic change in the low-wind events.

Lines 241-249. This discusses the correlation between SSA and d-excess. The coherence is better when the snow layer is homogeneous. Could that just be due to wind effects? When the wind speed is low and there is no wind drifting, the snow remains unperturbed and *a priori* homogeneous. On the contrary, under greater wind speeds, drifting takes place, heterogeneity is generated and SSA and d-excess become decorrelated. Furthermore, since SSA measurements probe about the top 1 cm while isotopic measurements probe the top 2.5 cm, it is clear that when wind drifting takes place, both measurements may measure highly different layers, explaining the decorrelation. How about limiting data analysis to those events without wind speed?

This is a useful insight from the reviewer, and we acknowledge that this could be the case. The correlation in 2019 is continuous throughout the season, which suggests that increases in PC1 of SSA, closely linked to precipitation, and decreases, closely linked to post depositional processes, are similarly influencing d-excess. The following text is added to the discussion:

*"PC1 of SSA is interpreted as depositional events causing increase in SSA in the positive mode (Domine et al., 2009), and snow metamorphism or wind erosion in the negative (Cabanes et al., 2002, 2003; Legagneux et al., 2003, 2004; Taillandier et al., 2007a; Flanner and Zender,*

*2006). However, we consider potential increases in SSA without precipitation in the instance of 1) surface hoar formation on an aged snow surface (SSA < 50 m² kg⁻¹), 2) the effective sieving of small, fragmented grains into the pore space via wind, and 3) from sublimation and subsequent fragmentation of snow grains while suspended by the wind (Domine et al., 2009).*

For the revised manuscript, we look in detail at the low-wind events only to ensure the same surface layer persists. By reducing the number of events, we can assess temperature gradients and latent heat flux for individual events, allowing for a more concise discussion.

An issue with limiting EOF analysis to the low-wind events alone is that the deposition/precipitation input is then removed, which is a key component of the relationship between SSA and d-excess while at the surface. A later comment from the reviewer observes that large increases in SSA (possibly precipitation, or another form of deposition) corresponds almost always to an increase in d-excess. We argue that this observation supports the argument that there is an overall decrease in d-excess during snow metamorphism.

Lines 256-257. Here the authors mention fog and negative LHF, i.e. likely surface hoar formation. Thus the authors may have observed snow conditions. All these observations must be mentioned when results are first presented. Data analysis must consider which processes were involved for each event. By the way, the standard abbreviation for latent heat fluxes is LE, not LHF.

We hope the previous responses have clarified the observations that were made. Observations are in Table A1 in the appendix, as well as a new plot with these observations indicated on the timeseries. LHF has been changed to LE throughout the text. Isotopic analysis of low-wind events now includes the LE and temperature gradient measurements to infer the vapour fluxes in the surface snow.

Line 268. The authors invoked re-exposed old snow to explain some d-excess values. Careful observations during sampling can answer this question. If there was 1 cm of recent snow over old snow, the SSA measurement will have measured recent snow while isotopic measurements will have measured predominantly old snow. This will affect the quality of the SSA-d-excess correlation analysis. Again, inadequate samples must be removed from the analysis.

Unfortunately, there is no precise documentation of layering of the snow used for samples. Instead, we refer to the accumulation data to identify changes in snow surface height during the analysis of isotopic change for the low-wind events.

Line 287-288. Changes in snow physical properties observed are probably not due to precipitation and metamorphism *sensu stricto* (i.e. involving only water vapor transport within the snow layer). Processes involved also include wind drift, fog deposition, surface hoar deposition, and also possibly sublimation crystal formation. This last process is due to vapor transport within the snow, but since the growth of completely new crystals is involved, I suspect their isotopic composition would be very different from that of the snow layer they originate from. Sublimation crystals are in fact very frequent on cold snow under intense sunlight, even though reports are few (Weller, 1969; Gallet et al., 2014).

These are really useful points from the reviewer. Looking at surface crystal growth through the perspective of isotopes to determine sublimation crystals from deposition of hoar crystals would be interesting, and a great contribution to the quantification of sublimation driven isotopic fractionation. The sampling strategy used here favoured a broad study looking at the macroscale relationships between snow metamorphism and isotopic composition, and the large decrease threshold was used to extract changes in SSA over the transect after high initial SSA values had been recorded.

Given that surface hoar/sublimation crystals were not documented, we use LE measurements to determine whether there was significant surface hoar formation during analysed events. Determining sublimation crystals is more ambiguous here but we look at temperature gradients throughout the events to explore the possibility. We refer here to the recent paper by Casado et al. (2021) where the snow isotopic composition and modelled precipitation isotopes were used to infer the relative influence of precipitation and snow metamorphism on the isotopic signal. To accommodate this comment and significantly strengthen our study, the revised discussion presents the expected fractionation effects of processes driving snow metamorphism and infers the mechanisms of isotopic change based on previous studies (Hughes et al., 2021; Wahl et al., 2021; Casado et al., 2021). The following text is added:

*"Three key mechanisms are expected to drive the rapid SSA decays; 1) large grains growing at the expense of small grains (Legagneux et al., 2004; Flanner and Zender, 2006), 2) diffusion of interstitial water vapour (Ebner et al., 2017; Touzeau et al., 2018; Colbeck, 1983), 3) sublimation due to the wind ventilating the saturated pore air, known as 'wind-pumping' (Neumann and Waddington, 2004; Town et al., 2008). The dominant mechanisms can theoretically be identified by a combination of the change in isotopic composition - indicating the fractionation effect - and the LE and temperature gradient data.*

*In theory, mechanism 1) causes minimal change in the bulk isotopic composition of a snow layer under isothermal conditions (Ebner et al., 2017). Therefore, observations of SSA decay corresponding to negligible isotopic composition change could be explained by this mechanism. We observe no events with consistent isotopic composition throughout. In the instance of 2) interstitial diffusion, light isotopes are preferentially diffused, while the heavy isotopes will be preferentially deposited onto the cold snow grains (Ebner et al., 2017; Touzeau et al., 2018; Colbeck, 1983). Thus, diffusion of water vapour in the pore space causes a decrease in d-excess and slight increases in δ18O due to kinetic fractionation (Casado et al., 2021). 3) Sublimation has been widely documented to cause an increase in δ18O of the remaining snow mass due to equilibrium fractionation, and a significant decrease in d-excess due to kinetic fractionation (Ritter et al., 2016; Madsen et al., 2019; Hughes et al., 2021; Wahl et al., 2021; Casado et al., 2021).*

*An overall increase in δ18O and decrease in d-excess during E10 can be attributed to a combination of 2) and 3) based on observation of net-sublimation and high amplitude diurnal temperature gradient variability indicating vapour transport within the pore space. The period between 9th June at 15:18 UTC and 10th June 10:40 UTC recorded net deposition corresponding to an overall decrease in δ18O during the first day and minimal decrease in d-excess, potentially due deposition of atmospheric water vapour (Stenni et al., 2016; Feher et al., 2021; Casado et al., 2021).*

*A 30% decrease in d-excess corresponds to negligible change in δ18O during E11. Net-sublimation, double that of E10 is measured, but with reduced amplitude in both TGs. Moreover, the largest decrease in d-excess occurs after the first day when the surface-subsurface TG is consistently negative. This indicates that vapour diffusion is controlling the isotopic composition, and the effect of equilibrium fractionation during sublimation from the surface only weakly influences the bulk isotopic composition (Casado et al., 2021)."*

Line 291-292. For older snow also, sublimation and vapor diffusion are not the only processes involved. In particular, wind drifting is probably important.

This has been included in the text.

Line 297. The correct reference is Cabanes 2003, not 2002

We apologise for this mistake and have changed this in the text.

Line 309-310. The comparison with the equation of Taillandier should be indicated in results. In fact, the choice of Cabanes' equation should be justified earlier on. Its interest as well. By the way, (Cabanes et al., 2003) used a temperature-dependent exponential coefficient.

In addition to a more extensive introduction to the models in the methods "modelling surface snow metamorphism" we have added a brief inter-model comparison to the results, using the temperature-gradient model from Taillandier et al. (2007), and the model from Flanner and Zender (2006), with tau and n determined by their look-up table based on the event conditions.

Lines 311-318. This paragraph is not physically very sound and is not based by any quantitative analysis. Since the temperature gradient near the snow surface is not evaluated, there is no basis to say that isothermal metamorphism is dominant after precipitation. Then, since the Authors do not find any significant effect of temperature, they assume their observations are explained by the temperature gradient, implicitly implying that the temperature gradient show little variations between events. This paragraph should just be removed. All the statements are unsubstantiated. Furthermore, what is important in TG metamorphism is not the magnitude of the temperature gradient, but the magnitude of the water vapor flux, which is temperature- dependent. Lastly, it can be affected by wind speed through wind pumping and also by convection (Trabant and Benson, 1972; Benson and Trabant, 1973; Johnson et al., 1987;Sturm and Johnson, 1991). All these aspects would need to be discussed and quantified to engage in the discussion proposed in this paragraph.

We apologise for not stating that there are snow temperature measurements from PROMICE from 2017 and 2018, and from a separate campaign from 2019. The inclusion of all events in the original manuscript did not facilitate in-depth analysis of individual events. However, thanks to the reviewers' suggestions, the revised manuscript now includes the temperature gradient and latent heat flux for the low-wind events. Ultimately, this paragraph has been removed, but the influence of temperature gradients on the low-wind events has been discussed in the following sections, where aspects of both paragraphs have been merged.

Lines 319-322. Here again, the authors make unfounded statements. How do they know the temperature gradient is negligible during polar night? Under clear sky conditions, radiative cooling will on the contrary induce strong temperature gradients near the surface of the snow. The authors may just conclude that since their model is empirical it only applies under the conditions where data were obtained. In fact, it may not even be valid at this site in summer during other years.

The paragraph has been corrected and instead of suggesting that temperature gradients are minimal, we have discussed this in terms of absolute temperatures being lower, and thus the SSA decay would be slower (Flanner and Zender, 2006), as evidenced by E7 with temperatures < -30°C.

*"Snow metamorphism is thermally activated given the dominant influence of sublimation and deposition (Cabanes et al., 2002, 2003; Legegneux et al., 2004). During winter, the temperatures are very low (<-30°C) and minimal insolation reduces the diurnal near-surface snow temperature gradients, resulting in isothermal metamorphism being dominant which reduces the rate of snow metamorphism, or SSA decay, compared to temperature gradient snow metamorphism (Dadic et al., 2008)."*

Furthermore, we appreciate that the empirical model construction in this study is limited by synoptic weather variability being consistent wind and potential for surface perturbation. With consideration to this limitation, we believe it is still useful to document with the decay model, given the relationship between SSA and surface energy budget.

Lines 324-331. Could not the authors compare their model to data obtained using the algorithms developed in (Kokhanovsky et al., 2019)? It seems possible to determine precipitation events using Sentinel data, as indicated by high-SSA periods, and then investigate the decay to test whether the model developed here indeed applied to the accumulation zone of the GIS. This paragraph lacks convincing arguments and sound a bit like just wishful thinking, while tests are possible.

Yes, we agree that this would be a useful comparison. However, for this paper we decided to focus on the relationship with isotopes. This paragraph is removed, and the satellite potential is mentioned briefly in the previous section instead related to the usefulness of defining the SSA decay rate as a function of different wind-regimes.

Lines 336-337. Why would this correlation between SSA and d-excess be observed in only 72% of cases? I think it would be interesting to explore which events actually monitored a constant layer, rather than a layer perturbed by wind drift, the formation of surface hoar or sublimation crystals, or fog deposition.

This is a useful point from the reviewer, and we have added a section in the results that looks only at the isotopic change during the minimally perturbed low-wind events. We have included LE and temperature gradient data to identify the dominant direction of vapour flux during the events. Isotopic change is now documented in the context of sublimation and deposition between the surface and atmosphere, while the 10 cm snow temperature data gives an indication of the direction of vapour flux within the snow. The discussion is edited in parallel

with a more concise comparison to expectations from previous studies such as Casado et al. (2021). The same analysis has now been applied to events with minimal perturbation from ground fog, snowdrift, and snowfall.

Lines 339-351. This discussion of snow metamorphism could be significantly improved. I am not sure surface curvature effects played a detectable role. In any case, the authors need to substantiate this with quantitative calculations, they cannot just make such statements without a demonstration. I would think water vapor fluxes caused by temperature gradients and wind pumping, and perhaps thermal convection, can explain most observations.

We acknowledge the reviewer's suggestions and have modified the discussion to explain the increased decay rate under moderate-wind conditions. The following text is added to the discussion:

*"The expected temperature dependence on the SSA decay rate is apparent during E9, where the mean air temperature in less than -20◦C, which agrees with the accepted knowledge that snow metamorphism is slower in colder conditions due to sublimation and deposition being thermally activated processes (Cabanes et al., 2003). In addition, we focus on the influence of wind-speed of the SSA decay rate and observe a more rapid SSA decay with increased wind-speed, potentially due to increased ventilation of saturated pore air acting as a catalyst for snow metamorphism (Cabanes et al., 2003; Flanner and Zender, 2006; Neumann and Waddington, 2004). Wind erosion cannot be definitively ruled out due to dis-continuous documentation of surface conditions. However, high wind-speeds are documented to increase SSA via fragmentation and sublimation of suspended snow crystals, which are then re-deposited and effectively sieved into the pore spaces of the surface snow layer (Domine et al., 2009)."*

Lines 360-366. This paragraph discusses the relationship between SSA increases and concomitant d-excess increases. However, this seems very misleading to me. This paper is focused on SSA decrease of a given snow layer over time. Here, the approach is different. The authors consider changes in the SSA of surface snow, regardless of whether these changes involve the same layer. In fact, their SSA increases seems to always involve a change in layer, e.g. due to precipitation. Therefore, plotting data obtained by the evolution of a given identified layer together with data involving a change of layer seems meaningless to me. What I understand from this paragraph is that new layers with high SSA have a higher d-excess value than older (and different) layers with low SSA. This may be interesting, but is different from the main topic of this paper, and should therefore not presented as the same topic.

We acknowledge that there was a lack of clarity here and appreciate the reviewer's comments to allow us to clarify and strengthen our findings. By measuring the isotopic composition of the SSA sample, we remove the uncertainty from spatial variability. Analysing isotopic change over 2-days ensures that new/re-deposited snow will have more time to equilibrate with the sub-surface snow. If repeated, precipitation and surface hoar isotopes would be measured to determine the influence of the surface depositions on the 2.5 cm isotope measurements. We add this as a limitation of our study.

Regarding the observations of snow with high SSA having a higher d-excess value than old

snow, we acknowledge the reviewer's comment that this is not the same topic but propose the this feature as supporting evidence. Given that we observe no seasonal trend in d-excess, the consistently increased d-excess values with high SSA cannot be attributed to increasing d-excess throughout the season. Moreover, the documentation of d-excess decrease during low-wind events ensures negligible removal of snow. Therefore, with the support of LE and temperature data, we argue that this feature is the result of decrease in d-excess during snow metamorphism due to the combined influence of grain growth via vapour diffusion, and sublimation into the atmosphere (Ebner et al., 2017; Hughes et al., 2021; Wahl et al., 2021; Casado et al., 2021).

Lines 368-373. It is surprising to see data presented in the discussion. This should be in the results section. So in fact there seems to have been observations of snow surface conditions and changes. Wind drifting, a key process for data interpretation, may have been observed after all. We need to see those data. Fig. A1 needs to also show mean hourly wind speed, and ideally maximum hourly wind speed if available, as well as observations of drifting. In fact, all surface snow observations, including fog deposition, the formation of surface hoar or sublimation crystals, and any other relevant information, must be shown in a Figure.

This figure has been incorporated into the results section, where we present the daily and 2-day change in one figure and only for events with minimal surface perturbation. The additional figures and description of conditions have been added at the start of the results.

Lines 393-399. The speculation between insolation, temperature gradient and d-excess may be potentially interesting, but lacks a clear basis. Since the authors did not measure T gradients and did not adequately discuss their role on d-excess, I think this paragraph is not very useful. Please substantiate or remove.

This paragraph has ultimately been removed. We agree with the reviewer that this is an interesting discussion point, but based on our results alone, we feel we cannot adequately substantiate the arguments.

The section on ice core implications could perhaps be strengthened a bit by treating specific examples. For examples, how is the d-excess signal affected by more frequent precipitation that metamorphose without wind perturbation, in comparison to precipitation events that rapidly form a wind slab with time-stable SSA? How does that relate to climate scenarios (e.g. glacial vs. interglacial). This is just a suggestion. I am sure the Authors can present other interesting cases. This is where I expected more in-depth discussions.

We agree that there is a lot of potential discussion points relating to implications for ice core studies. Specific examples are addressed in the section "Isotopic change during SSA decay events", where we compare our observations to the fractionation effects expected from the different processes driving snow metamorphism. In addition, we discuss the inter-play between precipitation intermittency and temperature conditions as an explanation for the different regimes between 2018 and 2019. We appreciate that there are numerous interesting discussion points which could be added, and we thank the reviewer for the suggestions made here.

[revised manuscript text omitted]

65  depending on surface conditions (Cabanes et al., 2003; Pinzer and Schneebeli, 2009a), each potentially modifying the isotopic composition of the snow (Ebner et al., 2017).

Models can provide a quantitative description of the rapid SSA decrease after deposition. Previous studies have proposed SSA decay models using a combination of field measurements and controlled laboratory experiments (Cabanes et al., 2002, 2003; Legagneux et al., 2003, 2004; Flanner and Zender, 2006; Taillandier et al., 2007).

70   Exponential models to describe SSA decay are documented to be the best fitting to in-situ data (Cabanes et al., 2003). However, the lack of a physical basis led Legagneux et al. (2003) to construct a theoretical equation to describe SSA decay based on grain growth theory, which was then developed by Flanner and Zender (2006) who defined parameters based on surface temperature, temperature gradient and snow density.

75  Existing SSA decay models have not yet been extensively applied to polar ice sheet surface snow. Conditions for surface snow on polar ice sheets  are not necessarily comparable to other alpine  and Arctic regions regarding negligible melt and the high-latitude radiation budget

. Moreover, while continuous surface SSA measurements exist from

80    Antarctica (Gallet et al., 2011, 2014; Picard et al., 2014 ice core studies. In particular, studies of SSA and~~

[revised manuscript text omitted]

Mean and standard deviation for weather variables, surface temperature (calculated from upwards and downwards long-wave radiation with long-wave em... set to 0.97), relative humidity, wind speed and latent heat flux during the three sampling seasons. Surface temperature, relative humidity and wind sp... PROMICE weather station based on 10-minute measurements. Latent heat flux an upwards flux from the eddy-covariance tower.

sample with a known volume. At the start of each season, sticks were placed at each site and snow height was determined by the distance between the snow surface and top of the stick. Accumulation was calculated using the cumulative sum of the daily difference between measurements of snow height from each site. The resultant datasets consist of 10 daily measurements of three parameters, SSA, density and accumulation, over a  89, 94 and 66 day period for 2017, 2018 and 2019 respectively.

 The samples were all taken in the day time, primarily in the morning.  The meteorological data is re-sampled to the SSA sampling time-periods to ensure consistent comparison.

**2.3 SSA measurements**

**2.4**

Each snow sample is placed into the Ice Cube sampling container below an Infra-Red (IR) laser diode (1310 nm), where the SSA is calculated based on IR hemispherical reflectance, explained in Gallet et al. (2009), while information on the Ice Cube device can be found in Zuanon (2013).  The *e*-folding depth of 1310 nm radiation in snow of 200 kg m$^{-3}$ is approximately 1 cm  2017, 2018 and 2019  is 293 kg m$^{-3}$ (307±40 kg m$^{-3}$, 278±47 kg m$^{-3}$, 294±50 kg m$^{-3}$ )for 2017, 2018 and 2019 respectively), resulting in each measurement being heavily weighted to the top <1 cm of the 2.5 cm sample. The light reflected from the snow samples is converted into inter-hemispheric IR reflectance using a calibration curve based on methane absorption methods (Gallet et al., 2009). A radiative-transfer model is used to

retrieve SSA from inter-hemispherical IR reflectance. To avoid influence from solar radiation, SSA was measured inside a
ventilated white tent kept at temperatures between -5 °C and -10 °C. SSA measurements have an uncertainty of 10 % for values
between 5–130 m² kg⁻¹ (Gallet et al., 2009).

**2.4 Surface snow isotopes**

 Individual SSA samples were put in separate bags and subsequently sampled for water isotopic composition. Thus, every day the  SSA samples have a corresponding isotopic composition. The resultant isotope value is the average composition over the top 2.5  cm of snow. Each sample was sealed in polyethylene bags to avoid any air to equilibrate with the snow and affect the isotopic composition. All samples were kept frozen during transportation and storage.

After melting, each bag was shaken to ensure the isotopic composition of the sample is representative. $1.25 \mu l$ of each sample was then pipetted into a vial ready for isotopic analysis. The snow samples were then analysed at Alfed Wegener Institute in Bremerhaven using a cavity ring-down spectroscopy instrument model Picarro L-2120-i and L-2140-i following the protocol of Van Geldern and Barth (2012). This technique is used to obtain measurements of $\delta^{18}$O and $\delta$D with an uncertainty of 0.15‰ and 0.8‰ respectively.  The calcualted values for $d$-excess have an uncertainty of 1‰. Observing relationships between our SSA and isotope data requires consideration for the depth offset between the SSA measurements and the isotopic composition measurement which measures the entire 2.5 cm snow layer.

**2.5 Data analysis**

**2.6**

**2.5.1 Defining SSA decay events**

 To systematically identify rapid decreases in SSA, which we use as a proxy for events of snow metamorphism after deposition (identified based on the high mean SSA values), a threshold is set using the bottom 10th  percentile of SSA decreases over a two-day period (-13 m² kg⁻¹  2-day⁻¹). This was found to result in the most equal number of events from each sampling year compared to 1- and 3-day changes. SSA decay events are defined as by the initial peak, identified by the threshold, through to the next increase in SSA (rather than decrease).

We here use the term deposition events to describe rapid increases in SSA, expected to be from precipitation, drifted snow
or hoar formation. Previous studies have indicated that surface hoar and sublimation crystal-like grain growth features at the
surface have an SSA value around $54\,\mathrm{m^2\,kg^{-1}}$, based on  SSA of hoar frost (Domine et al., 2009). Accumulation data and field observations are used to identify the
initial conditions.

A set of criteria are required to reduce the potential of analysing events with wind-perturbed surfaces, resulting in the
removal of surface snow. In Antarctica, unconsolidated surface snow has been observed to drift at wind speeds as low as
$5\,\mathrm{m\,s^{-1}}$ measured at $2\,\mathrm{m}$ height (Birnbaum et al., 2010). However, a study from Greenland documented snowdrift starting
at $6\,\mathrm{m\,s^{-1}}$ (Christiansen, 2001), likely due to warmer temperatures allowing for the surface snow to become more bonded
(Li and Pomeroy, 1997). At EastGRIP, calm conditions correspond to wind speeds from $0\text{--}5.2\,\mathrm{m\,s^{-1}}$ according to field diary
observations. The mean daily maximum wind speed for the three sampling seasons was $6.8\,\mathrm{m\,s^{-1}}$, while blowing snow was
documented only when wind speeds exceeded $7\,\mathrm{m\,s^{-1}}$.

Based on this assessment, we define two wind-speed categories for comparison of the effects of wind-speed on SSA decrease.
The first includes events with wind-speed consistently below $5.2\,\mathrm{m\,s^{-1}}$, hereafter referred to as low-wind events, to ensure no
surface perturbation. Secondly, we consider events where the maximum wind-speed is between $6\text{--}7\,\mathrm{m\,s^{-1}}$, hereafter referred
to as the moderate-wind events. The inclusion moderate-wind events allows an assessment of the influence of wind-speed on
SSA decrease.

**2.5.2 Modelling surface snow metamorphism**

The first empirical SSA decay model was proposed by (Cabanes et al., 2003) who described a temperature-dependent exponential
decay based on snow samples collected from the Alps (Cabanes et al., 2002) and Arctic Canada (Cabanes et al., 2003). A
following logarithmic equation (Eq. (2)) fit controlled to laboratory experiments was proposed by (Legagneux et al., 2003),
where parameters A and B were found to be arbitrarily related to the decay rate and initial SSA of each sample, and are linearly
correlated at -15 °C.

$$SSA(t) = SSA_0 \cdot e^{-\alpha t} \tag{1}$$

$$SSA(t) = B - A \cdot ln(t + \Delta t) \tag{2}$$

To improve the physical basis of the model, the theory of Ostwald Ripening, describing grain growth driven by a physical
need to reduce surface energy, was implemented into the model (Legagneux et al., 2004). The equation (Eq. (3)) has two
parameters $\tau$ and $n$; $\tau$ is the decay rate and $n$ relates to theoretical grain growth. The physical model was developed by
Flanner and Zender (2006) to incorporate more specific physical quantification to the parameters to include information about

temperature, temperature gradient, and density. Based on these three conditions, they created a look-up table for $\tau$ and $n$.

$$SSA(t) = SSA_0 \left( \frac{\tau}{t + \tau} \right)^{1/n} \tag{3}$$

Taillandier et al. (2007) proposed two equations based on the logarithmic model, defined by Legagneux et al. (2004), to define the decay rate under isothermal and temperature gradient conditions where they were able to directly incorporate a surface temperature parameter.

An empirical decay model is constructed building upon previous studies (Cabanes et al., 2002, 2003; Flanner and Zender, 2006; Legagneux et al., 2002, 2003; Taillandier et al., 2007). This model uses continuous daily SSA measurements from EastGRIP to describe the behaviour of surface snow SSA in polar summer conditions. All samples of defined SSA decay events are used to quantify surface snow metamorphism.

$$SSA(t) = SSA_0\,e^{-\alpha t}$$

**3 Results**

$_0$

**3.1 EastGRIP conditions**

Meteorological variables over the three sampling seasons vary substantially. Figure 2 shows the 10-minute mean values of air temperature, wind-speed, relative humidity and latent heat flux (LE). The accumulation in Fig. 2d are daily mean values (see Section 2.2). Air temperatures were below 30 °C between May 5th and May 8th, such low temperatures were not recorded for 2017 and 2019. However, when comparing the period from May 27th (start of 2019 season) to August 5th of each year, 2018 air temperatures (-13.3 °C) were still 0.5 °C lower than 2017 and 3.2 °C lower than 2019. Two days during 2019 recorded air temperature above 0 °C.

The 2017 season was characterised by high wind intrusions of $>13\,\mathrm{m\,s}^{-1}$  at approximately 20-day intervals. Considering all three sampling years, 2017, 2018 and 2019, the average daily maximum wind speed is $7\,\mathrm{m\,s}^{-1}$, with 209 out of the total 237 sampling days having maximum wind speed above $5\,\mathrm{m\,s}^{-1}$. The distributions of daily maximum wind-speed compared to 10-minute mean values are found in the Supplemental Fig. A1. Relative humidity is consistent throughout the years with mean values around 95% and similar variability of $\sim$7%.

[Figure]

**Figure 2.** *Meteorological data from 2017, 2018 and 2019*

Data is presented for the specific sampling periods for each year. The 10-minute mean data from PROMICE is shown for air temperature (a), wind-speed (b) and relative humidity (c). The bold lines indicate the mean values, based on the snow sampling time interval, for air temperature and relative humidity, and the maximum value for wind-speed. The Relative humidity is determined from vapour pressure and saturation vapour pressure. Latent heat flux (d) is 10-minute averages from the eddy-covariance tower, with the bold grey line showing the daily sum. Accumulation is presented in panel d).

**4 Results**

**3.1 SSA decay events**

There was a total of 5 cm accumulated snow over the 89-day season of 2017, half the amount of 2018 and 2019. The field season for 2018 started on the 5th of May, 9-days earlier than 2017 (14th May), and 22 days earlier than 2019 (27th May). Substantially more sublimation was recorded in 2019, where the daily sum was approximately double that of 2018.

**3.0.1 Spatial and temporal surface variability**

SSA data collected at EastGRIP indicate continuous changes in the physical structure of the snow crystals during all sampling
seasons, with both temporal and spatial variability. The temporal SSA variability shows changes in physical snow structure
with peak values closely associated with precipitation and decreases A recent study at EastGRIP has shown the significant
in-homogeneity in surface snow due to post-depositional re-working reworking of the snow . Summer seasonal SSA evolution
is presented in Fig. ?? for 2017, 2018 and 2019 with each faded line representing individual samples (10 per day), and the bold
line showing the daily mean. Spatial variability between sites is most prevalent when there are high SSA values, indicating
fresh snow.

A total of 21 rapid SSAdecay events are identified, with 6(Zuhr et al., 2021). To avoid attenuation of isotopic signal, each
sample is treated independently. Using a confidence interval of 95% (p<0.05), the relationship between SSA and isotopic

[Figure]

**Figure 3.** *SSA Timeseries 2017, 2018 and 2019* Time-series of SSA time-series between May and August for (a)2017, d18O (b)2018 and ,
*d*-excess (c) 2019. Faded lines represent and the 10 individual samples from principal components (PC1) of each variable (d). For each plot,
the 90 m markers indicate the individual sampling transect, while sites and the bold line link shows the daily mean values. Gaps The secondary
y-axis in panel a) shows the timeseries represent missing data accumulation. Grey The grey bars highlight indicate the periods of decrease in
SSA defined by the threshold algorithm for each year. Six decrease decay events are observed in 2017 and 2019, while nine are observed in
2018. Decrease events are interpreted as rapid grain growth due to snow metamorphism, and stars indicating days with precipitation.

composition is tested using Empirical Orthogonal Function (EOF) analysis. The purpose of EOF analysis is to identify the dominant modes of variance in both the temporal and spatial dimensions for each parameter - SSA,  and $d$-excess - which are all measured from the same sample.

All parameters continuously change throughout the field seasons of 2017, 2018 and 2019  (Fig.3), with large spatial variability in isotopic composition. SSA is characterised by peaks, often corresponding to large spatial variability, followed by gradual decreases over a number of days, a feature which is most prominent during 2017 and corresponds to negligible accumulation. The amplitude of SSA variability is largest in 2019. The start of the 2018 season has very high SSA values (daily mean $88\,\text{m}^2\,\text{kg}^{-1}$) corresponding to low and homogeneous $\delta^{18}$O. Maximum SSA values  of individual samples for 2017, 2018 and 2019 are 85.3 $\text{m}^2\,\text{kg}^{-1}$  95.3 $\text{m}^2\,\text{kg}^{-1}$ $^{-1}$ and $86.7\,\text{m}^2\,\text{kg}^{-1}$ respectively.

 Inter-annual variability is observed in $\delta^{18}$O, with seasonal mean values of -31.6‰, -32.7‰ and -27.3‰ for 2017, 2018 and 2019 respectively (Fig. 3a). Note that the 2019 field season started approximately 15 days later than 2017 and 2018, resulting in a bias towards mid-summer conditions. Throughout the season $\delta^{18}$O follows a gradual increasing trend from May to August. Some cases of abrupt decreases (-10‰) are observed in the late summer, for example, on July 12th in 2018 and July 25th in 2019. No clear seasonal trend is observed in $d$-excess (Fig. 3b) but with periods of gradual decreases. Total daily spread in $\delta^{18}$O and $d$-excess is approximately 15‰.

The spatial and temporal principal components of each variable are presented in Fig.  3d. During 2017, 2018 and 2019 all variables have one dominant mode of variance, or principle component (PC1). PC1 of SSA ($\text{PC1}_{\text{SSA}}$) explains 61%, 77% and 72% of variance for the respective years, PC1 of $\delta^{18}$O ($\text{PC1}_{\delta^{18}\text{O}}$) explains 69%, 83% and 75% of the total variance respectively, while PC1 of $d$-excess ($\text{PC1}_{d-\text{excess}}$) explains 47%, 51% and 60%.

Distinct differences are observed between the sampling years, most prevalent is the opposing regime from 2018 to 2019. During 2018 $\text{PC1}_{\delta^{18}\text{O}}$ and $\text{PC1}_{d-\text{excess}}$ exhibit a significant relationship, with a strong negative correlation for the spatial component of $\text{PC1}_{\delta^{18}\text{O}}$ and $\text{PC1}_{d}-\text{excess}$. A significant relationship is also observed for the temporal component of $\text{PC1}_{\text{SSA}}$ and $\text{PC1}_{\delta^{18}\text{O}}$. In contrast, data from 2019 are characterised by significant relationships between $\text{PC1}_{\text{SSA}}$ and of $\text{PC1}_{d}-\text{excess}$ in both the spatial (r=0.75) and temporal dimensions. No relationship is observed between $\text{PC1}_{\delta^{18}\text{O}}$ and $\text{PC1}_{d-\text{excess}}$ during 2019. For 2017, significant relationships (p<0.05, 95% confidence) are observed between the temporal component of $\text{PC1}_{\text{SSA}}$ and $\text{PC1}_{d-\text{excess}}$, and the temporal and spatial component of $\text{PC1}_{\delta^{18}\text{O}}$ and $\text{PC1}_{d-\text{excess}}$. A shift is observed after July 15th where $\text{PC1}_{d-\text{excess}}$ changes from co-varying with $\text{PC1}_{\delta^{18}\text{O}}$ to $\text{PC1}_{\text{SSA}}$.

**3.1 SSA decay events**

**3.2**

Continuous SSA measurements allow for the construction of an empirical model to describe SSA decay at EastGRIP through time while exposed to surface processes. A visual inspection of the SSA decay events highlighted in Fig. 3a indicates a relationship between initial SSA and subsequent magnitude of decrease. Prior to analysis, we assess the meteorological conditions and field observations to remove SSA decay events with potentially perturbed surface snow. This includes all events coinciding with observations of ground fog, snowdrift, and snowfall (indicated in Fig. 3), and events where the wind-speed exceeds the thresholds defined in Section 2.5.1.

From the years 2017, 2018 and 2019 a total of 21 events are identified that fulfil the SSA decay criteria (as defined in Section 2.5.1). These events are named E1, E2 etc (see Table A for more information on the individual events). Exploring weather conditions for these events reveals that 12 out of the 21 events are influenced by either snowdrift, snowfall, or ground fog according to field diary observations. Of the remaining 9 events, two are in the low-wind category (E10 and E11$=5.1\,\mathrm{m\,s^{-1}}$), and 7 in the moderate-wind category. Both E10 and E11 had consistent clear sky conditions. We note here that E11 was preceded by significant ground fog, not snowfall, indicating that the peak value of $46\,\mathrm{m^2\,kg^{-1}}$  was likely the result of surface hoar, and thus, rapid SSA decay follows an SSA peak not caused by precipitation.

SSA samples are treated individually to quantify SSA decay rate for the different categories. The rate of SSA decay is closely linked to the SSA value at the start of each event (initial SSA vs. magnitude of decrease during the decay period  (Fig. 4), suggesting the rate of change is proportional to the absolute value, as described by exponential decay law (r=-0.71 and r=-0.84 for low- and moderate-wind events respectively) (Cabanes et al., 2003).

The mean air temperature for all SSA decay events was between -20.8 °C and -7 °C. The first day of each event is characterised by the largest change in SSA, followed by a decrease in magnitude over the subsequent days. This feature is most apparent for the longer events (E1, E2 and E4), where SSA has minimal change below $25\,\mathrm{m^2\,kg^{-1}}$.

**3.2 EastGRIP SSA decay model**

SSA decay rate is quantified by plotting the rate of change in SSA per day against the absolute SSA value for all 10 sampling sites for  low- and moderate- wind events (Fig. 4a). We observe a linear relationship between the rate of change in SSA per day (ΔSSA) and SSA. ~~Outliers are measurements from days with mean air temperature below -25 °C as highlighted in Fig. 4a. This observation is in agreement with theoretical understanding of snow crystal formation transitioning from dendrites to columns at approximately -22 °C (Domine et al., 2008). We therefore define the SSA decay model for a temperature range between -25 °C and 0 °C and daily mean wind speeds below 6 m s⁻¹ based on hourly averaged values.~~ An overview of event conditions using field observations are presented in Table A.

[Figure]

**Figure 4.** *Decay Model Construction and Predictions*

Linear regressions for change in SSA against the SSA for the low-wind (blue) and moderate-wind (purple) SSA decay events (a). Filled markers indicate the daily mean values and transparent markers show the individual samples sites. The observed SSA decays are show for the moderate-wind events (b), and the low wind events (c), followed by the modelled SSA decays for the respective events in d) and c). The legend in d) and e) indicates the SSA decay event number, presented in Table A

 The SSA decay model for EastGRIP is  constructed using the differential equation for the linear relationship between ΔSSA and absolute SSA . Solving the differential with respect to time (t), produces the SSA decay model defined as Eq. (4), which follows the equation structure  of Eq. (1).

$$\frac{dSSA}{dt} SSA(t) = -0.54 (SSA_0 - C) e^{-\alpha \cdot t} + 14.69 C \tag{4}$$

$$SSA(t) = (SSA_0 - 26.8) e^{-0.54t} + 26.8$$

Where SSA(t) is the SSA measurement at a given time, $SSA_0$ is the initial SSA value,  $\alpha$ is the decay rate, and $C$ is the constant. The decay rate, determined by the slope of the linear regressions in Fig. 4, is higher for moderate-wind SSA decay events (-0.53 m$^2$ kg$^{-1}$ day$^{-1}$  ) than for low-wind SSA decay events (-0.41 m$^2$ kg$^{-1}$ day$^{-1}$). To account for a non-zero decay constant,  $C$ describes the 'background' SSA state which is defined by the value of $x$ when the linear regression crosses the  $y$-axis in Fig. 4a

**3.2.1**

**~~All samples for all events are included in plot a) showing the relationship between the rate of change in SSA per day ($\Delta$SSA day$^{-1}$) against the daily absolute values. Points are coloured by the daily mean surface temperature. The linear regression is based on values for surface temperatures between -25°C and 0°C, and daily mean wind speeds below $6\,\mathrm{m\,s^{-1}}$. b) shows a comparison between the model predicted SSA values using Eq. (4), against the SSA observations. The marker colour represents the day of the events (DOE). Marker style represents the sampling year to assess inter-annual variability for 2017 (o), 2018 (x) and 2019 (□). c), d) and e) show all included events in full-line and f), g) and h) show the model predictions as the dashed line. E1-E21 refers to events as listed in Table A. Missing data day-1 E1.~~

, equal to $21\,\mathrm{m^2\,kg^{-1}}$ and $24\,\mathrm{m^2\,kg^{-1}}$ for low- and moderate-wind events respectively. Note that events are here named E1, E2  etc. consistent with Table A.

~~There is a minor tendency for the model to underestimate the SSA decrease and thus overestimate the predicted values of SSA as seen in Fig. 4b. Model limitations are most evident during the first day, as seen in Fig. 4, where the modelled decay consistently underestimates the magnitude of decrease. The model has limited ability to predict observations below in the lower range of SSA observations as seen in Fig. 4f, g and h, where the modelled and observed values are compared for each event.~~

have an extent of 2-5 days. To assess model performance in predicting magnitude of SSA decrease for events of different time periods, we compare the predicted versus measured SSA E9 in 2018 is poorly represented by the moderate-wind SSA decay model from this study. The mean air temperature for this event was -20.8°C, 5°C less than the next coldest (E11 at -15.3°C). Fitting the model for E9 alone gives a decay rate of $0.44\,\mathrm{m^2\,kg^{-1}\,day^{-1}}$, similar that of the low-wind events.  We therefore observe a temperature dependence of SSA decay similar to Cabanes et al. (2003). Based on limited number of events, we document low-winds having a similar effect to air temperatures below -20°C on the SSA decay rate. Our results indicate a slower rate of decay under decreased wind-speed conditions. A similar effect is observed for low temperature, as the single SSA decay event in the moderate-wind category but with mean air temperature below -20°C followed the decay rate of  the

 low-wind events.

**3.2.1 Model evaluation**

**Table 2.** *RMSE - Model comparison*

| | Low-wind | | Moderate-wind | | |
|---|---|---|---|---|---|
| | Mean | Individual | Mean | Individual | |
| | $m^2\,kg^{-1}$ | $m^2\,kg^{-1}$ | $m^2\,kg^{-1}$ | $m^2\,kg^{-1}$ | |
| This Study | 3.64 | 4.76 | 2.48 | 3.50 | |
| FZ06 | 3.45 | 7.08 | 1.28 | 2.92 | |
| T07 | 6.34 | 7.11 | 5.63 | 6.10 | |

This Study uses the respective $\alpha$ and $C$ for the low- and moderate-wind events, using daily (mean) and individual samples. FZ06 parameters $\tau$ and $n$ are defined by the look-up table from Flanner and Zender (2006). T07 uses the mean surface temperature for each event.

Model performance is tested by 1) comparing daily predicted decrease to the 10 daily observations, and 2) comparing results from this study to previous models from Flanner and Zender (2006) and Taillandier et al. (2007). Model-data residuals are normally distributed, suggesting no systematic errors in model predictions. The root mean squared error (RMSE)  between model predictions and observed SSA is $4.76\,\mathrm{m^2\,kg^{-1}}$  and $3.50\,\mathrm{m^2\,kg^{-1}}$ for  the low-wind and moderate-wind SSA decay events.

Using the physical-based decay model from Flanner and Zender (2006), hereafter referred to as FZ06, the influence of wind-speed on observed SSA decay rate can be assessed. Low-wind SSA decay events (E10 and E11) are most accurately predicted by FZ06 using the parameter values of $\tau =$  4.5 and $n$  ~~$\mathrm{m^2\,kg^{-1}}$ for 2017 and 2018 respectively. The model adequately predicts rapid SSA decay at EastGRIP within the temperature range, while for colder temperatures, the decay rate is the same but the intercept is significantly higher (Fig. 4a). Overall, for all included events during the three sampling years, behaviour of SSA decay is clearly captured by the model (Fig. 4c,d and e) . Exploring temperature conditions alone we find that the model performs well when daily mean surface temperatures are between -25 °C and 0 °C.~~ 6.1 from the look-up table (Flanner and Zender, 2006). Both the empirical model from this study, and the model from Taillandier et al. (2007), hereafter T07, underestimate the rate of decrease for low-wind decay events, most apparent during the first day of the event E10 (see Fig. A3).

**3.2.2**

weather variables, relative humidity, surface temperature and wind speed on the model-data residuals, with linear regressions
resulting in $r^2 < 0.1$ for all variables. An overview of event conditions using field observations are presented in Table A. Temperatures below -25°C are characterised by the same slope defined by the model (-0.54The data indicates that in natural conditions, wind-speed (between 6 $m^2 kg^{-1} day^{-1}$), but with a significantly higher intercept of 29 $m s^{-1} day^{-1}$ compared to 14.7and 7 $m s^{-1} day^{-1}$for temperatures above -25) increases the surface SSA decay rate ($\alpha$°C. Significant wind drift is expected when hourly mean wind speed exceeds 6= m-0.53, $\alpha$ $s^{-1}$, which happens during 144 days out of the total 258 sampling days from 2017, 2018 and 2019. Results indicate weather has no systematic influence on SSA decay during the first 2-5 days exposed at the surface, and that conditions vary for each event. The model is able to predict all defined decay events between -25=°C and 0°C, indicating mechanisms of decay are the same. Daily mean values are more accurately predicted by the SSA decay model than individual sample sites due to snow surface variability. In-homogeneous surface snow is especially important to consider for isotopic composition, because there is potential for samples to contain snow from different precipitation and/or deposition events. -0.41). RMSE values presented in Table 2 indicate that FZ06 predicts decay with the least error, for both wind-speed categories. Moreover, all models have lowest errors when predicts events in the moderate-wind category.

**3.2.2   Surface snow spatial variability**

*Timeseries of snow isotopes and SSA*Timeseries of $\delta^{18}$O (a), d-excess (b) and SSA (c) for 2017, 2018 and 2019 sampling seasons. d) shows the principle components of each parameter with colors corresponding to the color used to show absolute values. The black vertical lines indicate a break in the x-axis. Each faded line represents individual sample site values, and the thick line is the daily mean. Grey shaded regions indicate periods of high spatial variability in isotopic composition.

**3.3   Isotopic change decay during events**

The characterization of the SSA decays provide a basis to explore how snow metamorphism of surface snow plays a role for the alteration of isotopic composition of Greenland snow after deposition. A recent study at EastGRIP has shown the significant in-homogeneity in surface snow due to post-depositional reworking of the snow (Zuhr et al., 2021). The focus for this manuscript is to identify signal coherence between physical properties and isotopic composition of surface snow subject to precipitation/deposition and post-depositional processes. Autocorrelation analysis shows that isotopic composition values are spatially decorrelated after 10 m ($r^2 < 0.3$ after 10 m). Therefore, to avoid attenuation of isotopic signal, each sample is treated as independent. Isotopic composition is measured from each SSA sample containing snow from the top 2.5 cm of the snow surface , potentially containing snow deposition layers from multiple precipitation events. Surface heterogeneity is considered by using Empirical Orthogonal Function (EOF) analysis to determine the dominant mode of variance for each sampling year. Figure **??** shows timeseries of

The rate of change in SSA during low- and moderate-wind events is explored with respect to the rate of change in isotopic composition, given the covariance identified from EOF analysis. The rate of change in *d*-excess is plotted against the rate of change in SSA (Fig. 5), considering 1- and 2-day time intervals. We here include analysis of 2-day to allow isotopic

[Figure]

**Figure 5.** Isotopic change during all the analysed events are shown, with each point indicating a specific sampling site. The daily change in *d*-excess (d$_{xs}$) and SSA is presented in a), with 0 indicated with the grey dotted lines. The change in *d*-excess and SSA over a 2-day period is shown in b), while the change in *d*-excess is plotted against the absolute *d*-excess values is shown in c). Linear regressions are presented from daily change (light green) and 2-day change (dark green).

equilibration between the existing surface snow and snow deposited in the day preceding the event. The change after 2-days is presented in Table 3 for each low- and moderate-wind event.

All events have an overall change in isotopic composition, with the percentage change in *d*-excess being an order of magnitude higher than that of $\delta^{18}$O
425  . Increasing $\delta^{18}$O

 corresponds to decreasing *d*-excess in 5 out of 8 events. E9, E11 and E13 deviate from this pattern. E9 and E13 both exhibit increases in $\delta^{18}$O
430  and  *d*-excess, whereas E11 is characterised by a slight decrease in $\delta^{18}$O and 27% decrease in *d*-excess.

Using a significance level of 0.05, the relationship between change in *d*-excess ($\Delta d$-excess) and change in SSA ($\Delta$SSA)
435 is assessed. The results are presented in Fig. 7. Firstly, the $\Delta d$-excess over 1-day are normally distributed around a mean of

$\Delta d$-excess values <-4%  tend to correspond to smaller $\Delta$SSA ( -15
440 m$^2$ kg$^{-1}$ –0

60% for 2017, 2018 and 2019 respectively. PC1 of $\delta^{18}$O and d-excess show strong coherence from May to early June during 2017 and 2018, while for the second half of the season, and throughout 2019, PC1 of d-excess corresponds to PC1 of SSA ( Fig. ??d)m$^2$ kg$^{-1}$), suggesting that large decreases in $d$-excess occur after an extended period of exposure. This feature is highlighted in Fig.

Surface variability due to post-depositional reworking of the snow is shown by a wide spread in SSA values during a given day. Time periods with low spatial variability indicate largely homogeneous snow cover over the transect, shown in Fig. ?? as shaded regions. High variability is defined by periods where the 5-day running-mean of spatial variance in $\delta^{18}$O is greater than one standard deviation. During periods of low spatial variability in isotopic composition, there is greater coherence between PC1 of SSA and PC1 of d-excess, due to a reduction of noise in the dataset. PC1 of SSA and d-excess show a coherence during 2018 and 2019 seasons, while the signal is less clear during 2017 7b, where $d$-excess decreases in 59 out of the 80 sites after two days of exposure to surface processes. Initial $d$-excess is observed to have a significant influence the magnitude of $d$-excess decrease over the defined period (Fig. ??b). However, the reduced signal coherence is concurrent with high spatial variability in isotopic composition. 7c), with high initial $d$-excess corresponding to the largest decreases in $d$-excess.

A clear relationship between PC1 of SSA and PC1 of d-excess is observed when there is a relatively homogeneous snow layer over the sampling transect, defined by low spatial variance in $\delta^{18}$O.

*Table of isotopic change for decay events*Behaviour

**Table 3.** Table of snow parameters during decay events are defined. The initial, 2-day and percentage isotopic change over a 2-day period are presented fo

| | | | | | | | | | |
|---|---|---|---|---|---|---|---|---|---|
| | | | | | E1 | | | | |
| | | | | | E2 | | | | |
| | | | E3 -29.55 -30.07 -1.8% -0.2 -0.6 -200% 50.4 31.1 -38.3% E4 | | | | | | |
| E5 -30.23 -29.88 1.2% 6.1 2.9 -52.5% 45.2 31.5 -30.2% E6 -30.50 -30.36 0.5% 9.8 6.2 -36.7% 47.9 31.8 -33.6% E8 | | | | | | | | | |
| | | | | | E10 | | | | |
| | | | | | E11 | | | | |
| | | | E12 -29.57 -29.33 0.8% 4.6 3.1 -32.6% 49.7 37.7 -24.1% E13 | | | | | | |
| E14 -33.89 -34.26 -1.1% 11.5 12.3 7.0% 53.2 37.4 -29.7% E15 -32.35 -31.94 1.3% 8.6 7.0 -18.6% 57.1 31.1 -45.6% E16 -29.06 -28.97 0.3% 7.1 4.9 -3 | | | | | | | | | |
| E19 -23.40 -23.31 0.4% 6.7 6.8 1.5% 47.7 29.4 -38.2% E20 -22.27 -22.26 0.1% 4.4 3.3 -25.0% 60.3 33.6 -44.2% E21 | | | | | | | | | |

Behaviour of snow parameters during decay events are defined. The initial, 2-day and percentage change over a 2-day period are presented for $\delta^{18}$O, $d$-e

**3.3.1 Low-wind event analysis**

**3.3.2 Isotopic change during decay events**

[Figure]

 ~~The relationship between the rate of change in SSA (ΔSSA 2*days*⁻¹) and d-excess (Δd-excess 2*days*⁻¹) over a 2-day period for a) individual samples for events presented in Table 3 for 2017 (o), 2018, (x) and 2019 (□), b) the same values colour coded by initial d-excess from each event. c) shows the relationship between change in d-excess after 2-days plotted against the initial d-excess value, with the linear regression line in black.~~

**Figure 6.** Isotopic change analysis for low-wind events, E10 and E11. Panel a) shows daily change in $d$-excess against change in d18O for E10 and E11 with corresponding linear regressions, b) shows change in $d$-excess against change in SSA, and c) shows change in d18O and change in SSA. The r- and p-value for each regression are indicating in the corresponding colours.

[Figure]

**Figure 7.** Latent heat flux (LE) (grey), relative humidity with respect to ice (purple), air-surface temperature gradient (TG) (red) and surface-10 cm subsurface TG (red) for the low-wind SSA decay events, E10 (a) and E11 (b) (Table Dark grey shading in LE indicates sublimation and light grey shows deposition.

~~Using all 10 sample sites as independent values, the behaviour of isotopes during defined SSA decay events is analysed. To determine the isotopic change in the surface snow during rapid SSA decays, the rate of change in d-excess is plotted against the rate of change in SSA (Fig. 5). The change in SSA over a 2-day period is used. The daily mean change over the first 48h of each event is presented in Table 3.~~

 As mentioned in section 3.1, ground fog preceded the SSA peak in E11, conccurent with negligible accumulation recorded. In contrast, approximately ~~1±1‰ mean increase, with the exception of E17 and E21 in 2019 (See Table 3). E17 is characterised by significant ground fog and snowfall during the event, while E21 has negative LHF (net-deposition) measured from the eddy-covariance system over the event. The percentage change of d-excess is an order of magnitude higher than $\delta^{18}$O - expected due to the definition of d-excess - and similar to SSA , with 14 out of 19 events showing a decrease in d-excess during the first 2-days of each event. Further analysis looks specifically at the relationship between d-excess and SSA given the coherence observed between their PCs, and the significant change observed in Table 3.~~ cm of snow was accumulated during the day prior to E10, corresponding to observation of snowfall.

 Figure 6 shows the relationship between  the daily change ($\Delta$d-excess) and change in SSA over the same time period ($\Delta$SSA)  in isotopic composition and SSA. For E10, both $\Delta\delta^{18}$O and $\Delta d$-excess and $\Delta$SSA and $\Delta d$-excess have significant negative correlations (r   -0.5, r=-0.8). ~~Thus suggests either low d-excess of deposited snow, or old snow that has been re-exposed. In addition, initial d-excess is observed to significantly influence that magnitude of d-excess change over the subsequent 48h of rapid SSA decay (Fig. 7a and b). The largest changes in d-excess corresponds to high initial d-excess values. Moreover, increases in d-excess during rapid SSA decay follow very low initial d-excess values. In summary, in 72% (78 out of 108 samples) of cases decreases in SSA correspond to a decrease in d-excess of the snow sample during the first 2-days. Moreover, the magnitude of change in d-excess during rapid SSA decay shows a weak but significant dependence on the initial d-excess signal.~~

  0.6), while no significant relationship is observed between the $\Delta$-parameters during E11. All samples exhibit negligible change (<0.7‰ ) in $\delta^{18}$O during E11.

The dominant direction of vapour flux is assessed using air, surface and subsurface (10cm depth) temperature data and LE between the snow and atmosphere. Net-sublimation is observed during both E10 and E11, with a total sum of 33.9 W m$^{-2}$ and 55.8  W m$^{-2}$ for the respective events. The LE is controlled primarily by the temperature gradient (TG)

kg$^{-1}$ 10 %

W  m$^{-2}$), corresponding a negative TG between the air and surface of 2.5°C on June 10th. A concurrent upwards vapour flux is indicated based on the TG between the subsurface and surface snow. Downwards LE flux up to 4 W m$^{-1}$ is observed each night corresponding to the transition from a negative to positive TG between the air and surface. The period between sampling on 9th June at 15:18 UTC and 10th June 10:40 UTC recorded net deposition, corresponding to significant increase in $\delta^{18}$O and decrease in $d$-excess.

The amplitude of all parameters is during for E11 compared to E10. A negative surface-subsurface TG persists throughout the first day of E11, indicating a downwards vapour flux.

**4 Discussion**

Continuous daily SSA measurements at EastGRIP during the summer season of 2017, 2018 and 2019 have enabled quantification of variations in snow physical properties due to  deposition and snow metamorphism. Understanding the relationship between rapid decreases in SSA and corresponding change in isotopic composition require clearly defined events and environmental context. Using a  set of criteria, nine SSA decay events during precipitation-free periods are defined and used to construct an empirical decay model. We firstly discuss the behaviour of SSA  decay at EastGRIP compared to existing models. The isotopic change associated with low-wind SSA decay events is then considered, in the context of sublimation, vapour diffusion and wind effects (Ebner et al., 2017; Hughes et al., 2021).

 SSA decay  at EastGRIP]

 SSA decay  at EastGRIP

The empirical decay model defined in this study accurately predicts the SSA decay of surface snow at EastGRIP over a limited time-period. We find that rapid SSA decay events are best described by an exponential function (Cabanes et al., 2002), and indicates that the crystal structure of a new snow layer is a key driver of decay rate within the defined conditions over 2-5 day periods.

Comparison with weather station data showed that decay function, in agreement with observations from Cabanes et al. (2003) . The expected temperature dependence on the SSA decay rate during events had no systematic influence from weather variables (wind speed, temperature and relative humidity). The only exception is for temperatures outside the set range for the model. Surface temperatures below -25 °C were characterised by a significantly higher background SSA (defined as the mean SSA value of the final day of decay events) (Fig. 4), indicating high background SSA due to reduced snow metamorphism is apparent during E9, where the mean air temperature is less than -20 °C, which is in agreement with the accepted knowledge that snow metamorphism is slower in colder conditions . This observation is supported by theory and observation that due to sublimation and deposition are being thermally activated processes (Cabanes et al., 2003). Taillandier et al. (2007) (T07) developed an SSA decay model with a surface temperature parameter in addition to initial SSA which is able to capture the behaviour of decay during the cold event, E7, at EastGRIP suggesting temperature is important to consider when predicting SSA outside the defined temperature range. However, (Cabanes et al., 2003; Legagneux et al., 2003; Flanner and Zender, 2006; Taillandier et al., 2007) . The narrow temperature range of SSA decay events does not facilitate a conclusive definition of a temperature-dependent decay rate.

In addition, we focus on the influence of temperature on wind-speed of the SSA decay rate within the defined temperature range is negligible. Model-observation comparisons show equal performance for the SSA decay model from this study ($r^2$ = 0.89) compared to T07 temperature gradient metamorphism model ($r^2$ = 0.9). and observe a more rapid SSA decay with increased wind-speed, potentially due to increased ventilation of saturated pore air acting as a catalyst for snow metamorphism (Cabanes et al., 2003; Flanner and Zender, 2006; Neumann and Waddington, 2004). Wind erosion cannot be definitively ruled out due to dis-continuous documentation of surface conditions. However, in some cases, high wind speeds are documented to increase SSA due to fragmentation and sublimation of suspended snow grains, which are then re-deposited and effectively sieved into the pore spaces of the surface snow layer (Domine et al., 2009).

The top 1 cm of the 2.5 cm SSA sample is measured by the Ice Cube device, and thus, is most likely to capture the precipitation signal (Gallet et al., 2009; Klein, 2014). Directly after precipitation, isothermal snow metamorphism is expected to be dominant due to to high surface curvature of fresh snow crystals (Colbeck, 1980). Alternative SSA decay models are proposed by Taillandier et al. (2007) to describe snow metamorphism under temperature gradient (temperature driven recrystallisation) and isothermal (curvature driven recrystallisation) metamorphism, with the surface temperature and initial SSA being variable parameters. Comparison to existing physical models allows for the assessment of the additional influence of wind-speed, not considered previously (Flanner and Zender, 2006). However, we find that all events are most accurately predicted using the temperature gradient decay equation, which accounts for the very low surface temperature observed in E7. The similarity in prediction for -25 °C to 0 °C suggests the EastGRIP SSA decays are not only driven by crystal curvature but by temperature gradient vapour diffusion as well.

 FZ06 most predicts the moderate-wind events with the lowest error. This is potentially due to the  initial conditions for low-wind event E10 likely corresponding to surface hoar, while the models from the literature tend to describe SSA decay  from precipitation. The initial SSA value of $46\,\mathrm{m^2\,kg^{-1}}$ for E10 is in agreement with documented SSA of surface hoar (Domine et al., 2009).

**4.1**

~~Conditions for the model are expected to be applicable over the Greenland Ice Sheet interior under mean summer conditions. The model predicts decay events at EastGRIP with a $r^2$ of 0.89, compared to observation, within defined conditions. SSA estimates from satellites have previously been compared to ground observations and show a strong correlation between daily mean SSA and satellite retrieved SSA at EastGRIP (Kokhanovsky et al., 2019). The SSA decay model has the potential to predict SSA decayGreenland Ice Sheet, the model can be evaluated for different sites to document the spatial variability in SSA over the entire ice sheet, and describe the summer SSA decay. This has additional benefits for quantification of surface mass balance and surface energy budget due to the relationship between snow microstructure and surface albedo.~~ following the methods in Kokhanovsky et al. (2019) would be an interesting future study, but is outside the scope of this manuscript.

**4.1  Inter-annual variability**

~~In this study, processes driving snow metamorphism are documented to influence isotopic composition of the snow after precipitation, supporting experimental observations and theoretical understanding (Ebner et al., 2017; Wahl et al., 2021; Hughes et al., 202 . Results from this study suggest that surface snow metamorphism following precipitation events corresponds to change in isotopic composition, most clearly observed in d-excess (Table 3)~~ The surface snow over the 90 m sampling transect is often non-homogeneous due to uneven distribution of accumulation. EOF analysis is used to account for spatial variability at each site, and to determine covariance between the parameters SSA, $\delta^{18}O$ and $d$-excess. The positive mode of PC1$_{SSA}$ is associated with depositional events, such as precipitation, surface hoar formation, and wind-fragmented snowdrift, causing an increase in SSA (Domine et al., 2009), while the negative mode is associated with snow metamorphism or wind scouring (Cabanes et al., 2002, 2003; Legagneux et al., 2003, 2004; Taillandier et al., 2007; Flanner and Zender, 2006). Based on  this interpretation, correlations between PC1$_{SSA}$ and PC1$_{d-excess}$ or PC1$_{\delta^{18}O}$ suggests the aforementioned mechanisms controlling SSA variability also influence the isotopic composition.

Accumulation intermittency and temperature conditions are proposed as a potential explanation for the change in regime from a coherence between $PC1_{\delta^{18}O}$ and $PC1_{d-excess}$ in 2018 and $PC1_{SSA}$ and $PC1_{d-excess}$ in $\delta^{18}O$ and decreases in d-excess ~~(Madsen et al., 2019; Hughes et al., 2021; Wahl et al., 2021). However, sublimation is not the only process occurring. Vapour pressure gradients due to surface curvature drive snow metamorphismvia vapour diffusion through the pore space and thus, kinetic fractionation is expected to influence the isotopic composition. A larger influence is expected for d-excess than $\delta^{18}O$ because kinetic fractionation influences $\delta$D more than $\delta^{18}O$ ($d-excess = \delta D - 8 \cdot \delta^{18}O$) with a stronger influence on d-excess than $\delta^{18}O$, which can explain the covariance between d-excess and SSA observed most clearly during 2019 (Cappa et al., 2003; Da . Our approach tochange over a 2-day period instead of daily change allows for increased propagation of the isotope signal during SSA decay to account for the 1~~strong correlation between $PC1_{SSA}$ and $PC1_{d-excess}$ can be attributed to increased surface exposure and warmer temperatures facilitating snow metamorphism, in agreement with findings from Casado et al. (2021).

Low accumulation during 2017 presents a caveat to this interpretation, with results from 2017 showing $PC1_{d-excess}$ to be influence by both $PC1_{SSA}$ and $PC1_{\delta^{18}O}$ during different periods. The period from May 15th to June 10th follows the regime observed during 2018 and corresponds to a negligible temperature gradient between the air, surface, and 10cm ~~bulk isotope measurements (Gallet et al., 2009; Klein, 2014). A significant relationship is observed between change in d-excess and change in SSA during the first 2-days compared to daily analysis (with an additional relationship observed during 2019 between daily change in d-excess and daily change in SSA). Decreases in d-excess are observed during rapid SSA decay, driven by a combination of sublimation, deposition and vapour diffusion through the pore space.~~

~~Surface snow metamorphism is not confined to rapid SSA decreases, and thus isotopic compositionchange is observed continuously. However, results from this study indicate that d-excess changes during rapid SSA decay have significantly different distribution than the background non-event fluctuations. Our findings are in agreement with a study from Antarctica which showed a significant relationship between d-excess and physical snow properties with depth, while negligible relationship was observed for $\delta^{18}O$ (Dadic et al., 2015). Our study has selected rapid SSA decays fitting todecay model to address how changes in snow crystal morphology after precipitation relates to change in isotopic composition. Future studies would benefit from using isotope flux models to account for the influence of sublimation and deposition, to determine unexplained isotopic composition change~~ air, surface, and subsurface. $PC1_{d-excess}$ covaries with $PC1_{SSA}$ during this period, much like 2019, suggesting that vapour diffusion driven by temperature gradients modifies the

snow isotopic composition. This agrees with previous studies documenting kinetic effects during snow grain growth resulting from pore space diffusion (Neumann and Waddington, 2004; Casado et al., 2016; Ebner et al., 2017; Casado et al., 2021).

An additional feature supporting the observation of processes driving surface snow metamorphism corresponds to a decrease in d-excess, is a clear relationship between substantial increases in SSA and increase in d-excess (Fig. ??) . The upper 10th percentile of $\Delta$SSA increases $(14.7\,m^2\,kg^{-1})$ corresponds to positive $\Delta$d-excess in 70% of cases (Fig. 7). Large increases in SSA are closely associated with precipitation, however, increases are observed in The opposing phases of the North Atlantic Oscillation (NAO) between the years can explain the different meteorological conditions. The NAO is in a number of other scenarios (Domine et al., 2009). Precipitation is expected to cause the largest SSA, suggesting that the d-excess of precipitation is most often higher than existing surface snow. Our results therefore suggest that the precipitation isotopic composition signal is not always preserved after snow metamorphism due to (kinetic) fractionationduring sublimation and other surface processes. positive phase during 2018 and the majority of 2017 bringing below-average temperatures, as observed at EastGRIP (Hanna et al., 2015). The opposite is observed during 2019, corresponding to a positive phase in the NAO.

*Change in d-excess per day ($\Delta$d-excess $day^{-1}$) vs. change in SSA per day ($\Delta$SSA$day^{-1}$)*The relationship between the rate of change in SSA per day ($\Delta$SSA$day^{-1}$) and d-excess ($\Delta$d-excess$day^{-1}$) for all summer seasons 2017-2019 (light grey), all events (dark grey) and selected events based on substantial accumulation (dark turquoise). The box indicates the values corresponding to daily decrease in d-excess during decrease in SSA, with 81% of selected events in this quadrant. Conclusive results from EOF analysis are limited by wind-effects, especially in the negative phase, corresponding to decrease in SSA, where wind scouring potentially removes the surface layer (Domine et al., 2009; Flanner and Zender, 2006; Hachikubo et al., 2014) . Decoupling snow metamorphism from wind scouring is considered in the following section on isotopic change during low-wind SSA decay events.

**4.2**

**4.2 Isotopic change during SSA decay events**

Three key mechanisms are expected to drive the rapid SSA decays, 1) large grains growing at the expense of small grains (Legagneux et al., 2004; Flanner and Zender, 2006), 2) diffusion of interstitial water vapour (Colbeck, 1983; Ebner et al., 2017; Touzeau et , 3) sublimation due to the wind ventilating the saturated pore air, known as 'wind-pumping' (Neumann and Waddington, 2004; Town et al., . The dominant mechanisms can theoretically be identified by a combination of the change in isotopic composition - indicating the fractionation effect - and the LE and temperature gradient data.

In theory, mechanism 1) causes minimal change in the bulk isotopic composition of a snow layer under isothermal conditions (Ebner et al., 2017). Therefore, observations of SSA decay corresponding to negligible isotopic composition change could be explained by this mechanism. We observe no events with consistent isotopic composition throughout. In the instance of 2) interstitial diffusion, light isotopes are preferentially diffused, while the heavy isotopes will be preferentially deposited onto the cold snow grains (Colbeck, 1983; Ebner et al., 2017; Touzeau et al., 2016). Thus, diffusion of water vapour in the

660     pore space causes a decrease in $d$-excess and slight increases in $\delta$18O due to kinetic fractionation (Flanner and Zender, 2006) . 3) Sublimation has been widely documented to cause an increase in $\delta^{18}$O of the remaining snow-mass due to equilibrium fractionation, and a significant decrease in $d$-excess due to kinetic fractionation (Ritter et al., 2016; Madsen et al., 2019; Hughes et al., 2021 .

    An overall increase in $\delta$18O and decrease in $d$-excess during E10 can be attributed to a combination of 2) and 3) based on

665 observation of net-sublimation and high amplitude diurnal temperature gradient variability indicating vapour transport within the pore space. The period between 9th June at 15:18 UTC and 10th June 10:40 UTC recorded net deposition corresponding to an overall decrease in $\delta$18O during

670  the first day  and minimal decrease in $d$-excess, potentially due a deposition of atmospheric water vapour (Stenni et al., 2016; Feher et al., 2021; Casado et al., 2021) .

    A 30% decrease in $d$-excess corresponds to a  negligible change in $\delta$18O during

675 E11. Net-sublimation double that of E10 is measured, but with reduced amplitude in both TGs. Moreover, the largest decrease in $d$-excess occurs after the first day when the surface-subsurface TG is consistently negative, indicating that vapour diffusion plays a role in modifying the isotopic composition, and the effect of equilibrium fractionation during sublimation from the surface only weakly influences the bulk isotopic composition over the 3-day period (Casado et al., 2021). Decoupling the influence of atmosphere-surface exchange and diffusion from subsurface snow requires additional measurements of isotopic

680 composition of atmospheric water vapour and precipitation isotopes, which is outside the scope of this study.

**4.3**

685 ~~for spatial variability at each site, and a coherence is observed between the principal components of d-excess and SSA. PC1 is weaker when spatial variability is high, and during these periods the coherence between d-excess and SSA are muted. During the start of 2017 and 2018 PC1 of d-excess is coherent with PC1 of $\delta^{18}O$, and decoupled from PC1 of SSA. At the start of the season, the 2.5 cm sample will contain winter snow layers which are less influenced by snow metamorphism (Libois et al., 2015; Town et al., 2008), and thus, a coherent signal between d-excess and $\delta^{18}O$ is observed. The transition~~

690

**4.3**

**4.3 Implications and perspectives**

695  Our results suggest that processes driving snow metamorphism modify the isotopic composition of

700  the snow while exposed at the surface, supporting experimental observations and theoretical understanding (Ebner et al., 2017; Wahl et al., 2021; Hughes et al., 2021). We find that $d$-excess is mostly influenced by vapour fluxes in the pore space, driven by temperature gradients. Net-sublimation appeared to have less influence on the isotopic composition, but this is expected to be  due to the depth of the sample and the short duration of both low-wind events.

705

~~Seasonal signals are influenced by millennial scale insolation variability (Masson-Delmotte et al., 2006; Laepple et al., 2011). An inverse relationship is observed between obliquity and d-excess over the past 250 ka years at Vostok which is attributed to the insolation gradient between high and low latitudes causing increases moisture transport from low latitudes relative to high latitudes (Vimeux et al., 2001, 1999). Results presented in our study document decreases in snow d-excess during surface~~

710

715  The findings of this exploratory study reiterates the importance of quantifying the isotopic fractionation effects associated with processes driving snow metamorphism during precipitation free periods. Moreover, the inter-annual variability observed at EastGRIP between 2018 and 2019 suggests that precipitation intermittency and temperature (gradients) play a role in isotopic change, which is not so readily identified in the surface snow SSA data. Future work to decouple the processes driving change in  $d$-excess (sublimation from surface or interstitial vapour diffusion in the pore space) is vital for modelling the

720 change in isotopic composition down to the close-off depth in the firn (Touzeau et al., 2018; Neumann and Waddington, 2004).  Future studies would benefit from obtaining direct measurements of the isotopic composition  of precipitation and surface hoar, to determine the fraction of  such deposits in the SSA samples. Furthermore, a quantitative representation of vapour fluxes in the surface snow would provide a basis from which to quantify the the relative influence of fractionation during sublimation and interstitial diffusion.

**5 Conclusions**

This study addresses the rapid SSA decay driven by surface snow metamorphism. In particular, the study aims to explore how rapid SSA decay relates to changes in isotopic composition of the surface snow in the dry accumulation zone of the Greenland Ice Sheet. Ten individual snow samples were collected on a daily basis at EastGRIP in the period between May and August of 2017, 2018 and 2019. SSA and isotopic composition was measured for each sample. Periods of snow metamorphism after  deposition events are defined using SSA measurements to extract periods of rapid decreases in SSA.

An exponential SSA decay model ($SSA(t) = (SSA_0 - 26.8)e^{-0.54t} + 26.8$ $SSA(t) = (SSA_0 - C)e^{-\alpha \cdot t} + C$) was constructed to describe surface snow metamorphism under mean summer conditions for polar snow, with surface temperatures above -30 °C  Two categories were defined to assess the influence of wind-speed on the SSA decay rate. The relationship between defined events of snow metamorphism and corresponding snow isotopic composition was then explored.

 Changes in isotopic composition corresponding to post-depositional processes driving  SSA decay is observed in all events. Over the first  2-days of SSA decay events,  d-excess is observed to decrease  from the initial . Analysis of SSA decay events with consistent low wind speed indicates that the combined effects of vapour diffusion and diurnal LE variability causes isotopic fractionation of the surface snow in the absence of precipitation.

[revised manuscript text omitted]

---

## Referee Report (RR1)

**Second review of:**

**Exploring the role of snow metamorphism on the isotopic composition of the surface snow at EastGRIP**

By Harris Stuart et al.

**General comments**

While again, this approach to link snow metamorphism to isotopic change is a great idea, I am skeptical about the way it is implemented. Again, my reservations are based on the fact that the samples used for SSA measurements (1 cm surface layer) and those used for isotopic measurements (2.5 cm surface layer) are not the same. In this revised version, the Authors do take care to some extend of surface processes by selecting low wind events where snow drift and surface perturbations may have been absent or minimal. However, even the low wind events feature surfaced perturbations. Event E10 was a 1 cm thick snowfall, and event E11 was a thin fog deposit. The surface layer sampled for SSA was then a different layer that the lower 1.5 cm, included in the isotopic sample.

Furthermore, the protocol is not explained clearly. It is really very confusing and figuring out what was done is a headache, at least for me. Some crucial elements are given too late in Results rather than in Methods, Table A1 is incomplete and therefore not very useful. Perhaps the isotopic data should be analyzed differently, by considering a surface layer with rapid changes and a lower layer with low changes. This would however add some uncertainty. I am therefore not sure what to recommend, because it is quite clear to me that the sampling protocol is just not adequate and inevitably skews the results, possibly irremediably.

This is unfortunate, because the topic has great potential. Time series of surface snow SSA and isotopic compositions are highly valuable, but correlating them requires both samples to be identical. In that line, I do not know what to make of the EOFs of Figure 3. I may have missed something, but were all the SSA and isotopic data used to obtain them? Surely some serious filtering would have been mandatory here.

Anyway, I am really undecided here. On the one hand, the experimental data does have some value as those time series are unique and deserve publication in some form. On the other hand, the interpretation and the correlations seem extremely weak to me. The Authors failed to sufficiently stress the enormous caveats in their protocol, and their conclusions have a very weak basis that may in fact be misleading. Perhaps the Authors could stress their caveats and tone down their interpretation to present their data in what I feel would be a more honest and humble manner. I am just trying to help here, by the way.

**Specific comments**

Line 42-43. Decrease in SSA is explained by Ostwald ripening only under almost perfect isothermal conditions, which is rarely the case here. How about "Decrease in SSA in dry snow is predominantly the result of water vapor transfer among grains, with smaller grains feeding

the growth of larger grains. Ventilation by wind can accelerate SSA decrease by enhancing water vapor transfer rates.". This statement also applies to Ostwald ripening but is much more general. Ostwald ripening is very specific.

Line 50-51. Stating that "Exponential models are documented to produce the best fit to in-situ SSA decay data". is probably exaggerated, as equations given in (Legagneux et al. 2004) and also used in the models of (Flanner and Zender 2006) probably are more accurate. Perhaps say something like "Are the most convenient to account for temperature effects under various wind conditions".

Line 52. Should be (Legagneux et al. 2004).

Line 107. "The e-folding depth of 1310nm radiation in snow of 200 kgm−3 is approximately 1 cm". This is true for a given value of SSA. Please specify which SSA value and perhaps give details regarding the impact of the SSA value on the e-folding depth. Calculations can be done done e.g. using https://snow.univ-grenoble-alpes.fr/snowtartes/

Line 126. The SSA of surface hoar is usually moderate to low, so that surface hoar formation can often lead to SSA decrease.

Section 2.5.1. It is not clear which SSA data were finally filtered out. Many confusing elements are given but no clear criterion is given in the end. So, were the events with wind speed <6 m/s kept and those >7 removed? Then there are moderate wind events. What was done with the SSA data in that case? This must be written simply and clearly. Some elements of section 3.2.1 should probably be placed in this methods sections, as they are not results.

Lines 194-200. Were all the SSA data used for the EOF? But some are unreliable, I think. This all needs to be clarified.

Line 147. Cabanes 2002 and Cabanes 2003 should be swapped.

Equation (4). Is that for isothermal or temperature gradient conditions?

Line 166. Not sure what signal attenuation means here.

Line 187-188. "Throughout the season $\delta^{18}O$ follows a gradual increasing trend from May to August following increasing temperatures." This may be overly simplified. There are drops

in $\delta^{18}O$, especially at the end of all seasons, while temperature does not drop. This is mentioned later, but nevertheless this statement is not warranted.

Line 221. It thus appears that even E11 is characterized by a thin fog deposit. Therefore, the SSA decay rate may pertain to the top few mm. Comparing its evolution to the isotopic evolution of the 2.5 cm thick isotope snow samples may then be meaningless.

Section 3.2.2 Which events are used to construct this model? This should be clearly mentioned and added to Table A1. Line 303 mentions 6 events were used, but we need to know that now.

Line 237. I am not sure the value 22 $m^2$/kg can be called a decay constant. A decay constant is expected to have $s^{-1}$ in its units, I would think.

Line 314. Please specify the temperature range.

Section 4.1. Please note that both T07 and FL06 ignore wind speed as a variable. Those studies are all based on data obtained under no wind or low wind speeds. Since wind speed accelerates SSA decay, as first noted by (Cabanes et al. 2002), it is not surprising that both T07 and FL06 underestimate the decay rate observed by the Authors under non-negligible winds. This may be explicitly mentioned.

Lines 333-338. I do not understand the difference between mechanisms 1 and 2. Large grains grow at the expense of small grains by water vapor diffusion, so I just do not see the difference with diffusion of interstitial vapor. Then the following discussion may need to be significantly rewritten. It may be somewhat more sensible to consider the temperature gradient. Elevated gradients drive fluxes throughout layers while low gradients mostly involve short distance vapor transfers less likely to result in isotopic changes. Wind pumping is more likely to result in the largest changes since this is where the exchanges of vapor with the atmosphere will be greatest. In any case, looking at Figure 2, my guess is that wind pumping will be important in all cases so that this will always be the predominant process. I therefore have the feeling that the Authors are on the wrong track and that their classification of events is inadequate. I would be more tempted to consider other criteria, such as perturbations of the snow surface, where the sign of LE would come in.

Line 340. Reference to Ebner is incorrect. The discussion paper is mentioned.

Line 357-359. "We conclude that SSA of the surface snow is strongly influenced by surface-subsurface TG while the changes in isotopic composition are likely to be influenced by other factors such as the magnitude of vapour-snow isotopic disequilibrium during sublimation". The Authors may be right here. But the picture would probably be clearer if the same samples

had been used to measure isotopes and SSA. The different sampling depths really skews the results and make any interpretation uncertain.

Line 383-385. Please consider that given the windy context, wind pumping is a much more efficient process that temperature gradient-induced diffusion to produce exchanges of water vapor and isotope fractionation in the surface snow.

Table A1 should have an extra column indicating wind conditions.

**References cited**

Cabanes, A., L. Legagneux, and F. Domine, 2002: Evolution of the specific surface area and of crystal morphology of Arctic fresh snow during the ALERT 2000 campaign. *Atmos. Environ.*, **36,** 2767-2777.
Flanner, M. G., and C. S. Zender, 2006: Linking snowpack microphysics and albedo evolution. *J. Geophys. Res.*, **111,** D12208.
Legagneux, L., A. S. Taillandier, and F. Domine, 2004: Grain growth theories and the isothermal evolution of the specific surface area of snow. *J. Appl. Phys.*, **95,** 6175-6184.

---

## Author Response (AR4)

**Editor's comments**

One concern from a reviewer was raised about the correlations presented in Figure 6b and c: Although the 2.5cm SSA sample of snow is reused for isotope analysis, Event 10 followed 1cm accumulation of snow, so it is this surface layer plus older snow that was sampled, with the SSA weighted towards the fresher snow but the isotope fractionation measurement of the whole 2.5cm sample. What are the implications of this for the correlations presented in Fig 5 and Fig 6? It might be worth quantifying in Table A1 the amount of snowfall for each event. Please could you discuss this limitation in the paper and could you make any recommendations for future analysis e.g. using x-ray tomography?

We thank the Editor for these comments, and we have tried to incorporate these suggestions in the individual responses. To respond directly to the questions from the Editor; quantifying the amount of snowfall (or rather accumulation to include snowdrift) would definitely be useful, however there are a couple of complications associated with this. Firstly, each manual snow height measurement has a 0.5 cm uncertainty, which is often more than the amount of accumulated snow before the SSA decay events. Secondly, the mean accumulation over the transect may be unrepresentative of the accumulation at individual sites. Bearing this in mind, we have added the mean accumulation to Table A1 in the manuscript.

The Editor's second point regarding greater clarity of the limitations of the study is addressed in the responses to the Reviewers. Following this suggestion, we found a recent paper from Martin and Schneebeli (2022) which compares the SSA measurements from the Ice Cube device and x-ray tomography. They show that considerable discrepancies between the devices correspond to SSA values below 25 $m^2$ $kg^{-1}$. Fortunately, the vast majority of our measurements fall above this value, and we are therefore confident in our Ice Cube measurements.

**2nd Response to Reviewer 1**

We want to thank the reviewer for agreeing to review the manuscript for the second time, and for the contributions to improving the study. We appreciate the time taken by the reviewer to detect and correct the grammatical and technical mistakes, which we had overlooked.

In the responses to the reviewer's comments, the blue text that follows is in response to the reviewer's suggestions and the italic text indicates proposed changes to the text in the manuscript. All line numbers in blue refer to the updated manuscript.

**Reviewer 1:**

The manuscript has significantly improved. The only area that needs clarification in my opinion is the separation from results to discussion in regard to the model performance. I think that the model performance comparison comes fairly early (too early) in the results and the discussion (4.1) draws on that a lot later to finally discuss reasons for their differences. Aside from that, I list a couple of minor language mistakes or suggestions. I suggest you go through the manuscript for spelling and grammar consistency.

We acknowledge the reviewer's suggestion regarding the separation of results and discussion. However, we decided to strictly separate the results from the discussion for the purpose of clarity.

We propose instead to slightly modify the model performance discussion section to help link back to the results section. Please see a comment below (Line 244) for the proposed text.

- Line 34: Since you use second order as an attribute here, please correct to "second-order" parameter.

Line 32-33: "Here we focus on processes influencing isotopic composition of the surface snow after deposition while exposed to surface processes and concentrate on the second-order parameter d-excess."

- Line 37: I'd say, surface snow metamorphism is not strictly driven by "a reduction in the snow-air interface to reach thermodynamic stability" but characterized by it. A different verb might be more fitting.

Line 35-37: "Surface snow metamorphism is initially characterised by a reduction in the snow-air interface in order to reach thermodynamic stability (Colbeck, 1980; Legagneux and Domine, 2005)."

- Line 39: I think that no comma is necessary after (Gallet et al. 2009).

The comma has been removed (Line 38).

- Line 43: The correct spelling is Ostwald ripening (without a capital R)? Please check and correct throughout the manuscript.

The reference to Ostwald ripening has been removed in the text based on a suggestion from the second reviewer. The new text is as follows:

Line 41-44: *"Decrease in SSA in dry snow is predominantly the result of water vapour transfer among grains, with smaller grains feeding the growth of larger grains (Legagneux et al., 2004; Flin and Brzoska, 2008; Sokratov and Golubev, 2009; Pinzer et al., 2012). Ventilation by wind can accelerate SSA decrease by enhancing water vapour transfer rates (Picard et al., 2019)."*

- Line 60: "The latter is of particular interest owing to observations…" – This sentence construction is a bit confusing to me, and I don't exactly get the "why have lab studies sparked interest?". Maybe you can be a bit more specific and rephrase?

We agree that this construction is not clear and is quite fundamental to the motivation for the paper. We propose to modify the text as shown below.

Line 58-64: "*Continuous datasets of daily SSA and corresponding isotopic composition measurements from the accumulation zone of the Greenland Ice Sheet are required for understanding the influence of surface snow metamorphism on surface energy budget (Picard et al., 2012), and for the interpretation of ice core water isotope records (Casado et al., 2021; Wahl et al., 2022). In this study we focus on the latter, which is of particular importance owing to observations of isotopic fractionation during snow metamorphism documented in laboratory studies (Ebner et al., 2017) and field experiments (Steen-Larsen et al., 2014; Casado et al., 2021; Hughes et al., 2021; Wahl et al., 2021). Nonetheless, few studies have focused on the direct relationship between physical snow properties, such as SSA, and post-depositional changes in isotopic composition.*"

- Figure 1: Could you please include a scale for the map in panel a?

A scale has been added.

- Line 96: You could refer to Figure 1d as well.

Reference to Figure 1d has been added (Line 98).

- Line 102: 9-days needs to be corrected to 9 days.

The hyphen has been removed from "9-days" (Line 104).

- Figure 4: "followed by the modelled SSA decays for the respective events in d) and c)." should be corrected to d) and e). and all captions should end with/without a full stop (please check author guidelines).

We apologise for the oversight of the author guidelines and have modified the captions to all end with a full stop. The letters have also been corrected.

- Line 181: Are you referring to differences between the seasons 2017, 2018 and 2019 or to changes during each with time? I'd change the phrase "Seasonal variability" to something a bit clearer, as this is the start of the chapter, and I don't know what you will tell yet.

Seasonal variability is used to describe the variability in each variable within one field season. The text has been modified to explicitly state what we mean by seasonal variability.

Line 198-200: "*Spatial and temporal variability – defined as the daily standard deviation over transect and the standard deviation of the daily mean values over the season respectively – is observed in SSA, δ18O and d-excess throughout the field seasons of 2017, 2018 and 2019 (Fig.3), with highest daily spatial variability in isotopic composition.*"

- Line 233: "Solving the differential with respect to time (t)" – Insert differential "equation"?

The text has been modified accordingly (Line 245).

- Line 244: I'd call the chapter "Model performance comparison" or similar to clarify that you compare here. I also suggest including the comparison aspect of this section in the discussion, because it contextualises your work and you discuss reasons for the performance there.

The title has been changed to 'Model performance comparison'. We propose to add the following text in the discussion (Section 4.1) to connect the results and discussion, without modifying the structure of the manuscript.

Section 4.1

Line 329-335: "*Unsurprisingly, given the parameters are fit to the data, the model defined in the study predicts observed SSA decay for the low- and moderate-wind events with the lowest RMSE. T07, Eq. 4 from (Taillandier et al., 2007), underestimated the observed SSA decay rate in all the low- and moderate-wind events, except for E18. The largest error is associated with E1, which also had the highest mean wind-speed (6.9 m s$^{-1}$) of all analysed events. The tendency for T07 to*

*underestimate the observed SSA decay can be explained by the additional influence of wind-speed which accelerates SSA decay (Cabanes et al., 2002), which is not considered in either T07 or FZ06. In contrast, FZ06 consistently overestimates the observed SSA decay rate, most pronounced in E10 and E18. The original parameter values τ and n of FZ06 were tuned to data from alpine regions, potentially explaining the poor fit."*

- Line 247: "Taillandier et al. (2007), hereafter T07, as defined in Section 2.5.2. Residuals between our model and the observations are normally distributed, suggesting no systematic errors in model predictions." I don't exactly understand the sentence structure.

This statement refers to the fact that our model shows no clear systematic offsets or biases compared to the observations (Line 261).

- Line 249: Here, a comma is needed after wind speed rather than a full stop, and please check the entire manuscript for consistent spelling of wind speed when used as a subject/object (and wind-speed when used as attribute).

We apologise for the lack of consistency and have modified the text accordingly.

- Line 277: "this is to ensure that surface layer " – Please insert "the" before surface layer.

This has been corrected in the manuscript (Line 288).

- Line 287: "Net-sublimation" – Please change to net sublimation (same as for net deposition in line 293) throughout the manuscript (same reason as for wind speed, comment line 249)

The entire text has been checked for the inclusion or exclusion of hyphens in these instances (Line 297).

- Line 300-307: Here, I first wondered whether you described what you had done in the earlier results chapter or what you will do in the discussion. Could you state more clearly how the discussion chapter takes the interpretation of results to the next level? It might be helpful to discriminate between the measurement results description and their discussion a bit more clearly. (see comment for line 244)

The opening text for the discussion is currently written as a brief recap of the aims and an overview of our approach. We propose to modify this section of text to the following:

Line 309-314: "*The new SSA and snow isotopic composition datasets presented in this study have revealed concurrent decreases in d-excess (and d18O to a lesser degree) and SSA during precipitation free periods with minimal snow drift. A simple empirical model describing the SSA decay rate under different wind regimes reveals more rapid SSA decay when wind-speeds are higher. The following sections look firstly at why existing models tend to be inaccurate when predicting in-situ SSA decay at the surface, followed by a second section discussing the possible mechanisms driving the relationship between SSA and d-excess during precipitation-free periods."*

- Line 366: Reorder the years to 2017 and 2018 to be chronological

The chronology of years has been fixed (Line 383).

- 376: "variance in d-excess from that of δ18O (Fig. 3) in 2019 which is can be attributed to" - Please insert comma after 2019 and correct the grammar of the sentence.

We apologise, this error has been corrected. The suggested corrections are shown in the following text.

Line 389-391: "*We observe a decoupling of the temporal variance in d-excess from that of δ18O (Fig. 3) in 2019, which can be attributed to the d-excess signal being more sensitive to kinetic effects during sublimation and interstitial diffusion (Ebner et al., 2017; Casado et al., 2021).*"

- 379: I think what you mean is disentangle (whether the are coupled or not).

Yes, this is correct, we have replaced decoupled with disentangled (Line 392).

- 410: Maybe rephrase "Wind speed is observed to increase surface SSA decay rate" to "Higher wind speeds lead to quicker SSA decay.." to simplify the sentence structure.

This sentence has been modified to the following:

Line 423-424: "*Higher wind speeds increase the SSA decay rate due to enhanced snow metamorphism with increased ventilation of the pore space.*"

- The setting of acknowledgments and references is ruptured by the setting of appendix figures. This needs checking before publication.

We will be sure to correct the setting of the appendices etc. before publication. Thank you for pointing this out.

- Data availability: Is a DOI assigned yet? Otherwise, please rephrase.

The DOI's have now been added.

-Appendix figures: Could you provide more detailed figure captions for A4 and A6? That would aid the key messages from each of them when going through the appendix separately.

"*Figure A4. The change in SSA after a day (Day-1 to Day-0) plotted against the absolute value from Day-0 for all 21 SSA decay events (E1-E21 in Table A1) captured by the decrease threshold described in Section 2.3. The markers are coloured by mean air temperature between samplings (i.e., the mean air temperature from the time between sampling on Day-0 and Day-1).*"

"*Figure A6. Daily mean 2 m air temperature (red), surface temperature (yellow) and 10 cm subsurface temperature (grey). The data are presented for the 2017, 2018 and 2019 measurement campaigns.*"

**2nd response to Reviewer 2**

We firstly want to thank the reviewer for agreeing to review the manuscript for the second time, and for the contributions to improving the study. We would like to raise one point of possible miscommunication related to the SSA and snow isotopes data as we are uncertain whether there is still some confusion in this regard. To clarify, the SSA and isotopes measurements were conducted on the same snow samples. All comparisons between the isotopic composition and SSA are considering the same snow sample.

In the responses to the reviewer's comments, the blue text that follows is in response to the reviewer's suggestions and the italic text indicates proposed changes to the text in the manuscript. All line numbers in blue refer to the updated manuscript.

**General comments**

While again, this approach to link snow metamorphism to isotopic change is a great idea, I am skeptical about the way it is implemented. Again, my reservations are based on the fact that the samples used for SSA measurements (1 cm surface layer) and those used for isotopic measurements (2.5 cm surface layer) are not the same. In this revised version, the Authors do take care to some extend of surface processes by selecting low wind events where snow drift and surface perturbations may have been absent or minimal. However, even the low wind events feature surfaced perturbations. Event E10 was a 1 cm thick snowfall, and event E11 was a thin fog deposit. The surface layer sampled for SSA was then a different layer that the lower 1.5 cm, included in the isotopic sample.

We want to address the reviewers concerns in a two-fold way - 1) looking at the SSA measurements and its representativity and 2) looking at isotopes in surface snow.

In general, we have to clearly state, that technically the samples used for SSA and isotopic composition are the same (i.e. snow was sampled by taking the upper 2.5cm of surface snow, measured for SSA and then put in a plastic bag to be measured for stable water isotopes). It is exactly the same snow. This is mentioned in the manuscript in line 125.

Line 125-126: "*Individual SSA samples were put in separate bags and subsequently measured for water isotopic composition. Thus, every day the 10 SSA samples have a corresponding isotopic composition.*"

1) In our study we state that the SSA measurement is weighted towards the upper 1 cm of the 2.5 cm sample due to the e-folding depth (i.e., the depth to which the light irradiance within the snowpack is reduced to 1/e (approx. 37%) of its initial value). It is not a 1 cm sample.

Further, our study focuses on changes in SSA e.g., during decay events. Clearly, SSA measurements using different methods might lead to different values - i.e., there might be uncertainty associated with the values we present for the 2.5 cm sample. However, a recent study comparing SSA measurements from an Ice Cube device (as used for our study) and CT scans (possibly the most precise method available to quantify SSA) indicate a good agreement for SSA values over 25 m$^2$ kg$^{-1}$ (Martin and Schneebeli, 2022), and most of the presented measurements are above 25 m$^2$ kg$^{-1}$. Moreover, we found a fair comparison of our SSA measurements to for example remote sensing studies on snow grain sizes (Kokhanovsky et al., 2019; Vandecrux et al., 2022). Thus, we think that

our findings on the link between changes in SSA and the isotopic composition of snow are reliable and not influenced or perturbed by the measurement itself or the effect of the e-folding depth.

2) Stable water isotopes were measured on the snow samples used for the SSA measurements. As the SSA tool requires the sampling of 2.5 cm of snow, the isotope values are determined for the full 2.5 cm snow sample. From our observations of surface snow and its isotopic composition we can state that the 2.5 cm sample is a weakened signal of the upper 1 cm based on isotope measurements from a neighbouring transect at EastGRIP (Figure R1 using data from Wahl et al., 2021). Changes in the isotopic composition of the upper 1 cm and the upper 2 cm etc. are following the same trends and direction, with the upper 1 cm showing the strongest signal. This was also observed by Hughes et al. (2021) during an experimental study at EastGRIP. We are therefore confident that our correlations of SSA decay with isotope changes both using values representative of 2.5 cm is meaningful.

[Figure]

Figure R1. Published surface isotopes data from EastGRIP during the 2018 and 2019 field campaigns (Wahl et al., 2021). The light blue line shows the 1 cm d-excess measurements, and the dark blue line shows the 2 cm d-excess measurements. Grey bars have been added to indicate the two low-wind events from 2018.

Please see the specific comments for suggested changes in the text.

Furthermore, the protocol is not explained clearly. It is really very confusing and figuring out what was done is a headache, at least for me. Some crucial elements are given too late in Results rather than in Methods, Table A1 is incomplete and therefore not very useful. Perhaps the isotopic data should be analyzed differently, by considering a surface layer with rapid changes and a lower layer with low changes. This would however add some uncertainty. I am therefore not sure what to recommend, because it is quite clear to me that the sampling protocol is just not adequate and inevitably skews the results, possibly irremediably.

We apologise that the protocol is still unclear to the reviewer and propose edits to the text in a reply to a later comment (for Section 2.5.1). In accompanying projects, we conducted measurements of stable water isotopes from different depths (upper 1, 2, and 5 cm) (Hughes et al., 2021; Wahl et al., 2022). We therefore know that the daily changes and trends are the same in the surface and

subsurface layer (see Figure R1). Thus, the signal in a 2.5 cm snow layer shows the same but weakened changes as the signal in the 1 cm snow layer. As the SSA signal is measured at the 2.5 cm snow layer (as our isotope measurements presented here) we are confident that we do capture daily changes in the snowpack with both our measurements at the 2.5 cm snow samples.

This is unfortunate, because the topic has great potential. Time series of surface snow SSA and isotopic compositions are highly valuable, but correlating them requires both samples to be identical. In that line, I do not know what to make of the EOFs of Figure 3. I may have missed something, but were all the SSA and isotopic data used to obtain them? Surely some serious filtering would have been mandatory here.

All samples were used for the EOF analysis, and no filtering was applied to the data. This analysis was specifically used to identify relationships in both the spatial and temporal domains.

Anyway, I am really undecided here. On the one hand, the experimental data does have some value as those time series are unique and deserve publication in some form. On the other hand, the interpretation and the correlations seem extremely weak to me. The Authors failed to sufficiently stress the enormous caveats in their protocol, and their conclusions have a very weak basis that may in fact be misleading. Perhaps the Authors could stress their caveats and tone down their interpretation to present their data in what I feel would be a more honest and humble manner. I am just trying to help here, by the way.

We appreciate these suggestions from the reviewer and while we agree that the discussion would benefit from more clarity regarding the caveats in sampling strategy, we would argue that it would be difficult to refine the sampling strategy to substantially reduce the uncertainty while comparing the same snow samples. To make a direct comparison between the SSA and isotopic composition we must use the same snow cups for both measurements. Adapting the sampling protocol to e.g., removing the top 1 cm layer after the SSA measurement to analyse isotopic composition would only increase uncertainties, especially given that the profiles of SSA and density within the 2.5 cm sample would vary between sampling days.

**Specific comments**

Line 42-43. Decrease in SSA is explained by Ostwald ripening only under almost perfect isothermal conditions, which is rarely the case here. How about "Decrease in SSA in dry snow is predominantly the result of water vapor transfer among grains, with smaller grains feeding the growth of larger grains. Ventilation by wind can accelerate SSA decrease by enhancing water vapor transfer rates.". This statement also applies to Ostwald ripening but is much more general. Ostwald ripening is very specific.

We agree and appreciate this clarification. This has been changed in the manuscript as suggested.

Line 41-44 to: "*Decrease in SSA in dry snow is predominantly the result of water vapour transfer among grains, with smaller grains feeding the growth of larger grains (Legagneux et al., 2004; Flin et al., 2008; Sokratov et al., 2009; Pinzer et al., 2012). Ventilation by wind can accelerate SSA decrease by enhancing water vapour transfer rates (Picard et al., 2019).*"

Line 50-51. Stating that "Exponential models are documented to produce the best fit to in-situ SSA decay data". is probably exaggerated, as equations given in (Legagneux et al. 2004) and also used

in the models of (Flanner and Zender 2006) probably are more accurate. Perhaps say something like "Are the most convenient to account for temperature effects under various wind conditions".

This is a useful point and much better aligned with what we want to say. This has been modified in the text.

Line 49-51 to: *"Exponential models are documented to produce the best fit to in-situ SSA decay data given that they can account for temperature effects under various wind conditions while being simple in the formulation (Cabanes et al., 2003)."*

Line 52. Should be (Legagneux et al. 2004).

The reference has been corrected in the text.

Line 107. "The e-folding depth of 1310nm radiation in snow of 200 kgm−3 is approximately 1 cm". This is true for a given value of SSA. Please specify which SSA value and perhaps give details regarding the impact of the SSA value on the e-folding depth. Calculations can be done done e.g. using https://snow.univ-grenoble-alpes.fr/snowtartes/

The SSA value quoted in Gallet et al. (2009) is 35 $m^2$ $kg^{-1}$, while the mean SSA value for the three measurement campaigns at EastGRIP was 37.5 $m^2$ $kg^{-1}$. However, the snow density at EastGRIP was slightly higher with a mean value of 295 kg $m^{-3}$ over the three field campaigns. We therefore propose to modify the text in Section 2.3 to:

*Line 109-118: "Gallet et al. (2009) show that SSA measurements for snow with density of 200 kg $m^{-3}$ and SSA of 35 $m^2$ $kg^{-1}$ mostly reflect the top 1 cm of a 2.5 cm snow sample when using 1310 nm radiation, due to the e-folding depth. The properties of each 2.5 cm snow sample will determine the e-folding depth (i.e., the depth to which the light irradiance within the snowpack is reduced to 1/e (approximately 37%) of its initial value), with higher SSA and density causing a decreased e-folding depth. Given that the mean snow density from all field seasons is 293 kg $m^{-3}$ (307 ± 40 kg $m^{-3}$, 278 ± 47 kg $m^{-3}$, 294 ± 50 kg $m^{-3}$ for 2017, 2018 and 2019) and the mean SSA is 37.5 $m^2$ $kg^{-1}$, the SSA values measurement will be weighted towards the top <1 cm of the 2.5 cm sample. However, recent studies have shown that the SSA values obtained from the instrument used here (Ice Cube) agree well with measurements from computed microtomography on the same samples (Martin and Schneebeli 2022). Further, our data fairly agreed in comparison to remote sensing products (Kokhanovsky et al., 2019; Vandecrux et al., 2022). Thus, we consider our SSA values as representative for the upper 2.5 cm surface snow."*

Line 126. The SSA of surface hoar is usually moderate to low, so that surface hoar formation can often lead to SSA decrease.

The mean SSA at the start of the events is rarely higher than 60 $m^2$ $kg^{-1}$, compared to an SSA of around 54 $m^2$ $kg^{-1}$ for surface hoar (Domine et al., 2009). This would indeed be a decrease, but of a much smaller magnitude than we are focussing on (13 $m^2$ $kg^{-1}$). In any case, we propose to clarify in the text that surface hoar causes a decrease in SSA from fresh snow.

*Line 136-138: "Based on this understanding, two terms are defined:*

*1) SSA increase: Increases in SSA indicate deposition events in the form of precipitation or drifted snow.*

*2) SSA decrease: Decreases in SSA are due to snow metamorphism and other post-depositional processes such as wind scouring and, in some few cases, surface hoar formation, where the SSA decreases."*

Section 2.5.1. It is not clear which SSA data were finally filtered out. Many confusing elements are given but no clear criterion is given in the end. So, were the events with wind speed <6 m/s kept and those >7 removed? Then there are moderate wind events. What was done with the SSA data in that case? This must be written simply and clearly. Some elements of section 3.2.1 should probably be placed in this methods sections, as they are not results.

We apologise that the criteria used to define SSA decay events was unclear. The low-wind events are those where the maximum wind speed is consistently less than 6 $m\ s^{-1}$. The moderate wind events include those where maximum wind speed is between 6-7 $m\ s^{-1}$ (i.e., does not exceed 7 $m\ s^{-1}$). We propose the following modifications to the methods section taking some components from the results section 3.2.1.

*Section 2.5.1*

*Line 142-158*: "*A threshold is derived to systematically identify periods of rapid SSA decay - hereafter referred to as SSA decay events. SSA decay events captured by this threshold are defined by the peak SSA value (Day-0), through to the next increase in SSA. A set of criteria are applied to the SSA decay events to avoid events with wind-perturbed surfaces. While in Antarctica drifting of unconsolidated snow has been observed at mean hourly wind-speeds as low as 4.5 m $s^{-1}$ at 2 m (Birnbaum et al., 2010), a study from Northeast Greenland, with similar conditions to EastGRIP, documented snowdrift starting at 6 m $s^{-1}$ (Christiansen et al., 2001), due to warmer temperatures facilitating bonding of the surface snow (Li and Pomeroy 1997). Additional field-diary observations from EastGRIP document significant snowdrift when wind speeds exceed 7 m $s^{-1}$, and based on these observations, two wind-speed categories are defined.*

    1) *Low wind events: The first includes events with daily maximum wind-speed (computed from 10-minute averaged wind-speed) consistently below 6 m $s^{-1}$, hereafter referred to as low-wind events, where negligible surface perturbation is ensured.*

    2) *Moderate-wind events: A second category considers events with daily maximum wind-speed between 6 - 7 m $s^{-1}$, hereafter moderate-wind events. The inclusion of these events facilitates an assessment of the influence of wind-speed on SSA decay.*

*Subsequent isotopic analysis is first broadly applied to both low- and moderate-wind events over 1- and 2-day periods, followed by a focused assessment of isotopic change and corresponding temperature fluxes is applied to low-wind events alone given the assurance of unperturbed snow. All events with wind-speed above 7 m $s^{-1}$ are excluded from analysis.*"

Lines 194-200. Were all the SSA data used for the EOF? But some are unreliable, I think. This all needs to be clarified.

All SSA data were used for the EOF analysis. The inclusion of all 10 samples was important - and the purpose - to analyse spatial variability in snow SSA compared with isotopic composition.

Line 212-215: *"Empirical Orthogonal Function (EOF) analysis is applied to the data to identify the dominant modes of variance in both the temporal and spatial dimensions for each parameter - SSA, δ18O and d-excess. Using a confidence interval of 95% (p < 0.05), the relationships between SSA, δ18O and d-excess are tested including all 10 identical samples, covering the entire measurement period."*

Line 147. Cabanes 2002 and Cabanes 2003 should be swapped.

The references have been corrected in the text.

Equation (4). Is that for isothermal or temperature gradient conditions?

This is for temperature gradient conditions. The isothermal equation was tested and performed very poorly - as expected given that it is highly unlikely to have isothermal conditions in the top centimetres. The text in Line 159 has been modified to clarify which equation we use.

Line 175-177: *"Taillandier et al. (2007) proposed two equations based on Eq. (2) to define the decay rate under isothermal and temperature gradient conditions where they were able to directly incorporate a surface temperature parameter (Tm). Here we use the model for temperature gradient conditions (Eq. 9 in Taillandier et al. (2007))."*

Line 166. Not sure what signal attenuation means here.

We use "signal attenuation" here to explain the possibility of losing a localised signal when taking the average values over all 10 sites. This is especially important when comparing the isotopic composition measurements to the SSA. The following text is proposed for clarification.

Line 181-183: *"A recent study at EastGRIP has shown the significant heterogeneity in surface snow due to post-depositional reworking from the wind (Zuhr et al., 2021), and therefore each sample location is treated individually to avoid the smoothing out of localised signals when averaging."*

Line 187-188. "Throughout the season δ18O follows a gradual increasing trend from May to August following increasing temperatures." This may be overly simplified. There are drops in δ18O, especially at the end of all seasons, while temperature does not drop. This is mentioned later, but nevertheless this statement is not warranted.

We do observe a gradual increasing trend in δ18O for all years. It is true that there is large variability and that the rapid decreases at the end of the season are uncoupled with temperature, but instead coincide with large snowfall events. We propose to modify the statement from an indication of causation to a covariance.

Line 205-207: *"Throughout the season δ18O follows a gradual increasing trend from May to August concurrent with increasing temperatures. Cases of abrupt decreases (-10 ‰) are observed in the late summer, for example, on July 12th in 2018 and July 25th in 2019, originating from late-summer snowfall events."*

Line 221. It thus appears that even E11 is characterized by a thin fog deposit. Therefore, the SSA decay rate may pertain to the top few mm. Comparing its evolution to the isotopic evolution of the 2.5 cm thick isotope snow samples may then be meaningless.

Firstly, we should highlight that fog was observed during the day prior to the event, but that the LE data shows negligible deposition during the same period. To avoid speculation, we remove the text which indicates the presence of surface hoar, and instead simply state that the precedent day had fog and not a significant snowfall.

Line 235: "*Note that E11 was preceded by ground fog, and not snowfall (Table A1).*"

In any case, as we have shown in earlier comments, the 2 cm isotopic composition follows the 1 cm values (low-wind events E10 and E11, are shaded in Figure R1). While we agree that the persistent presence of a surface hoar layer would add uncertainty when analysing the bulk isotope measurements, we have little evidence to suggest the presence of such a layer. We propose to add a sentence to the discussion to clarify that the observations of fog do not necessarily correspond to surface hoar formation.

Add to Section 4.2.1

Line 363-366: "*Although ground fog was documented on the day preceding E11, no significant deposition is observed in the LE data in the day preceding E11, indicating the absence of lasting surface hoar formation. The 30% decrease in d-excess concurrent with no change in δ18O suggests strong kinetic fractionation during E11.*"

Section 3.2.2 Which events are used to construct this model? This should be clearly mentioned and added to Table A1. Line 303 mentions 6 events were used, but we need to know that now.

E10 and E11 were used to construct the low-wind scenario model, while E1, E13, E18 and E19 were used for the moderate wind scenario. We make the following edits to the text, as well as stating this in Table A1.

Add to Section 3.2.2

Line 242-243: "*SSA decay rate is quantified by plotting the rate of change in SSA per day against the absolute SSA value for all 10 sampling sites for low- (E10 and E11) and moderate-wind (E1, E13, E18 and E19) events (Fig. 4a).*"

Line 237. I am not sure the value 22 m2/kg can be called a decay constant. A decay constant is expected to have s-1 in its units, I would think.

We agree that decay constant is the incorrect term. Instead, we use offset to describe the lowest SSA values or background SSA.

Line 249-251: "*Where SSA(t) is the SSA measurement at a given time in days since the first measurement (initial SSA), $SSA_0$ is the initial SSA value, and α is the decay rate. An offset of 22 $m^2$ $kg^{-1}$ is required to account for the non-zero asymptote and is defined as the SSA value where the derivative of SSA is equal to 0 $m^2$ $kg^{-1}$.*"

Line 314. Please specify the temperature range.

The temperature range has been added based on the mean temperatures of the low- and moderate-wind events. The text has been updated as follows:

Line 336-337: "*The simple empirical model presented here is limited to conditions at EastGRIP within a narrow temperature range (-18◦C to -7◦C) and therefore might be unsuitable for sites with different conditions.*"

Section 4.1. Please note that both T07 and FL06 ignore wind speed as a variable. Those studies are all based on data obtained under no wind or low wind speeds. Since wind speed accelerates SSA decay, as first noted by (Cabanes et al. 2002), it is not surprising that both T07 and FL06 underestimate the decay rate observed by the Authors under non-negligible winds. This may be explicitly mentioned.

While we agree that the underestimation of SSA decay rate using T07 can be attributed to the fact that wind influences are not considered, we actually observe an overestimation of the SSA decay rate using FL06. We suggest the following in-text modifications based on the reviewer's suggestion.

Line 330-335: "*T07 (Eq. 4) underestimated the observed SSA decay rate in all the low- and moderate-wind events, except for E18. The largest error is associated with E1, which also had the highest mean wind-speed (6.9 m s$^{-1}$) of all analysed events. The tendency for T07 to underestimate the observed SSA decay can be explained by the additional influence of wind speed, which accelerates SSA decay (Cabanes et al., 2002) but is not considered in either T07 or FZ06. In contrast, FZ06 consistently overestimates the observed SSA decay rate, most pronounced in E10 and E18. The original parameter values τ and n of FZ06 were tuned to data from alpine regions, potentially explaining the poor fit.*"

Lines 333-338. I do not understand the difference between mechanisms 1 and 2. Large grains grow at the expense of small grains by water vapor diffusion, so I just do not see the difference with diffusion of interstitial vapor. Then the following discussion may need to be significantly rewritten. It may be somewhat more sensible to consider the temperature gradient. Elevated gradients drive fluxes throughout layers while low gradients mostly involve short distance vapor transfers less likely to result in isotopic changes. Wind pumping is more likely to result in the largest changes since this is where the exchanges of vapor with the atmosphere will be greatest. In any case, looking at Figure 2, my guess is that wind pumping will be important in all cases so that this will always be the predominant process. I therefore have the feeling that the Authors are on the wrong track and that their classification of events is inadequate. I would be more tempted to consider other criteria, such as perturbations of the snow surface, where the sign of LE would come in.

Here we should first clarify that the first mechanism refers to Ostwald ripening under isothermal conditions - as the reviewer describes in a previous comment. However, we appreciate that this mechanism is very unlikely to occur and therefore propose to modify this section accordingly.

Section 4.2.1

Line 343-378: "*In the absence of snowfall or other surface perturbations, multi-day periods of snow metamorphism – indicated by SSA decay events - correspond to change in snow isotopic composition. The second-order parameter d-excess decreases with SSA through time in most cases, which indicates that the mechanisms driving snow metamorphism also influence the isotopic composition. However, the mechanisms linking these changes are unclear and are not always consistent in space and time. The following section will explore the possible mechanisms driving isotopic change during SSA decay events, by assessing the LE and TG conditions during events with minimal surface perturbations.*

*Two key mechanisms are expected to drive the rapid SSA decay and concurrent change in snow isotopic composition: 1) snow grain growth via diffusion of interstitial water vapour due to near-surface temperature gradients (Colbeck et al., 1983; Ebner et al., 2017; Touzeau et al., 2016), observed to cause a decrease in d-excess and slight increase in δ18O in the defined snow layer (Colbeck et al., 1983; Ebner et al., 2017; Touzeau et al., 2018); 2) grain rounding via sublimation from convex regions of snow grains (Neumann et al., 2004), observed to cause an increase in δ18O and a significant decrease in d-excess of the remaining snow (Ritter et al., 2016; Madsen et al., 2019; Casado et al., 2021; Hughes et al., 2021; Wahl et al., 2021). Sublimation is enhanced by ventilation of the saturated pore air, known as 'wind-pumping' (Neumann et al., 2004). We note that isothermal metamorphism driven by Ostwald ripening causes a decrease in SSA (Ebner et al., 2016), but is associated with only minimal change in bulk isotopic composition. However, conditions that favour Ostwald ripening were not observed in our analysis.*

*Increases in δ18O and decrease in d-excess during E10 can be attributed to a combination of 1) and 2) based on observation of net-sublimation and high amplitude diurnal TG variability over the course of the event. Net-deposition was measured during the period between 9th June at 15:30 UTC and 10th June 10:30 UTC 2018, corresponding to an overall decrease in δ18O, agreeing with previous studies (Stenni et al., 2016; Casado et al., 2021; Feher et al., 2021), and minimal decrease in d-excess, which is not necessarily expected during deposition. However, disequilibrium between water vapour isotopic composition and snow isotopic composition may explain the deviation from expectation (Wahl et al., 2022). Although ground fog was documented on the day preceding E11, no significant deposition is observed in the LE data, indicating the absence of surface hoar formation. The 30% decrease in d-excess concurrent with no change in δ18O suggests strong kinetic fractionation during E11. Continuous variations in δ18O and d-excess throughout June 2018 (Fig. 7) show no clear relationship to total LE or temperature gradients. Field experiments looking at sub-diurnal variability show a stronger dependence of snow isotopic composition on LE (Hughes et al., 2021), potentially explaining the lack of a strong diurnal relationship.*

*Conclusively identifying the mechanisms requires water vapour isotopes to model the fractionation effects. In the absence of this data, we infer potential explanations for isotopic change during the low-wind events. Our analysis suggests that SSA of the surface snow is strongly influenced by surface-subsurface TG and wind speed, while the changes in isotopic composition are likely to be influenced by other factors, such as the magnitude of vapour-snow isotopic disequilibrium during sublimation (Wahl et al., 2022). Decoupling the influence of sublimation and interstitial diffusion within the snow requires additional measurements of isotopic composition of atmospheric water vapour to model associated fractionation effects (Wahl et al., 2022). Our results show that while snow isotopic composition does indeed change during SSA decay events, predicting the magnitude, and even the sign, of the isotopic change associated with snow metamorphism is not possible when information about the interstitial vapour isotopic composition is missing."*

Line 340. Reference to Ebner is incorrect. The discussion paper is mentioned.

We apologise for this oversight; this has now been fixed.

Line 357-359. "We conclude that SSA of the surface snow is strongly influenced by surface-subsurface TG while the changes in isotopic composition are likely to be influenced by other factors such as the magnitude of vapour-snow isotopic disequilibrium during sublimation". The Authors may be right here. But the picture would probably be clearer if the same samples had been used to

measure isotopes and SSA. The different sampling depths really skews the results and make any interpretation uncertain.

The same samples had been used to measure isotopes and SSA (as explained above).

Line 383-385. Please consider that given the windy context, wind pumping is a much more efficient process that temperature gradient-induced diffusion to produce exchanges of water vapor and isotope fractionation in the surface snow.

The following modifications are proposed to explicitly refer to the influence of wind.

Line 411-413: "*Moreover, the inter-annual variability observed at EastGRIP between 2018 and 2019 suggests that precipitation intermittency, temperature (gradients) and wind regimes play a role in isotopic change, which is not readily identified in the surface snow SSA data.*"

Table A1 should have an extra column indicating wind conditions.

We agree that this would be useful and have now added the maximum wind speeds for each event.

**References cited**

Cabanes, A., L. Legagneux, and F. Domine, 2002: Evolution of the specific surface area and of crystal morphology of Arctic fresh snow during the ALERT 2000 campaign. *Atmos. Environ.*, **36,** 2767-2777.

Flanner, M. G., and C. S. Zender, 2006: Linking snowpack microphysics and albedo evolution. *J. Geophys. Res.*, **111,** D12208.

Legagneux, L., A. S. Taillandier, and F. Domine, 2004: Grain growth theories and the isothermal evolution of the specific surface area of snow. *J. Appl. Phys.*, **95,** 6175-6184.

Martin, J. and Schneebeli, M. (2022). 'Impact of the sampling procedure on the specific surface area of snow measurements with the IceCube', *EGUsphere [preprint]*, https://doi.org/10.5194/egusphere-2022-501, 2022.

Gallet, J.-C., Domine, F., Zender, C.S., Picard, G. (2009). 'Measurement of the specific surface area of snow using infrared reflectance in an integrating sphere at 1310 and 1550 nm', *The Cryosphere*, **3:2**, pp. 167-182.

Hughes, A. G., Wahl, W., Jones, T. R., Zuhr, A., Hörhold, M., White, J. W. C. and Steen-Larsen, H. C. (2021). 'The role of sublimation as a driver of climate signals in the water isotope content of surface snow: laboratory and field experiments results', *The Cryosphere*, **15:10**, pp. 4949-4974.

Kokhanovsky, A., Lamare, M., Danne, O., Brockmann, C., Dumont, M., Picard, G., Arnaud, L., Favier, V., Jourdain, B., Meur, E. L.,Di Mauro, B., Aoki, T., Niwano, M., Rozanov, V., Korkin, S., Kipfstuhl, S., Freitag, J., Hoerhold, M., Zuhr, A., Vladimirova, D., Faber, A. K., Steen-Larsen, H. C., Wahl, S., Andersen, J. K., Vandecrux, B., van As, D., Mankoff, K. D., Kern, M., Zege, E., and Box, J. E. (2019). 'Retrieval of snow properties from the Sentinel-3 Ocean and Land Colour Instrument', *Remote Sensing*, **11**, https://doi.org/10.3390/rs11192280.

Vandecrux, B., Box, J. E., Wehrlé, A., Kokhanovsky, A. A., Picard, G., Niwano, M., Hörhold, M., Faber, A. K., and Steen-Larsen, H. C. (2022). 'The Determination of the Snow Optical Grain Diameter and Snowmelt Area on the Greenland Ice Sheet Using Spaceborne Optical Observations', *Remote Sensing*, **14**, https://doi.org/10.3390/rs14040932.

Wahl, S., Steen-Larsen, H. C., Hughes, A. G., Dietrich, L. J., Zuhr, A., Behrens, M., Faber, A-K. and Hörhold, M. (2022 in press). 'Atmosphere-Snow Exchange Explains Surface Snow Isotope Variability', *Geophysical Research Letters*.

Hörhold, M., Behrens, M., Wahl, S., Faber, A-K., Zuhr, A., Zolles, T. and Steen-Larsen, H. C. (2022). Snow stable water isotopes of a surface transect at the EastGRIP deep drilling site, summer season 2018, *PANGAEA*, https://doi.org/10.1594/PANGAEA.945544

Hörhold, M., Behrens, M., Wahl, S., Faber, A-K., Zuhr, A., Meyer, H. and Steen-Larsen, H. C. (2022). Snow stable water isotopes of a surface transect at the EastGRIP deep drilling site, summer season 2018, *PANGAEA*, https://doi.org/10.1594/PANGAEA.945563